# Catalytic trajectory of a dimeric nonribosomal peptide synthetase subunit with an inserted epimerase domain

Jialiang Wang [1,4], Dandan Li [1,4], Lu Chen[1], Wei Cao[1], Liangliang Kong[2], Wei Zhang[1], Tristan Croll[3], Zixin Deng [1✉], Jingdan Liang [1✉] & Zhijun Wang [1✉]

Nonribosomal peptide synthetases (NRPSs) are modular assembly-line megaenzymes that synthesize diverse metabolites with wide-ranging biological activities. The structural dynamics of synthetic elongation has remained unclear. Here, we present cryo-EM structures of PchE, an NRPS elongation module, in distinct conformations. The domain organization reveals a unique "H"-shaped head-to-tail dimeric architecture. The capture of both aryl and peptidyl carrier protein-tethered substrates and intermediates inside the heterocyclization domain and L-cysteinyl adenylate in the adenylation domain illustrates the catalytic and recognition residues. The multilevel structural transitions guided by the adenylation C-terminal subdomain in combination with the inserted epimerase and the conformational changes of the heterocyclization tunnel are controlled by two residues. Moreover, we visualized the direct structural dynamics of the full catalytic cycle from thiolation to epimerization. This study establishes the catalytic trajectory of PchE and sheds light on the rational re-engineering of domain-inserted dimeric NRPSs for the production of novel pharmaceutical agents.

[1] State Key Laboratory of Microbial Metabolism, Joint International Research Laboratory of Metabolic & Developmental Sciences, and School of Life Science & Biotechnology, Shanghai Jiao Tong University, Shanghai, China. [2] National Facility for Protein Science in Shanghai, Chinese Academy of Sciences, Shanghai 201204, China. [3] Cambridge Institute for Medical Research, University of Cambridge, Keith Peters Building, Cambridge CB2 0XY, UK. [4] These authors contributed equally: Jialiang Wang, Dandan Li. ✉email: zxdeng@sjtu.edu.cn; jdliang@sjtu.edu.cn; wangzhijun@sjtu.edu.cn

Nonribosomal peptide synthetases (NRPSs) are multidomain, modular megaenzymes that synthesize various bioactive natural products, ranging from therapeutic drugs (antibiotics, antitumor agents, and immunosuppressants)[1,2] to virulence factors (cancer-causing agents and siderophores)[3,4]. The effectiveness of the catalysis relies on the intramodule and intermodule organization of the assembly lines of NRPSs. These enzymes select specific amino acids, incorporate them into the growing peptide, and modify the peptide backbone in a ribosome-independent manner, and altogether the family of NRPSs are able to use more than 500 precursors[5,6]. These large molecular machines have been widely valued as one of the most promising resources for the production of novel bioactive compounds.

Pyochelin is an iron-chelating siderophore and a virulence factor synthesized by the opportunistic human pathogen *Pseudomonas aeruginosa*[7,8]. The synthetic pathway for pyochelin has been solidly established. Briefly, this product is produced by a typical NRPS assembly line of four subunits: PchD, PchE, PchF, and PchG. PchD activates salicylate, PchE and PchF successively integrate two L-cysteines into the growing peptide to form two thiazoline rings, and PchG then reduces the second thiazoline ring to thiazolidine before the release of mature pyochelin (Supplementary Fig. 1c)[9–13]. After the loading of salicylate by PchD, PchE synthesizes dihydroaeruginoic acid. PchE shares a common domain organization with tailoring domain-inserted NRPSs (Supplementary Fig. 1a, b)[14]. The adenylation domain (A) in PchE selects and activates L-cysteine; aryl carrier protein (ArCP) and peptidyl carrier protein (PCP) connect and transfer salicylate and L-cysteine using phosphopantetheine (pPant) arms, respectively; the heterocyclization domain (Cy) condenses and cyclizes the two substrates to form a thiazoline ring; and the inserted epimerase (E) catalyzes the epimerization of an L-cysteinyl moiety to D-cysteinyl during thiazoline ring formation[9,10].

The knockdown of siderophore biosynthesis partially or completely suppresses the pathogenicity of many different pathogens. As such, elucidation of the structures of siderophore-production enzymes will not only lay a foundation for the rational re-engineering of NRPSs[15] but also clarify these potential drug targets for mechanistic inhibitor development[16].

The current structural knowledge of full NRPS modules was derived from the termination module structures of SrfA-C[17], EntF[18], and AB3403[18], the initial module structures of LgrA[19] that adopt multiple catalytic states, the tri-domain module structure of FmoA3[20], the penta-domain module structure of ObiF[21], the cross-module structure of DhbF[22], and the LgrA di-module[23] structures. Depending on the dynamic $A_{sub}$ and PCP domain positions, the NRPS synthetic cycle has been observed in different catalytic states: adenylation, thiolation, and condensation (or tailoring). In the adenylation state, the $A_{sub}$ domain of AB3403[18] adopts the closed state for substrate activation, and this finding has also been observed with LgrA[19] and ObiF[21]. In the thiolation state of EntF[18], DhbF[22], and LgrA[19], the $A_{sub}$ domain rotates dramatically relative to $A_{core}$ to adopt the thiolation-forming conformation, and the PCP domain contacts the A domain for loading the activated substrate. Subsequently, in the condensation or tailoring state, the PCP domain with its pPant-tethered substrate binds at the C domain (or formylation domain of LgrA[19]) for peptide bond formation or formylation, whereas the $A_{sub}$ domain is restored to the adenylation-state position for the next round of substrate activation. Together, these crystal structures have provided excellent insights into how these enzymes are assembled, catalyze reactions, and are structurally rearranged during catalytic state transitions[24].

However, the currently resolved NRPS module crystal structures serve as steady snapshots that do not provide a direct view of conformational transitions. In PchE with an inserted domain, the E domain plays a key role in ensuring the proper stereochemistry of the intermediate, and such tailoring domains have been found in various NRPS enzymes and serve as a natural tactic for the diversification of nonribosomal peptides[14,25]. The structure of this noncanonical epimerase and the mechanism through which it integrates into the structural dynamics during elongation catalysis are unknown. More specifically, for the Cy domain, the interactions of pPant-tethered substrates with the catalytic tunnel and substrates with catalytic residues have not been fully clarified. Moreover, despite the structurally solved dimeric FmoA3 (by both EM and crystallography)[20] and StsA (in crystals)[26]; the biochemically identified dimeric VibF (by ultracentrifugation)[27] and SfmC (by BN-PAGE analysis)[28], the NRPS dimer formation has remained a minority nature in the field. Understanding the organization of E domain-embedded NRPS dimerization is hampered by the lack of such a dimeric architecture.

In this work, we determine the structures of PchE at 2.97 Å, 3.78 Å, and 3.47 Å resolutions in three distinct catalytic states using cryo-electron microscopy. The structure of PchE adopts an "H"-shaped head-to-tail dimeric architecture containing five connected catalytic domains. By binding with the ArCP and PCP tethered ligands, the Cy domain reveals the detailed constitution of substrate tunnel. The switch of the tunnel is controlled by F372, and the heterocyclization reaction is catalyzed by D480, Q482 and (or) T474. The L-cysteine-binding pocket of the A domain is identified, and the catalytic states transition from adenylate-forming to thioester-forming is guided by the large-scale rotation of the $A_{sub}$ domain, which is interrupted by the E domain that performs the epimerization reaction. Additionally, we visualize the complete molecular trajectory across different states using the deep neural network to analyze the cryo-EM dataset. Together, our observation provides substantive insights into the dimeric PchE catalytic cycle, which is collaboratively contributed by multilevel structural dynamics.

## Results

**PchE forms a homo-oligomeric quaternary structure**. PchE and PchF were previously successfully purified with biochemical activities[9] reconstituted by the Christopher T. Walsh lab, and the predicted (theoretical) monomer molecular weights of the two subunits were approximately 156 kDa (PchE) and 197 kDa (PchE) (Supplementary Fig. 2a). However, the size exclusion chromatography profile clearly shows that the apparent molecular weight of PchE is larger than that of PchF, and the native-PAGE analysis also supports this observation (Supplementary Fig. 2b, c). Based on these experiments, we speculated that PchE exhibits a homo-oligomeric organization. To gain further insight into this enzyme, we determined the structure of PchE by cryo-EM analysis.

**Overall architecture of PchE**. We purified full-length His-tagged PchD and Strep II-tagged PchE separately (Supplementary Fig. 2d, e). PchD is responsible for loading salicylate onto the ArCP domain of PchE. We then mixed PchD and PchE at a stoichiometric ratio of 0.5:1 in the presence of all the substrates and cofactors. PchE was then further purified using a final size-exclusion column (Supplementary Fig. 2f). All the substrates were added again into the PchE solution for cryo-EM sample preparation. We collected cryo-EM images with a K2 Summit direct electron detector equipped on a Titan Krios electron microscope, and RELION was used for image processing[29–34]. Rounds of 2D and 3D classification for particle selection and refinements were performed (Supplementary Fig. 3). Three maps of PchE at overall resolutions of 2.97 Å, 3.78 Å and 3.47 Å were ultimately

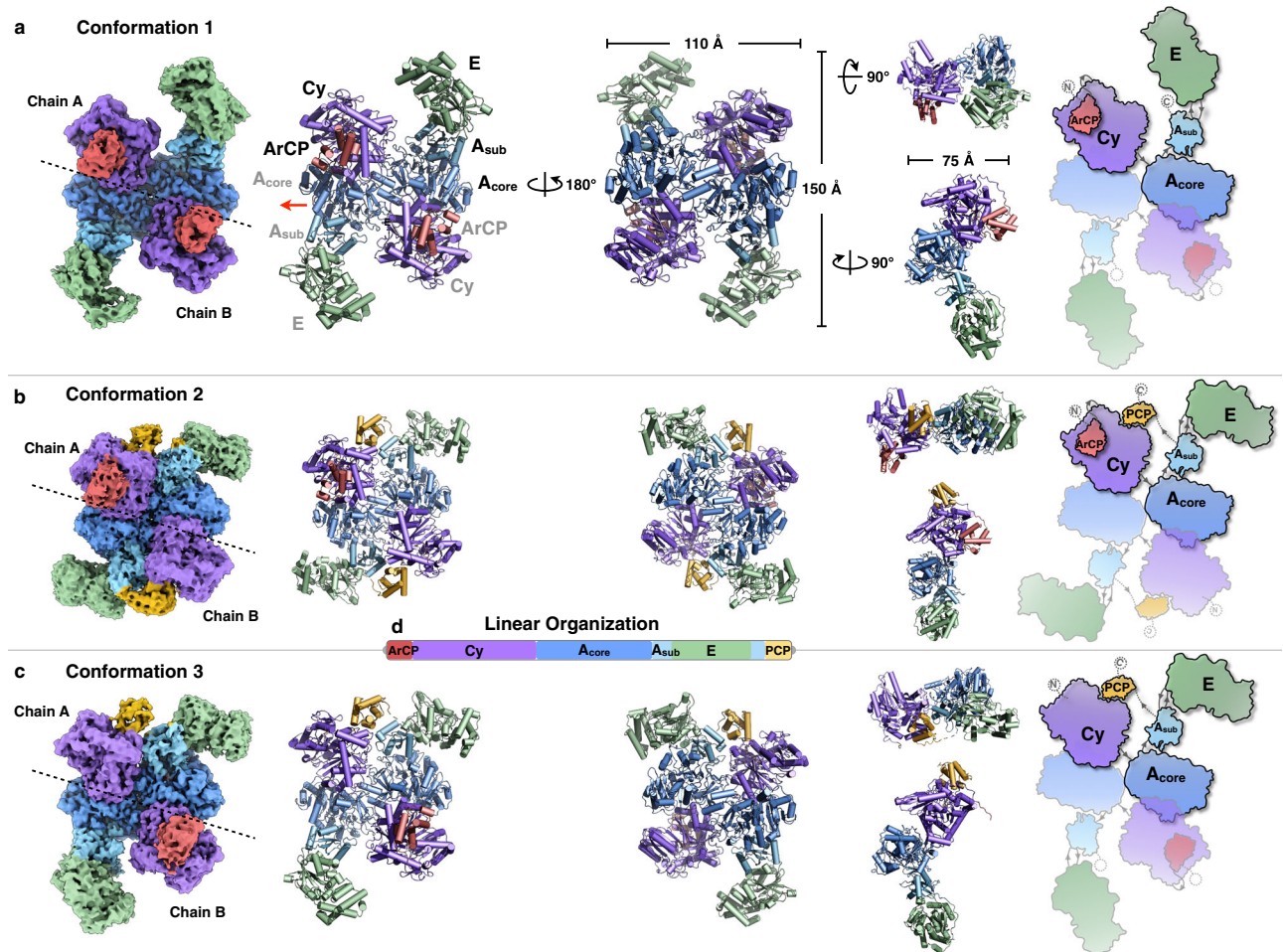

**Fig. 1 Overall architecture of the dimeric PchE protein.** The linear domain organization of PchE is shown at an approximate sequence scale, and each domain is colored a unique color (**d**). Left panel: Cryo-EM maps of PchE at overall resolutions of 2.97 Å (**a**, conformation 1), 3.78 Å (**b**, conformation 2), and 3.47 Å (**c**, conformation 3). Each chain is labeled by dashed lines. Middle panel: Atomic models of PchE are shown in front, back, top and side views, and the dimensions are indicated. The pseudo-twofold symmetry axis is indicated by a red line with an arrow. Right panel: Schematic diagrams illustrating the "H" shape of the domain arrangements of PchE.

reconstructed (Supplementary Figs. 4 and 5). The resolution of the maps enabled us to reliably assign the individual domains, dissect their linker junctions, identify the substrates and cofactors, and build atomic models of PchE (Supplementary Figs. 6 and 7).

The dimeric PchE adopts an "H" shape in the front and back views, and two monomers contact each other in a head-to-tail orientation (Fig. 1, left). The width of PchE is approximately 110 Å. Although the monomer alone has a height of ~110 Å, the PchE dimer extends to a height of ~150 Å. The thickness of PchE is ~75 Å (Fig. 1a, middle). Starting from the N-terminus of one monomer, the ArCP sits on the Cy domain, which is then connected to the $A_{core}$ domain. After completing the A domain, the $A_{sub}$ domain is interrupted by the E domain, a noncanonical epimerase that catalyzes the epimerization of L-cysteinyl to D-cysteinyl. The C-terminal PCP domain is clearly resolved in conformation 3 of PchE (Fig. 1c).

All PchEs in the three conformations are dimeric which each of them is composed of two chains. The stable Cy-$A_{core}$ didomain forms the main body of the chain, which is nearly identical to the FmoA3[20] Cy-$A_{core}$ domain arrangement. The Cy-$A_{core}$ didomain is similar to the common C-$A_{core}$ interactions of the previously characterized NRPSs[17,18,21,23] with slightly varied rotational ranges of the positions of the C domain (Supplementary Fig. 8). Relative to Cy-$A_{core}$, there are multiple highly dynamic domains,

namely, the ArCP, PCP, and $A_{sub}$-E-$A_{sub}$ domains, and these domains allow the identification of different catalytic states. In conformation 1 (Fig. 1a), chains A and B are identical with an r.m.s.d. of 0.037 Å. The ArCP domains of both chains bind at the donor sites of the Cy domains, with the salicyl-pPant ligands observed, representing the substrate donation states. The $A_{core}$ with interrupted $A_{sub}$ domains (complexed with Cys-AMP and $Mg^{2+}$ ligands) are in the thioester-forming conformation but without the PCP domain captured. The E domain in this conformation may represent the epimerization conformation because the low-resolution map of the PCP domain was only observed binding to the E domain in this position (described later in Fig. 5). In conformation 2 (Fig. 1b), both ArCP and PCP domains in chain A bind to the Cy domain, representing the condensation state; the pPant arm of the ArCP domain was observed without tethered salicylate, and the pPant of PCP is merely modeled in Fig. 2b. In chain B, only the PCP domain contacts the acceptor site of the Cy domain, and the pPant arm of the PCP domain was obtained. Both chains have their $A_{sub}$ domain rotated relative to the $A_{core}$ (complexed with AMP), resulting in the adenylate-forming conformations. The superposition between two chains gives an r.m.s.d. of 0.851 Å (without ArCP). In conformation 3 (Fig. 1c), the PCP domain of chain A binds at the acceptor site of the Cy domain, with the

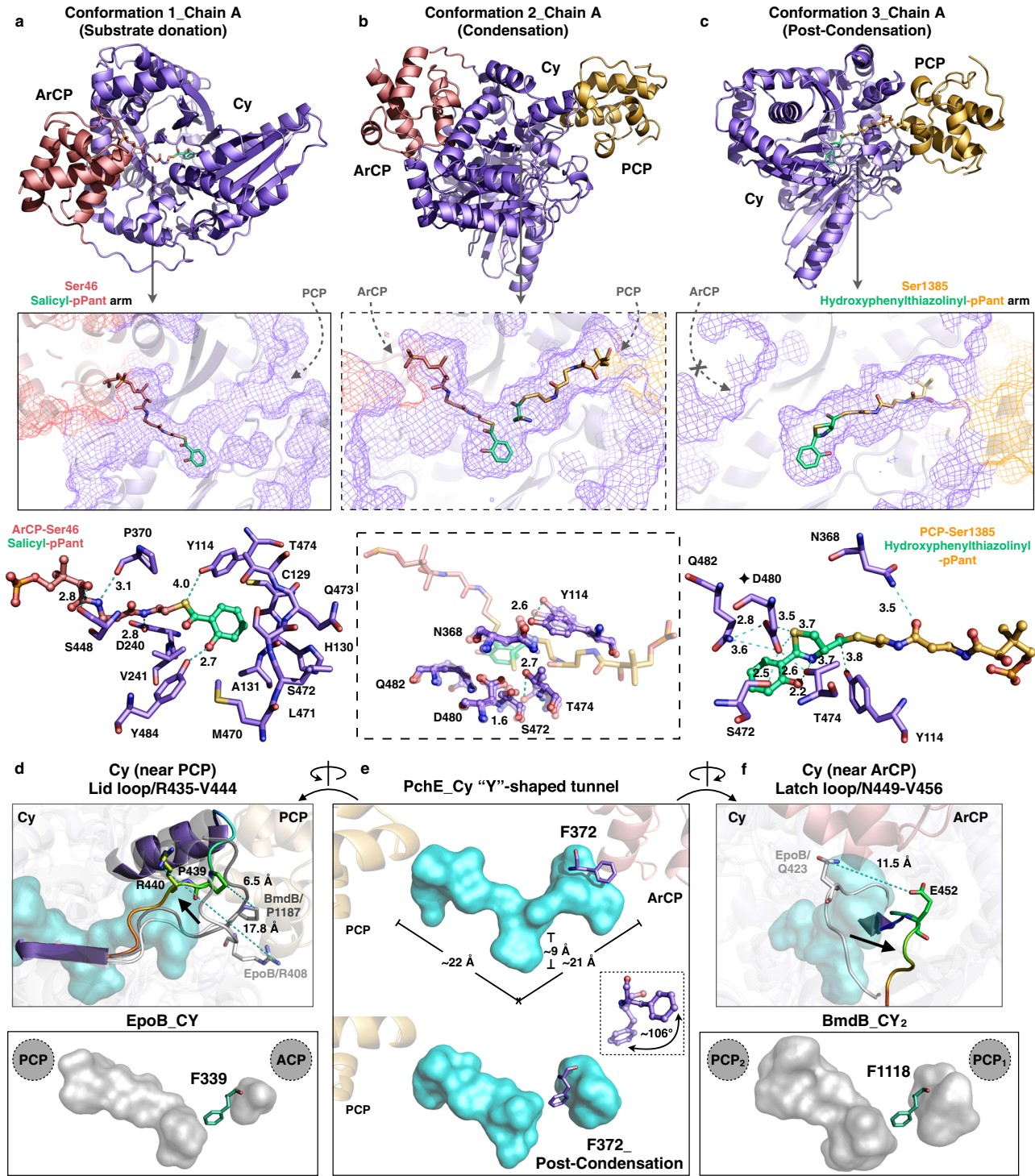

hydroxyphenylthiazolinyl-pPant ligand observed, representing the post-condensation state. The A domain (complexed with Cys-AMP and $Mg^{2+}$ ligands) adopts the adenylate-forming conformation. Chain B duplicates the catalytic state as both chains of conformation 1. In addition to the dynamic domains, the Cy-A$_{core}$ domains between two chains of conformation 3 have an r.m.s.d. of 0.566 Å. In summary, because each protomer of the three conformations is observed in the different catalytic states, dimeric PchE may still be monofunctional[35], with two chains forming separate catalytic units.

The architecture of PchE uncovers a basic model for dimeric NRPS organization. Rather than being planar, the two monomers exhibit a slightly curved "arch"-shaped dimeric structure. The two chains contact each other extensively with an interface of approximately 2337 Å² which is entirely contributed by the Cy and A$_{core}$ domains (Supplementary Fig. 9a–c). The interface is derived from the reverse interaction of Cy-A$_{core}$ didomains through both homophilic and heterophilic interactions, which involves nearly fifty residues (49) per chain (Supplementary Fig. 9d). The ~800-Å² surface buried between two chains and the ~668-Å² buried surface of internal Cy-A$_{core}$ interactions within one chain leads to tight formation of the quadrangular stable core comprising (Cy-A$_{core}$)$_{chain A}$-(A$_{core}$-Cy)$_{chain B}$, with the overall antiparallel organization of the Cy-A$_{core}$ pair, which resembles a

**Fig. 2 Substrate donation, condensation and PCP post-condensation conformations adopted by the Cy domain.** The ArCP, PCP and Cy domains of the atomic models are shown as ribbon representations (top), close-up views of salicyl-pPant and hydroxyphenylthiazolinyl-pPant arms binding to the tunnel of Cy are presented as mesh (middle), and detailed views of the catalytic residues are shown (bottom). The substrate precursors and the intermediate are highlighted in green. Potential hydrogen bonds are indicated with cyan dashed lines, and the distances are marked by numbers (Å). **a** Substrate donation conformation. **b** Condensation conformation. Bordered with dashed lines, the putative salicyl/cysteinyl-pPant arms are merely modeled in the tunnel without steric clash based on the superimposition with the pPant arms of two structures in **a** and **c**. The catalytic residues are aligned to show changes in position (marked with distance, Å) between the three conformations. **c** PCP post-condensation conformation. Structural comparisons of the Cy domain with its Cy homologs EpoB and BmdB, which focus on the conformations of the catalytic tunnels, are shown as surfaces. The structures are superimposed, the shifting loops are rainbow colored, and the distances between residues are labeled (per homologous pair). The tunnel of PchE Cy is colored cyan, whereas the tunnels of its homologs are colored gray. **d** The PCP side of the tunnel with the R435-V444 lid loop (rainbow colored) of Cy of PchE reveals its motions compared with the loops in its homologs. **e** The ~106° switchable residue F372 controls the tunnel conformation. The ArCP and PCP domains are shown as ribbons and mark the substrate entry points. The tunnel is a complete "Y" shape in the structure in which both carrier proteins contact Cy and F372 points away from the tunnel. In the structure in which ArCP departs the Cy domain, F372 is oriented in the opposite direction, which blocks the tunnel on the ArCP entry side (see also Supplementary Movie 1 for more visualization) and causes the disconnected tunnel to be similar to the tunnels in the EpoB and BmdB structures (tunnels shown in gray). **f** The ArCP side of the tunnel.

clip (Supplementary Fig. 9e). The $A_{sub}$-E-$A_{sub}$ didomains then protrude from the stable core and complete the individual protomer as a swing arm.

A comparison of the overall architectures between PchE and the very recently reported FmoA3[20] revealed similar head-to-tail dimeric arrangements (Supplementary Fig. 10a). The dimer formations are entirely contributed by the stable Cy-$A_{core}$ didomain interaction, whereas the dynamic domains of $A_{sub}$ and PCP for FmoA3 and the ArCP, PCP, and $A_{sub}$-E-$A_{sub}$ for PchE make no contact with the dimer interface. However, the relative angle between the two protomers is different. The interface helices (S473-N484) of FmoA3 are roughly parallel (170.3°), and this finding is distinct from that obtained for corresponding helices (P534-G546) of PchE, which adopt a "V"-shaped arrangement with an interhelix angle of 141.8° (Supplementary Fig. 10b). Subsequently, rather than the coplanar dimer of FmoA3, one Cy-$A_{core}$ protomer of PchE bends in the opposite direction relative to the other one, resulting in an overall tilted conformation (Supplementary Fig. 10c). Consequently, the number of interacting residues and the buried surface between the two protomers of PchE is more than twice that of FmoA (Supplementary Fig. 10d). Together, PchE forms a more complicated and compact dimeric structure than that of FmoA.

**Structural analysis of PchE enzymatic domains.** Similar to all the C and Cy domain structures[17,18,20,23,36–38], the PchE Cy domain is a "V"-shaped pseudodimer with two lobes, each of which adopt the chloramphenicol acetyltransferase fold (Fig. 2). The superposition of PchE Cy with known Cy domains[20,36,37] shows a subtle motion of the overall structure, whereas the comparison of PchE Cy with C domains[17,18,23,38] displays different rotational offsets and varied ranges of "openness"[38] between the N- and C-lobes (Supplementary Fig. 11a, b). The ArCP and the PCP contact the Cy domain on the two opposite sites, mainly through α2 and α3 of both carrier proteins. The overall PchE ArCP:Cy:PCP binding mode is similar to that of the homologs[18,23,39] with the orientation of CPs rotated slightly (Supplementary Fig. 12a–c) but is distinct from the SrfA-C[17] and FmoA3[20] at the acceptor CP sites in which the pPant-attached serine residues of PCPs mutated to alanine (Supplementary Fig. 12c). A comparison between PchE Cy structures with or without ArCP/PCP revealed that the interacting residues of Cy move slightly, indicating the subtle flexibility of these interface residues (Supplementary Fig. 13). The tunnel surface for salicyl-pPant binding was clearly identified in the A chain of conformation 1 (Fig. 2a). In particular, the D240, P370 and S448 residues are in the range for hydrogen bond formation with the pPant arm atoms, and Y114 can form a polar interaction with the

thioester sulfur. The residues C129, H130, A131, L236, L471, Q473, and M470 form the salicyl moiety-binding pocket, and an additional Y484 residue forms a hydrogen bond with its hydroxyl group. These interactions position the salicyl-pPant donor in place for the condensation reaction.

Key catalytic residues interacting with hydroxyphenylthiazolinyl-pPant can be clearly recognized in the A chain of conformation 3 (Fig. 2c). N368 can form a hydrogen bond with the pPant arm atom. The catalytic residue D480 aligns well with D1226 and D449 in BmdB[36] (r.m.s.d. 1.837 Å) and EpoB[37] (r.m.s.d. 2.127 Å) Cy homologs, respectively. Notably, D480 may form a complex hydrogen bond network with S472, T474, Q482, and Y114 to form polar contacts with the thiazoline ring atoms.

A comparison of the conformations of the catalytic residues among the substrate donation, condensation and PCP post-condensation states revealed that D480 and Q482 are relatively static; however, S472, Y114 and T474 move between the three catalytic states, and the hydroxyl groups show distances of 1.6 Å, 2.6 Å and 2.7 Å, respectively (Fig. 2b).

A typical Cy domain catalyzes two separate reactions: condensation and a two-step cyclodehydration. Based on mutational analyses of BmdB and EpoB, an N residue was proposed to facilitate the positioning role in the condensation reaction, and D and T residues were identified as crucial for the heterocyclization reaction[36,37]. The significance of these active site residues for the Cy domain of PchE was also confirmed by site-directed mutagenesis (Supplementary Figs. 14–17). We used SrfD, an external thioesterase enzyme involved in surfactin biosynthesis by *Bacillus subtilis*[40], to release linear and heterocyclic products. Mutations of N368, Q482, S472, Y114 and T474 to alanine reduced the heterocyclic product of PchE to approximately 28.0%, 9.8%, 50.3%, 4.6%, and 15.3%, respectively, whereas the linear intermediate production of Q482A and T474A tripled. The mutation of D480A almost abolished the peptide formation activity and was particularly detrimental to the cyclization reaction, as indicated by the substantially large ratio of the linear to heterocyclic product (Supplementary Fig. 14a). Thus, consistent with previous mutagenesis analyses[36,37], D480 may act as a base and abstract a proton from the thiol sidechain of L-cysteinyl; the carbonyl oxygen could then abstract a proton donated by T474 or Q482, which is near the conserved D480, compensates for the deprotonation role of T1196 in the BmdB Cy domain, and ultimately forms the thiazoline ring. The catalytic residues identified in this study indicate the versatility of the active site of Cy domains. By interacting through polar contacts with pPant arm atoms or the substrate precursors, the other catalytic residues perform structural positioning functions.

The Cy domain reveals a complete "Y"-shaped substrate-binding tunnel traversing the two lobes (Fig. 2e, top). The short stem region of the tunnel has a height of ~9 Å, and the two long arm regions have a length of approximately 21 and 22 Å for salicyl/cysteinyl-pPant arm binding. Each tip of the "Y" of the tunnel represents the ArCP/PCP contact regions. By overlaying Cy with the two known Cy structures, we observed that the lid loop shifted ~6.5 Å (compared with BmdB[36]) or ~17.8 Å (compared with EpoB[37]) on the side of the PCP-binding region; the opposite loop (the latch[41]) moved ~11.5 Å on the side of the ArCP-binding region compared with EpoB[37] (Fig. 2d, f, respectively). Although there is no evidence showing that the lid loop (near PCP) and the latch open and close[38], the translations between the two loops with the Cy homologs reveal their conformational dynamics.

Intriguingly, in the PCP post-condensation conformation, in which ArCP departs from the Cy domain, the tunnel adopts a similar conformation previously observed with both EpoB and BmdB. In the BmdB Cy structure, the residue F1118 completely blocked the tunnel. It was previously proposed that F1118 must move to allow alanyl-PCP to bind and subsequently participate in the condensation reaction[36]. Similarly, the "Y" tunnel of PchE Cy is truncated at the side of ArCP by the residue F372. In contrast, this residue is rotated ~106° away from the tunnel and restores the tunnel to the "Y"-shaped conformation, which allows salicyl-pPant to enter the Cy domain in the substrate donation and condensation conformations (Fig. 2e, Supplementary Movie 1). The mutagenesis of F372A reduced the activity of PchE to 31.6%. Together, the results indicate that the residue F372 may play a role in changing the catalytic state in the Cy domain of PchE by switching the closed tunnel to open for donor ArCP-attached substrate interactions and the subsequent condensation reaction.

Similar to the known adenylation domain structures[17–19,42,43] (Supplementary Fig. 18a), the A domain comprises two subdomains: the small $A_{sub}$ together with the connected epimerase rotates dramatically depending on the catalytic states from conformation 1 (chain B) to 3 (chain A), and the large $A_{core}$ is responsible for cysteinyl-AMP and $Mg^{2+}$ binding (Fig. 3a). Near cysteinyl-AMP, multiple side chain conformational variations of residues have been resolved, and these include those of R942, R1328, and R1329, which indicate the small-amplitude dynamics of these arginine residues (Fig. 3f). Although it existed transiently, the native cysteinyl-AMP ligand within the internal pocket of the $A_{core}$ domain was captured by cryo-EM (Fig. 3b). The structure-projection method successfully predicted the specificity-conferring code[44] for L-cysteine recognition and yielded the DLFNLSLIWK 10 AA signature sequence[45], and the residues in the cysteine-binding pocket were identified as F741, D742, L743, S813, G814, A841, T842, I846, W847, and K1325 (Fig. 3c). The active site of PchE is closely similar to that of the AB3403[18] A domain, which contains two pairs of hydrophobic interaction residues, namely, I846:W847 (V768:W769 for AB3403) and F741:L743 (F669:I671 for AB3403), whereas the latter pair does not exist in the LgrA[19], DhbE[42], PheA[43], and SrfA-C[17] A domains (Supplementary Fig. 18b). Compared with the homologs, the two pairs of hydrophobic interactions narrow the pocket to create a specific cysteine sidechain-binding surface.

The domain alteration strategy of the adenylation domain has been firmly established[46]: the $A_{sub}$ domain rotates relative to $A_{core}$ to adopt distinct catalytic states for adenylation or thiolation reactions. Similar to the known states of homologs[18,19,47,48] (Supplementary Fig. 19), in PchE, a large-scale ~155° $A_{sub}$ rotation is mediated by the hinge residue[49] D944 and coupled to catalytic state switching (Fig. 3a). Chain A of conformation 3 adopts the adenylate-forming conformation, as featured by the universally conserved residue[50] K1325 in the A10 motif[51], which forms a hydrogen bond with the 5' bridging oxygen of Cys-AMP (Fig. 3d), similar to the results found for the A domain of the DhbE[42] structure. In the thioester-forming conformation of conformation 1_chain B, the aromatic residue F741 in the A4 motif[51] was observed in two side chain conformations with a rotation angle of ~68° (Fig. 3e), similar to the H207 movement of 4CBL[52] crystal structures. When the side chain of F741 points toward the active site, the pantetheine tunnel is closed, whereas the tunnel is restored to the open state when this aromatic residue rotates out of the active site, making the tunnel ready for the subsequent PCP-pPant entering. The observation of two F741 conformations in one chain indicates that these two states coexist in the map of the thioester-forming conformation, and these states were not completely separated in our cryo-EM dataset.

In PchE, the A domain is interrupted by an ~32.0-kDa tailoring domain: the noncanonical epimerase. Rather than an independent domain generally following the carrier protein, the embedded E domain is inserted into the $A_{sub}$ domain between the A8 and A9 motifs[51] (Fig. 4a, b), and this domain followed the residue P990 and located just before the residue D1291. Similar to the insertion point and noncovalent interactions of the MT domain-embedded TioS[53] structure, the PchE E domain significantly contacts the small $A_{sub}$ domain with a buried surface of 865 Å$^2$ (Supplementary Fig. 20b-f). Due to this architecture, the large-scale $A_{sub}$ domain rotation results in the dramatic backward swinging of the E domain (Fig. 3a), and the E and $A_{sub}$ domains rotate as a rigid body (Supplementary Fig. 20a).

Rather than the canonical form[54], which belongs to the C domain superfamily proteins, the PchE E domain is more structurally similar to the methyltransferase (MT) domain[53,55,56] (Supplementary Fig. 21). The E domain comprises two subdomains: the N-terminal α-helix bundle and the C-terminal typical Rossmann α/β-fold (Fig. 4c). A cleft between the two subdomains has been clearly identified and represent the substrate-binding tunnel and mark the entry region for the pPant arm of PCP. H1204, which is located in the middle of the tunnel, has been proven essential for this activity[10]. The mechanism of the catalysis of the epimerization reaction by canonical epimerization domains has been well investigated and proposed[54,57,58]. Briefly, a pair of histidine and glutamate residues function as the general acid-base catalyst. In PchE, a general acid residue needed for the mechanism has not been pinpointed due to the lack of an embedded epimerase structure. In the search for acidic residues, we found E1234 located at the end of the tunnel. Nonetheless, the H1204 and E1234 residues likely act as general acid and base and catalyze the formation of D-thiazoline, which mimics the mechanism observed with the epimerization domain from the initiation module of *B. brevis* gramicidin S synthetase[54] and tyrocidine synthetase A[58].

**Catalytic trajectory of PchE**. To investigate the dynamic transition between distinct structures and explore additional potential catalytic states of PchE, we used cryoDRGN, a neural network calculation method[59], to analyze the structural heterogeneity of PchE (focused on one chain). By creating neural networks based on cryo-EM particle images, cryoDRGN is able to encode particle images in a latent variable that describes these images using several dimensions, learns the underlying structures including any motions, and decodes them into three-dimensional volumes (structures), including any structural heterogeneity in the cryo-EM dataset. Because cryo-EM images themselves do not contain temporal and kinetic information, cryoDRGN does not inform the kinetic order of conformational changes. Nevertheless, this method provides greater insights than just the single static structure. After training an 8-dimensional latent variable model

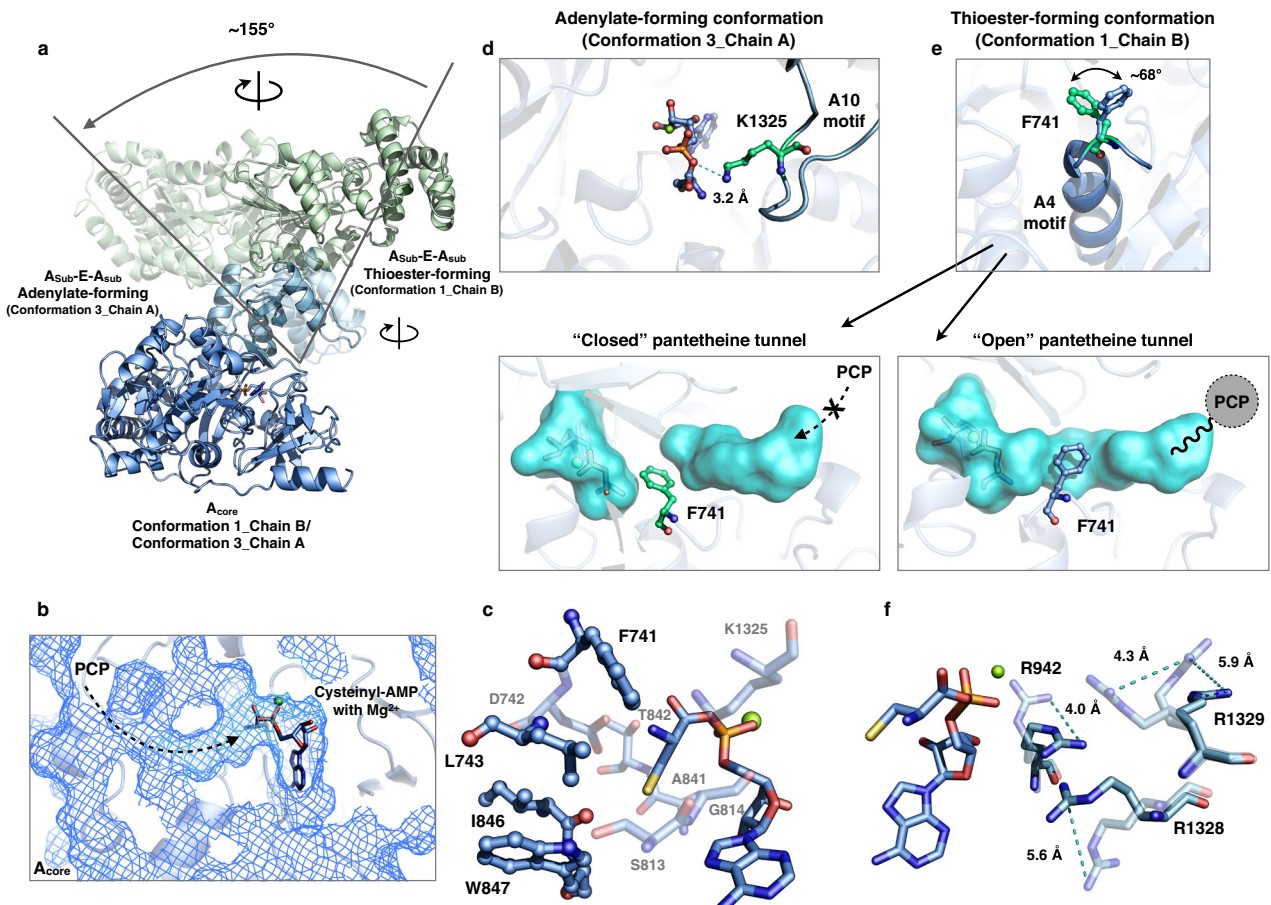

**Fig. 3 Conformational changes coupled with A$_{sub}$ domain rotation. a** Mediation of conformational changes by the A$_{sub}$ rotation relative to the A$_{core}$ domain. Structures of the A (blue) and E domains (green) in conformation 1_chain B and conformation 3_chain A are superposed and shown as ribbon representations to reveal the large-scale rotation of the A$_{sub}$-E-A$_{sub}$ didomains. **b** Close-up view of cysteinyl-AMP with Mg$^{2+}$ binding to the substrate pocket of the A domain (mesh) in conformation 1_chain B with the open tunnel and a detailed view of the binding site residues (**c**, sticks) in conformation 3_chain A. **d** In the adenylate-forming conformation, the K1324 in the A10 motif forms the hydrogen bond with the oxygen of Cys-AMP; the distance is labeled. **e** Alternative side-chain conformation of F741 in the thioester-forming conformation (conformation 1_chain B). The ~68° switchable F741 in the A4 motif causes the pantetheine tunnel to transition from closed (F741 colored green) to open (F741 colored blue). The tunnel is shown as a cyan surface, and the direction of pPant entering is indicated. **f** Alternative side-chain conformations of residues R942, R1328, and R1329 (near cysteinyl-AMP) are shown with distances.

on our dataset images using image poses derived from the 3.67-Å consensus reconstruction map (Supplementary Fig. 3), we visualized the PchE particles with uniform manifold approximation and projection (UMAP)[60] and principal component analysis (PCA)[61–63] (Supplementary Fig. 22a–c). By sampling structures from the PCA projections, a total of 6 representative cryo-EM maps (i to vi) were reconstructed from our EM dataset, and we additionally discovered the PCP map docking at different locations (Supplementary Fig. 22d), including the C-terminal site followed by the A$_{sub}$ domain (i), the A$_{core}$ domain (ii), and the E domains (vi). Furthermore, by sampling the latent space, we generated a total of 120 maps traveling through the path connecting conformations i to vi for trajectory visualization (Supplementary Movie 2). These cryo-EM maps allow the identification and in silico docking of each individual PchE domain structure.

Based on biochemical knowledge and structural information of both static and dynamic conformations, we proposed the trajectory of the PchE catalytic cycle (Fig. 5, Supplementary Movie 2). The cycle initiates from state i. At this state, ArCP contacts the Cy domain at the donor site, representing the substrate donation conformation, the A domain adopts the

thioester-forming conformation, and PCP is resting on the C-terminus site (i). PCP then moves and binds to the A domain, which adopts the thiolation state for loading the L-cysteine onto the pPant arm of the PCP domain (ii). The A$_{sub}$ domain then rotates dramatically together with the embedded tailoring E domain to adopt the adenylate-forming conformation with catalytic A10 motif[51] lysine present in the active site for activation of the next L-cysteine molecule (iii), which is the same conformation as that found in the iv and v states. For the subsequent condensation reaction to occur, the PCP domain moves and binds at the acceptor docking site of Cy. The Cy domain catalyzes condensation and heterocyclization (iv). Subsequently, ArCP departs from the donor docking site on the Cy domain (v), quite likely interacts with upstream PchD to load a new molecule of salicylate, returns to Cy in conformation vi, restores the substrate donation conformation, and prepares to donate for the next round of condensation. At state vi, the A$_{sub}$ domain rotates back to the position of the thioester-forming conformation. Due to the significant surface contact between A$_{sub}$ and the E domain, the embedded epimerase simultaneously rotates as a rigid body; PCP shifts to the docking site of the E domain (vi), altering the stereochemistry of the thiazoline moiety.

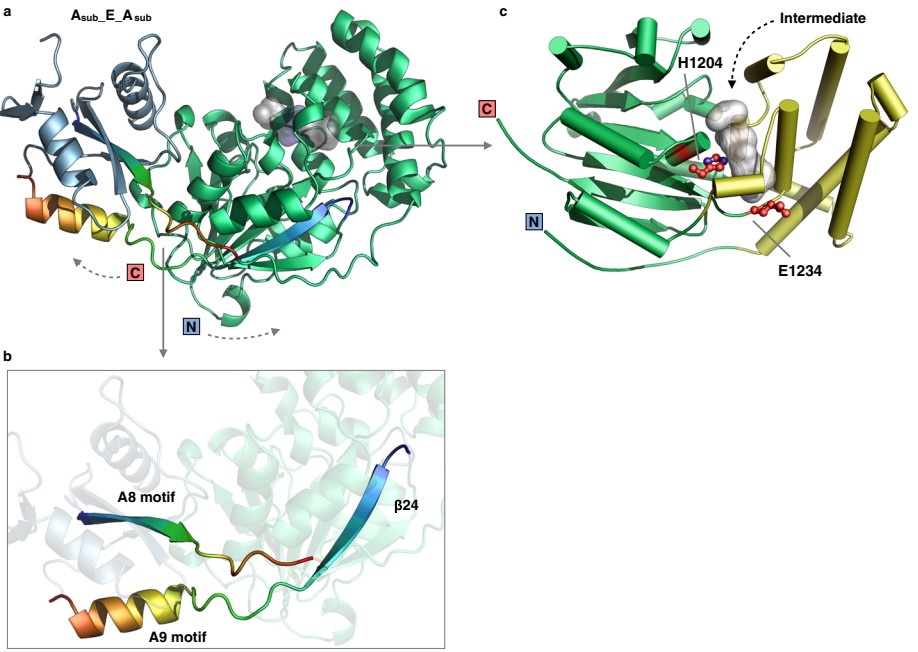

**Fig. 4 Inserted epimerase domain. a** Structure of $A_{sub}$-E-$A_{sub}$ domains. The N and C termini are labeled. The insertion of the E domain causes interruption of the $A_{sub}$ domain. **b** Close-up view of the insertion region (rainbow colored) of the E domain, which starts from the A8 motif of the $A_{sub}$ domain and ends with the A9 motif back to $A_{sub}$. **c** The topology of the E domain shows the overall two-subdomain organization with the substrate tunnel (gray surface) located at the interface. The active site residues H1204 and E1234 are shown as sticks.

After delivery of the intermediate to the downstream PchF, the PCP domain returns to the C-terminus (i) and is then ready to begin the next round of the catalytic cycle.

## Discussion

Among the NRPS family of megaenzymes, homodimerization is a minor and unique phenomenon, and architecture information is indispensable for the engineering of chimeric NRPS modules. Compared with the very recently reported dimeric FmoA3[20] structure, PchE exhibits a similar antiparallel head-to-tail architecture. However, the curved PchE dimer results in a significantly more extensive interface between the two Cy-$A_{core}$ protomers connected with additional moving domains (ArCP, PCP, and $A_{sub}$-E-$A_{sub}$). The Cy-$A_{core}$ dimer is rigid and stable in all the conformations observed in this study, and the observation of two carrier protein condensations within one monomer indicates that they are most likely monofunctional, with each monomer catalyzing reactions independently. The overall structural arrangements of FmoA3 and PchE complement each other and provide a better understanding of NRPS homodimerization.

This study reveals that PchE combines multiple levels of structural rearrangements to achieve the catalytic cycle. The domains move dramatically at different catalytic states. The tunnels exhibit changing shapes and conformations to control the entry of the pPant arm and small substrate molecules. The catalytic residues are shifted to mediate the enzymatic reactions.

Even though the particles of different classes might occupy a small fraction of the total, the large-scale conformational changes can be effectively detected by cryoDRGN calculation. Using this neural network calculation, we further modeled the thiolation and epimerization conformations and ultimately visualized the full catalytic cycle (Fig. 5). The cycle presented in Fig. 5 is also supported by previously well-established biochemical conclusions[9,10]. Although cryoDRGN by itself does not necessarily reveal the kinetically preferred path of the trajectory, this molecular trajectory certainly provides insight into the dynamic transition process between catalytic states. The structural basis of dimeric PchE with the active site details revealed in this study highlights how this megaenzyme delicately processes but still controls the elongation steps.

## Methods

**Strains, plasmids and culture conditions.** The strains, plasmids, and primers used in this work are listed in Supplementary Tables 2 and 3. The *Escherichia coli* strains were grown in liquid or solid Luria broth (LB).

The DNA fragments of the *pchE* gene (4317 bp) and *pchD* (1645 bp) were amplified using the primers fPchE28_S, fPchE28_A, fPchD28_S and fPchD28_A, respectively, and total DNA of *Pseudomonas aeruginosa* PAO1 as the template. The vector pET28a was amplified using the primers V28_PchE_S, V28_PchE_A, V28_PchD_S and V28_PchD_A. The resulting DNA fragments were fused with an 18-bp terminus overlapping sequence together using a Trelief™ SoSoo Cloning Kit (Tsingke Biological Technology). Transformation of the fusion product into *E. coli* DH10B generated the expression plasmids pPchE and pPchD_nH. For construction of the pPchE_cHcSII plasmid with two 6X histidine and StrepII tags synthesized sequences (Supplementary Table 3, Tag_cHcSII_S and Tag_cHcSII_A) with an 18-bp overlapping sequence between the C-terminus of the *pchE* gene end and vector bone, pPchE was used as a template using the primers V_PchE_cHS_S and V_PchE_cHS_A to obtain a PCR fragment with the aim of using annealed two-tag sequences. These sequences were fused at the C-terminus with 6X His and StrepII tags of the protein (159 kDa). PchE mutants were constructed using pPchE_cHcSII as a template with a circular PCR method to amplify fragments and primers with a partial target mutated base pair (Supplementary Table 3, base underlined), and the PCR products were digested with DpnI separately and then directly transformed into *E. coli* DH10B to screen the correct plasmids and the resulting plasmids. The primers were listed in Supplementary Table 3. All resulting PCR products obtained using plasmids as templates were digested with DpnI to remove the template and then transformed into *E. coli* DH10B. The resulting plasmids were extracted from *E. coli* DH10B. After confirmation by sequencing, pPchE_cHcSII and derived mutational plasmids were transformed into *E. coli* BAP1 for protein expression.

Bacterial cells were routinely cultured in LB medium (10 g/L tryptone, 5 g/L yeast extract, and 10 g/L sodium chloride) supplemented with 50 μg/mL kanamycin. Specifically, the *E. coli* strains used in protein expression experiments were grown in 1 L of LB medium containing 50 μg/mL kanamycin at 37 °C with shaking at 220 rpm until the culture optical density at 600 nm ($OD_{600}$) reached 0.6; protein expression was induced by the addition of isopropyl-D-thiogalactopyranoside (IPTG) to a final concentration of 0.1 mM, and the culture was allowed to incubate for an additional 24 h at 16 °C. The cells were then harvested by centrifugation at 6000 × g for 20 min, flash-frozen and stored at −80 °C.

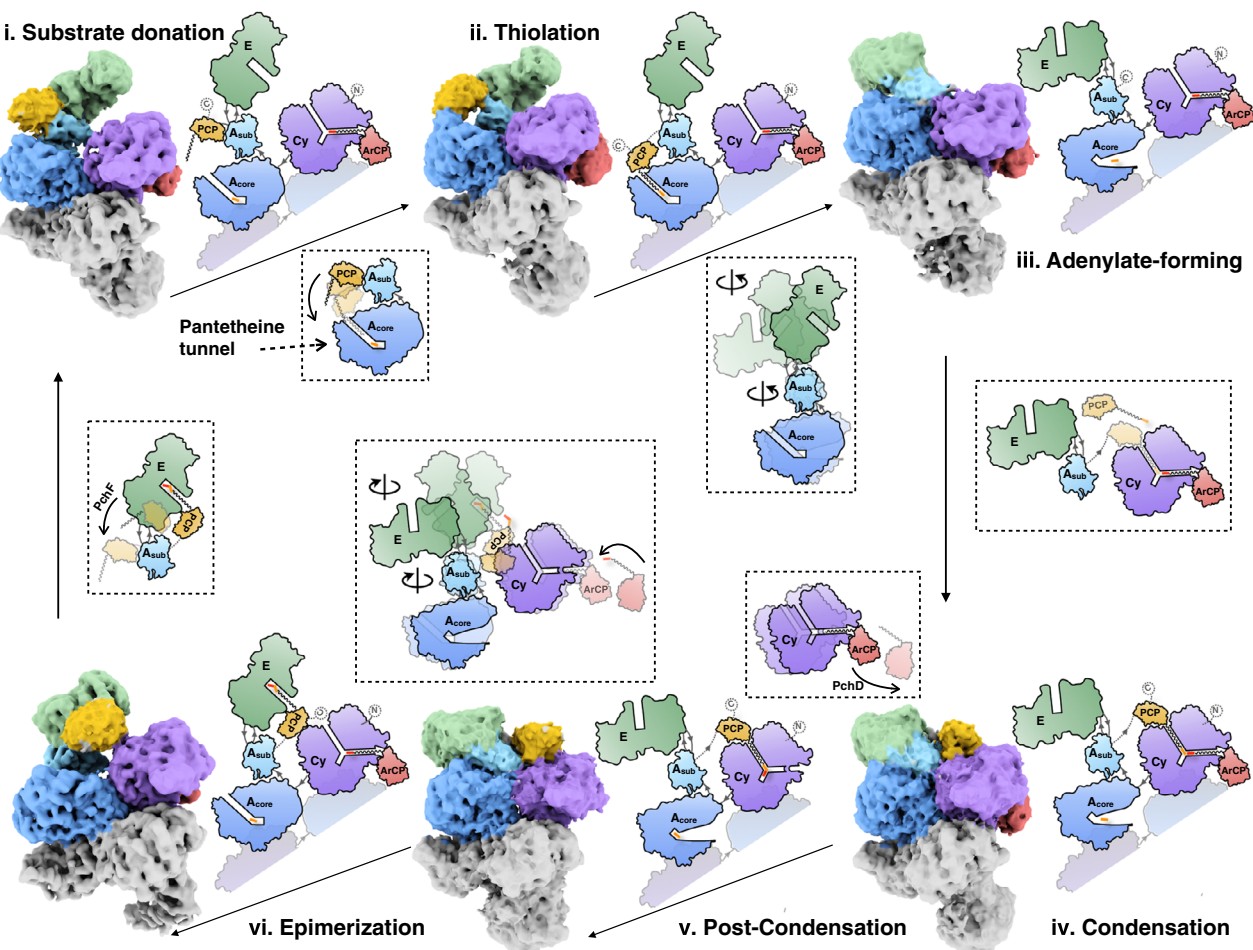

**Fig. 5 Proposed catalytic trajectory of PchE.** Six representative cryo-EM maps reconstructed by cryoDRGN (labeled from i to vi) are docked with atomic models of each domain (highlighted with one chain and colored as in Fig. 1), and these represent distinct catalytic steps of PchE. The proposed structures are shown as schematics next to each map, which depict the working model of substrate processing by PchE guided by large-scale motions of the ArCP, PCP, and A$_{sub}$-E-A$_{sub}$ domains in collaboration with fine-tuned switches of the substrate tunnels in the Cy and A domains. The catalytic direction is indicated with arrowed lines. The substrates cysteine and salicylate are represented as red and yellow bars, respectively. The proposed changes between each state are emphasized within dotted boxes (related to Supplementary Fig. 22). For more visualization, the PchE catalytic trajectory of a series of 120 cryo-EM maps are displayed in Supplementary Movie 2.

**Purification and sample preparations of His-tagged PchD and Strep II-tagged PchE.** Dimethylformamide (DMSO) and gravity columns were purchased from Sangon Biotech (Shanghai) Co., Ltd. TWEEN 20, *d*-Desthiobiotin and Millipore's Amicon® Ultra0.5 10k centrifugal filter devices were purchased from Sigma-Aldrich (Merck KGaA, Darmstadt, Germany). Ni-NTA resin and Superose 6 Increase 10/300 GL columns were purchased from GE Healthcare (GE Healthcare Life Sciences, Little Chalfont, UK). Strep-Tactin® Sepharose® resin was purchased from IBA (IBA Lifesciences, Göttingen, Germany). All experiments were performed at 4 ℃ unless indicated.

For the purification of PchE, 10 g of frozen cells was thawed, resuspended in 50 mL of buffer A (50 mM Tris·HCl pH 7.9, 150 mM NaCl, and 5% glycerol) and lysed using a French press (Union-Biotech, Shanghai, China) operated at 4 ℃ (600 bar). Cell debris was removed by centrifugation at 18,000 × g for 30 min, and the resulting supernatant was incubated with 2 mL of Strep-Tactin resin pre-equilibrated with 20 mL of buffer A and then gently rotated for 2 h using a QB-206 multipurpose shaker (Kylin-Bell, Haimen, China) for sufficient protein-resin interaction. After spinning at 800 g for 3 min, the supernatant was discarded, and the resin was transferred to a 12-mL gravity-flow column using buffer A. The column was then washed with 60 mL of buffer A and eluted with 10 mL of elution buffer (50 mM Tris·HCl pH 7.9, 150 mM NaCl, 5% glycerol, and 2.5 mM *d*-Desthiobiotin). The concentration of purified PchE was measured using the Bradford assay. Approximately 6 mg of PchE can routinely be obtained from 10 g of cell paste.

For the purification of PchD, 5 g of frozen cells was thawed, resuspended in 30 mL of buffer B (50 mM Tris·HCl pH 8.0, 150 mM NaCl, 5% glycerol, and 40 mM imidazole) and lysed using a French press (Union-Biotech, Shanghai, China) operated at 4 ℃ (600 bar). Cell debris was removed by centrifugation at 18,000 × g for 30 min, and the resulting supernatant was loaded onto a 12-mL

gravity-flow column packed with 2 mL of Ni-NTA resin pre-equilibrated with 20 mL of buffer B. The resin was then washed with 40 mL of buffer B. PchD was eluted using 5 mL of elution buffer (300 mM imidazole pH 8.0, 50 mM NaCl, and 5% glycerol) and incubated on ice until use in the PchDE enzymatic reaction. The concentration of purified PchD was measured using the Bradford assay. Approximately 3 mg of PchD can routinely be obtained from 5 g of cell paste.

Six milligrams of PchE was mixed with 1 mg of PchD in buffer containing 50 mM Tris pH 7.9, 10 mM MgCl₂, 1 mM salicylate, 5 mM cysteine, and 5 mM ATP and incubated for at least 1 h. The enzymatic reaction mixture was then concentrated to 1 mL using a Millipore Amicon® Ultra 10k centrifugal filter device according to the protocol provided by the company. The solution was centrifuged at 16,000 × g for 10 min to remove precipitates and then subjected to size-exclusion chromatography using a pre-equilibrated Superose 6 Increase 10/300 GL column in sizing buffer (50 mM Tricine pH 8.0, 1 mM MgCl₂) on an ÄKTA fast protein liquid chromatography system (GE Healthcare Life Sciences). The peak fractions were pooled and concentrated with the centrifugal filter device to a concentration of approximately 4 mg/mL (determined by the Bradford assay using BSA as a standard). The sample was then combined with cysteine at a final concentration of 1 mM and ATP for cryo-EM specimen preparation.

**In vitro reconstitution experiments.** PchD (10 μM) was incubated with 10 μM WT or mutants of PchE and 10 μM SrfD, 1 mM salicylate, 5 mM cysteine, and 5 mM ATP in buffer (50 mM Tris pH 7.9 and 10 mM MgCl₂) in a 300-μL volume at 25 ℃ for 2.5 h. The reactions were quenched with 100 μL of 1 M HCl, after which 900 μL ethyl acetate (EA) was added into the reaction mixtures to extract the salicylate-containing compounds. The mixture was allowed to settle for 10 min to permit phase separation, the upper ethyl acetate layer was collected and dried under vacuum in a rotary evaporator for 2 h. The residues were redissolved in

200 µL of 10% acetonitrile/water, after which the precipitate was removed by centrifugation at 15,000 × g for 10 min for the following HPLC and LC-MS analysis.

HPLC analysis was conducted with an Agilent C18 reverse-phase column (Pursuit XRs C18, 3 µm, 250 × 4.6 mm; Agilent) on an Agilent HPLC system (Agilent 1260 Infinity; Agilent) at a flow rate of 0.8 mL/min, a sample injection volume of 10 µL and mobile phase A (water/0.01% formic acid) and B (acetonitrile/ 0.01% formic acid). Mobile A was gradually replaced by 25% to 60% volumes of mobile B over a period of 5 to 30 min, 60% to 80% volumes for 30 to 32 min, 80% volumes for 32 to 37 min, and then 80% to 25% volumes for 37 to 38 min. The elution was monitored by UV spectroscopy at $\lambda$ 254 nm and the amount of product present in the reaction mixtures was calculated from the peak area by Agilent OpenLAB CDS ChemStation Edition software.

LC-MS was conducted with an Agilent 1290 Infinity Liquid chromatography and 6545 Quadrupole Time-of-Flight Mass Spectrometer using positive electrospray ionization and an Agilent C18 reverse-phase column (Pursuit XRs C18, 3 µm, 250 × 4.6 mm; Agilent) on Agilent Masshunter Workstation software. Samples were eluted at a flow rate of 0.4 mL/min with a sample injection volume of 20 µL and the same mobile phases as HPLC analysis. Mobile A was gradually replaced by 25% to 60% volumes of mobile B over a period of 10 to 60 min, 60% to 80% volumes for 60 to 64 min, 80% volumes for 64 to 74 min, and then 80% to 25% volumes for 74 to 76 min. Product masses was determined offline using ESI-MS in positive-ion mode with an ESI nebulizer and the mass spectrometer was set to acquire spectra in the mass range 50–1000 m/z. $[M + H]^+$ parent masses for both the linear (242.04) and heterocyclic (224.03) products were placed into an include list and product ions specific for each product were used to confirm identity for each peak. Extracted ion currents for each parent ion were extracted to provide relative concentrations for each species.

**Cryo-EM specimen preparation and data acquisition**. Four-microliter aliquots of specimens at ~4 mg/mL were applied to glow-discharged holey carbon grids (Quantifoil Cu, R1.2/1.3, 200 mesh) for 6 s of incubation, blotted for 2.5 s and plunge-frozen into liquid ethane precooled by liquid nitrogen using a Vitrobot Mark IV (FEI) operated at approximately 100% humidity and 4.5 °C. Cryo-EM images were collected with a Titan Krios electron microscope (FEI) operated at 300 kV and equipped with a K2 Summit direct electron detector (Gatan). Thirty-eight frames were recorded for each movie stack at a nominal magnification of 22500-fold in super-resolution mode with a pixel size of 1.0 Å within the defocus range of 1.5 to 2.5 µm. A total of 3749 movie stacks of PchE were automatically collected using SerialEM[29] with an exposure time of 7.6 s (0.2 s per frame) and a total dose of 60.8 e-/Å$^2$.

**Image processing**. All movies of the dataset were aligned and dose-weighted using MotionCor2[30]. The contrast transfer function (CTF) and defocus parameters were determined by Gctf[31]. Micrograph checking, particle autopicking, 2D and 3D classification, autorefinement, postprocessing and resolution estimation of each cryo-EM map were performed using RELION 3.0[32–34]. Approximately 2,000 particles of each dataset were manually picked and subjected to reference-free 2D classification. The best representative 2D classes were selected as templates for autopicking.

For the reconstruction of PchE, the datasets were cleaned by removing ice contaminants and junk particles after two rounds of 2D classification, and good classes were kept to generate the 30-Å 3D initial model, which was low-pass filtered to 70 Å as the reference for subsequent 3D classification (Supplementary Fig. 3). The best classes of 393,498 particles were selected for autorefinement, postprocessing, CTF refinement and Bayesian polishing, which resulted in the reconstruction of a 3.67-Å PchE consensus cryo-EM map. Three-dimensional classification with finer, local angular searches was further performed for conformational difference detection. For conformation 1, one class of 219,049 particles was autorefined and postprocessed, which resulted in the reconstruction of a 2.97-Å cryo-EM map. For conformation 2, two classes were combined with a soft mask on the rotated E-PCP domains for focused 3D classification, and the best class of 19,661 particles was selected for 3D autorefinement and postprocessing, which resulted in the reconstruction of a 3.78-Å cryo-EM map. For conformation 3, one class of 58,112 particles was autorefined and postprocessed, which resulted in the reconstruction of a 3.47-Å cryo-EM map.

For better integrity of the moving parts of PchE, focused 3D classifications of the particles of the three conformations were further performed for structural heterogeneity detection. For the E domain in conformation 1, two classes were combined with a soft mask on the A$_{sub}$-E-A$_{sub}$ domains for focused 3D classification, and the best two classes of 93,571 particles were selected for 3D masked autorefinement with signal subtraction, which resulted in the reconstruction of a 3.16-Å cryo-EM map. For the cooccurrence of ArCP and PCP with rotated A$_{sub}$-E-A$_{sub}$ domains in conformation 2, one class was selected with a soft mask around the ArCP-Cy-A$_{sub}$-E-A$_{sub}$-PCP domains for focused 3D classification, and one best class of 13,742 particles was selected for 3D autorefinement, which resulted in the reconstruction of a 3.90-Å cryo-EM map. For PCP and rotated A$_{sub}$-E-A$_{sub}$ domains in conformation 3, two classes were combined with a soft mask on the A$_{sub}$-E-A$_{sub}$-PCP domains for focused 3D

classification, and the best class of 11,849 particles was selected for 3D autorefinement, which resulted in the reconstruction of a 3.78-Å cryo-EM map.

The resolution of all cryo-EM maps was estimated based on the corrected gold standard Fourier shell correlation (FSC) at the 0.143 criterion (Supplementary Figs. 4 and 5).

**Model building and refinement**. First, the HHpred server was used for protein homology analysis using the HMM-HMM comparison method[64,65]. Multiple homologous crystal structures for each domain of PchE (Cy, A, E, ArCP, and PCP) were rigid-body fitted into the cryo-EM maps using UCSF-Chimera[66] for comparative model rebuilding using RosettaCM[67–69]. The resulting atomic coordinates were further manually adjusted and built using Coot[70] and ISOLDE[71]. Structure refinement was performed using Phenix in real space with secondary structure and geometry restraints to prevent overfitting[72]. Subsequently, MolProbity[73] was used for model validation. The statistics are summarized in Supplementary Table 1. All cryo-EM densities and atomic models were visualized, and the figures depicting them were prepared using PyMOL, UCSF-Chimera and ChimeraX[74].

**Analysis of PchE dynamics by deep neural networks**. The dynamics of PchE were analyzed by deep neural networks using cryoDRGN Version 0.3.0 and following the software protocols[59].

PchE cryo-EM dataset high-resolution training: An 8-dimensional latent variable model was trained for 50 epochs using a total of 393,498 particles with a full-resolution image size of 240 × 240 (1.1 Å per pixel). The image poses and CTF parameters were extracted from the 3.67-Å consensus map, which was reconstructed in RELION (Supplementary Fig. 3). The encoder and decoder architectures were 1024 × 3 (nodes per layer x layer).

Generation of six representative cryo-EM maps: After training, we used k-means clustering to partition the latent space into k regions, and k = 400 was used with the latent encodings for our PchE cryo-EM dataset. The latent encoding closest in Euclidean distance to the k-means cluster center was defined as the 'on-data' cluster center, and volumes (cryo-EM maps) were generated at the 'on-data' cluster centers using the decoder network. Six structurally distinct representative structures were manually selected for visualization in Supplementary Fig. 22.

PchE molecular trajectory generation: For the graph traversal of the PchE in Supplementary Movie 2, the cryoDRGN software creates a nearest-neighbors graph from the latent encodings of the PchE particle images, in which a neighbor was defined if the Euclidean distance was below a threshold computed from the statistics of all pairwise distances. For the PchE cryo-EM dataset, we selected a value of 2.5 for the average number of neighbors across all nodes. We then used the latent encodings of the six representative maps shown in Supplementary Fig. 22 as the anchor points. Dijkstra's algorithm was used to find the shortest path along the graph connecting these six anchor points, and a total of 120 cryo-EM maps were generated along the latent space data manifold at all the visited nodes. Subsequently, the series of structures were visualized, and the movie depicting the PchE dynamics was prepared using ChimeraX.

**Substrate tunnel generation**. The substrate tunnels within each domain of PchE were calculated with the program Hollow[75] and adjusted using PyMOL (The PyMOL Molecular Graphics System, Version 2.0 Schrödinger, LLC.).

**Reporting summary**. Further information on the research design is available in the Nature Research Reporting Summary linked to this paper.

## Data availability

The 3D cryo-EM maps generated in this study have been deposited in the Electron Microscopy Data Bank [https://www.emdataresource.org/] under the accession numbers EMD-31198, EMD-31201, EMD-31199, EMD-31202, EMD-31200, and EMD-31203 (Supplementary Table 1). The atomic coordinates have been deposited in the Protein Data Bank [https://www.rcsb.org] under the accession numbers 7EMY, 7EN1, and 7EN2. The movies are provided as Supplementary Movies 1 and 2. Other data are available from the corresponding authors upon request. Source data are provided with this paper, as a Source Data file. Source data are provided with this paper.

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

## Acknowledgements

We thank F. F. Wang, G. Y. Li, L. H. Xin, J. L. Duan, and N. Liu for their help with the sample preparation and data collection. Cryo-EM images were collected at the National Facility for Protein Science in Shanghai (NFPS), Zhangjiang Lab. The computations in this paper were run on the π 2.0 cluster supported by the Center for High Performance Computing at Shanghai Jiao Tong University. We also thank L. L. You and Ellen D. Zhong for the help with the cryo-EM and cryoDRGN calculations, respectively. This work was financially supported by the National Key R&D Program of China (2018YFA0900700, 2019YFA0905400, 2021YFA0910500), the National Science Foundation of China (32171252, 91753123, 31470830), and Shanghai 'Super Postdoctoral' Incentive Program (202112). This research was funded in part by the Wellcome Trust [Grant number 209407/Z/17/Z]. For the purpose of open access, the author has applied a CC BY public copyright license to any Author Accepted Manuscript version arising from this submission.

## Author contributions

Z.W., J.W. and J.L. conceived the study. J.W., J.L., L.C., W.Z., Z.W., D.L., W.C. and L.K. performed the experiments. Z.W., J.W. and J.L. analyzed the data. T.C. participated in the model building and refinement work. All authors wrote the paper. Z.W., J.L. and Z.D. supervised this project.

## Competing interests

The authors declare no competing interests.
