## [Peer Review File · Nature Communications]

Catalytic trajectory of a dimeric nonribosomal peptide synthetase subunit with an inserted epimerase domainREVIEWER COMMENTS

Reviewer #1 (Remarks to the Author):

In “Catalytic trajectory of a dimeric nonribosomal peptide synthetase subunit with an inserted epimerase domain”, Wang and coworkers collect and analyze cryo-EM data on PchE, an NRPS subunit with domains ArCP-Cy-A-E-PCP. The EM, gel filtration and native gel show PchE to be dimeric. Three reconstructions, at resolutions of 3.0 Å, 3.8 Å and 3.5 Å are described, and each appears to be in a biologically relevant state. The authors perform mutagenesis on Cy and E domains to evaluate contributions of individual residues. Then, the authors perform cryoDRGN analyses to find additional conformations in the dataset, as well as generating 120 maps for domain docking.

The structures are interesting, mainly for the overall architecture, and because this is the first module to be determined which contains an E domain and the second with similar dimeric architecture.

The EM experiments appear well performed, with some additions / edits required:

- The cryoEM maps for the ligands (acyl-pPants, adenylates) are not shown anywhere and absolutely must be. It is quite odd that the authors seemingly observe the two substrates at the Cy domain pre-catalysis, so the evidence establishing proper modelling of substrates must be compelling.
- There are no details in the methods section about the cryoDRGN analyses. It is not described how training was performed, 8 dimensions were used, nor how the points were selected to make Video 3
- It is not appropriate to claim “near-continuous structural dynamics of the full catalytic cycle, from thiolation to epimerization.” Video 3 clearly shows most of the transition being a disappearance of a domain in one confirmation followed by its reappearance elsewhere.

For the biochemical assay, the authors do not show any of the data, just bar graphs of the integrated measurements. Mass spectra, EICs and UV traces must be shown to be able to evaluate data. (I believe Nature Publishing group now requires this, as well.) Also, in this data, are the authors summing signal only cyclized hydroxyphenylthiazoline, or also the uncyclized hydroxyphenylcysteine? That would be important to know for proper interpretation of Cy domain mutations, in particular. Furthermore, the assay design does not seem optimal. The assay performed appears to include no enzymatic nor chemical release step. It seems to rely on spontaneous hydrolysis of hydroxyphenylthiazoline from the PCP domain which is not very rapid and thus is likely the rate-limiting step, at least for the wildtype PchE. This very likely makes the data (which has not been shown) much weaker than what could be obtained in a true multiple turnover assay, and makes the mutational effects more indirectly related to the observed decreases in products.

Also importantly, the manuscript requires substantial rewriting before it is acceptable for publication anywhere. First, the authors must read Katsuyama et al. *Angew Chem Int Ed Engl.* 2021, which was published online in March 2021, and which includes crystallography and cryoEM of FmoA3, and adjust their manuscript accordingly. FmoA3 is an NRPS subunit with domains Cy-A-PCP which also forms a head-to-tail dimer. The authors must include with a structural comparison between PchE and FmoA3. Then, the manuscript must be rewritten in a more scholarly fashion. There are almost no references called in the results or discussion, and essentially no structural comparisons with known structures. This will both lead non-field expert readers to assume many well-established observations are novel to this study and prevent a field expert from understanding these data in the context of the existing body of published work. For example, the features discussed in the text about the Cy and A appear to be already well known (including their binding to CP domains), though it is difficult to tell because there is almost direct comparison to published structures. Conversely, from the E domain might to be a novel type of E domain, unlike the NRPS E domain that has been solved previously, but it is difficult to tell from the text and the authors only mention that it is noncanonical. (The methods state an E domain homolog was used for modelling, so perhaps it is not novel.) In any case, the text descriptions of the domains of which tens of homologues are known (A, Cy domain) and of which this is perhaps the first (E domain) are so similar in nature that the reader would not know what is new and what is not.

It is not particularly easy to properly place a novel structural study in the context of a large body of

existing literature, but including one sentence in the introduction with 11 references to say that other structures exist and then largely ignoring published work for the rest of the manuscript is not sufficient.

Specific examples of the issues raised above and other issues to address:

Line 21-23: "A typical synthetic elongation accomplished in a module involves thiolation, peptidyl carrier protein (PCP) donation, condensation and post-condensation steps":

This is not a good description of a typical elongation cycle, because:

- No mention of adenylation
- Despite the PCP domain being also called the thiolation domain, "thiolation" does not describe well the transfer of the aminoacyl moiety onto the thioester of the ppant of the PCP domain
- "PCP donation" seems to imply the PCP domain is the moiety being donated
- donation by the PCP domain is the same as condensation. The upstream PCP donates its attached acyl group in the condensation reaction
- Many (most?) elongation cycles do not have post-condensation steps.

Line 31-32: "The multilevel structural transitions guided by a large-scale rotation of the adenylation domain, together with the inserted epimerase":

The authors show that it is a large-scale rotation of the Asub domain, not the main A domain, but this sentence makes it seem like the whole A domain (or main portion of the A domain). This is important because the Asub rotation has been seen many times, originally by Gulick and coworkers, but a rotation of the main A domain (relative to the Cy domain) would be novel and unexpected. That the Asub rotates is made clear in the main body, and is easy to fix in the Abstract by just changing to "large-scale rotation of the adenylation C-terminal subdomain".

Line 44: Inappropriate sentence structure: "select, incorporate, and modify amino acids into the growing peptide"

Line 45: "with more than 500 precursors." Add a modifying phrase or it implies NRPSs all use more than 500 substrates.

ED Fig 1

-As the Asub is drawn each time in light blue in this figure, it should be also drawn that way in the second module of Vlm1.

Lines 79-82, NRPS dimers. The authors mention VibF, but not SfmC as biochemically identified as dimers. The recent FmoA3 structure, solved by both EM and crystallography is a Cy-A-T domain head-to-tail dimer, and StsA crystals also showed a dimeric arrangement. Therefore, this statement is incorrect: "all available crystal structures support the widely accepted idea is that NRPSs are monomeric".

Moreover, lines 79-82 implies that the field was unaware that some NRPSs are dimers and that this study will reveal that NRPSs are dimers. The overwhelming majority of NRPSs are monomers, shown not only by crystal structures but also by solution techniques such as gel filtration experiments.

ED Fig 6. The authors are showing maps "carved" close to their model. This is fine, but can be used to artificially make the maps look better than they use a low carving distance. The carving distance should not be less than 2.5-3 Å, and must be stated in the figure legend.

Line 102: Edit the phrase: "the Asub domain is inserted by the E domain"

Lines 106 – 108. The terminology being used is unclear.

In the condensation reaction, ArCP donates its acyl moiety to Cys-ppant-PCP. Therefore, the ArCP donating state and the condensing state should be the same, at least with respect to the ArCP. It would be much clearer to say in state 1 ArCP is bound to Cy, in state 2, ArCP and PCP are bound to Cy, and in state 3, one protomer has ArCP bound to Cy and the other has PCP bound to Cy.

Likewise, in Fig 2b, why it is called “Condensation/Cyclization” state? What evidence exists that both CPs are required to be bound for cyclization? On the contrary, the existence of NRPSs with Cy domains dedicated for either condensation or cyclization suggests that the donor CP is not required for cyclization.

Line 110: “two chains forming separate catalytic chambers” – This is borrowing terminology from FAS/PKS field, which would be fine, but those systems have largely enclosed catalytic chambers, which does not appear to be the case here.

Line 111-121: This description of head-to-tail dimerization sounds very much like the recently-described FmoA3 structure. A full comparison, including text passage and figure is necessary.

Lines 123-139: There are decades of structural work on C-superfamily domains as well as a handful of structures that observe CP:C or C:CP binding. It is unscholarly not to reference them here. For example, without a reference and acknowledging that this was expected, a statement like “The PchE Cy domain is a “V”-shaped pseudodimer that has two lobes, with each adopting the chloramphenicol acetyltransferase fold” implies this as a present discovery.

Lines 134-153: Likewise, please cite references 23 and 24 in this sentence: “The catalytic residue D480 aligns well with D1226 and D449 in the BmdB (r.m.s.d. 1.80 Å) and EpoB (r.m.s.d. 2.27 Å) Cy homologs, respectively”? A comparison between the mutagenesis analyses here and in those papers, or at the very least a statement about the mechanism of the Cy domain based on the conclusions of those papers is necessary to understand the mutational analysis here.

Lines 157-162: By “loop lid”, do the authors mean the element which has been called “the latch” by many papers? If so, the statement “Consequently, these open conformation-adopting two lids relieve a strong steric clash that would otherwise exist upon ArCP and PCP joining with Cy and subsequent pPant substrate entry” requires further explanation. There are many C domains that have the latch closed and still show a conformation that have an open tunnel that would accommodate pPant binding. Why would latch opening be required here? In general, the authors seem to only make structural comparisons of their Cy domain to the two excised Cy domain structures (and not the recent Cy-A-PCP structure), but ignore the large body of structural information from the related C domain.

Lines 163-171: Was the blockage of the donor side of the tunnel discussed in BmdB or EpoB papers? This is presented as novel and not referenced. Furthermore, a ~20% reduction in activity is not compelling for an important gatekeeping residue. Why isn't this just as easily explained by the mutation causing partial loss of interaction between Cy and ArCP domains?

Lines 172-185: There are ~60 structures of A domains available, including with aminoacyl adenylates captured by crystallography. Is the brief comparison with PheA the most relevant? Is this the first Cys-specific A domain to be determined?

Lines 178-182: “The predicted specificity-conferring code for L-cysteine recognition is the DLFNLSLIWK 10 AA signature sequence.... The inaccuracy of prediction by the structure-projection method probably results...” The 10 AA signature sequence is meant to allow prediction of substrate specificity, and it does. Submitting the PchE sequence into “PKS/NRPS Analysis Web-site” or “NRPS Predictor2” for example accurately predicts cysteine as the substrate. The code is not necessarily meant to list the residues that contact the substrate amino acid. For example, A domains specific for glycine will have a 10 AA signature sequence as well, but 10 residues won't contact the substrate glycine.

Lines 186-195: It is impossible for the reader to understand whether this movement is the same of different to previously-observed movements. And because there is no reference to the statement “It is known that the Asub domain rotates relative to the Acore domain”, the authors don't direct the readers to the papers they would need to read / structures they would need to download to figure that out.

Also in Video 2, it's somewhat confusing that the open confirmation for substrate binding (called the "open" confirmation here and in the literature) is labelled as "closed" for PCP blocking.

Lines 196-200: How does the E domain insertion compare structurally to Ref 22? Are similar non-covalent contacts made between E and A_{sub} as are made between MT and A_{sub}? Is the insertion point precisely the same? ("Between A8 and A9 motifs" does not precisely define the insertion point.)

Lines 197 and 206: The E domain is referred to as a "stuffed E domain". This follows one group's terminology from work with MT domains, but it is not used by any other group in the field and is jargony. Please use a more formal term than "stuffed", perhaps that the E domain is "embedded" or the A_{sub} is "interrupted" by the E domain.

Lines 201-214: Canonical E domains are C domain superfamily proteins. This E domain appears to be of different evolutionary origin. The authors simply mention that it is "a noncanonical epimerase". What is the E domain evolutionarily related to? Is this a novel domain structure?

Line 211: "E1234 could donate a proton": A mention of the typical protonation state and pK_a of glutamate residues, and how that might be altered here is warranted.

Line 230: "near-continuous conformations", Video 3 and Line 33-35 "Moreover, using deep neural network calculations, we visualized the near-continuous structural dynamics of the full catalytic cycle, from thiolation to epimerization."

The cryoDRGN work is nice, but the Video 3 clearly shows most of the transition being a disappearance of a domain at one position followed by its reappearance elsewhere. This should not be described as near-continuous.

Lines 247-254. Must be modified in the re-write after authors read Katsuyama et al. *Angew Chem Int Ed Engl.* 2021, which was published online in March 2021.

Figure 2, 3, ED Fig 6.

-Maps absolutely have to be shown for all the ligands. Only models and distances (to the tenth on an Å) are shown, and the reviewers and readers must be able to judge the quality of the data that leads to this model. State 2 is modelled as pre-condensation and state 3 is modelled as post-cyclization without showing the evidence.

Figure 5 and elsewhere in the text: Cryo-EM maps are Coulomb potential maps, not density maps.

ED Fig 10 and Video 3: The point cloud representation of the cryoDRGN processing saturates, so the ability to see how many particles are where is lost. This representation is far more useful if you can see localised accumulations of species within the cloud.

References: There are several formatting issues (e.g. capitalization, journal names) in the references.

Reviewer #2 (Remarks to the Author):

Nonribosomal peptide synthetase (NRPS) proteins are modular assembly line proteins that produce important peptide natural products. The NRPS enzymes use a multidomain architecture and a fascinating biosynthetic strategy in which the nascent peptide is tethered to an integrated carrier protein domain and transported to catalytic domains for synthesis. Recent structure advances have provided clearer views of multiple states of the NRPS modules in different stages of the active cycle. Additional structures of multi-domain NRPS proteins, as well as insights into the molecular trajectories between individual states, are important to understand completely the nature of the NRPS structural and catalytic cycle.

Here, Wang and colleagues describe the cryo-Electron Microscopy structures of PchE, an NRPS protein from the *Pseudomonas pyochelin* biosynthesis pathway. This protein contains several interesting features. It is a five domain protein with a ArCP-Cyclization-Adenylation-Epimerization-PCP architecture. The Adenylation domain contains two subdomains (N- and C-terminal) and the epimerization domains is inserted into the highly dynamic C-terminal subdomain. The observation of multiple states of the PchE protein provides a better understanding of the interactions of the different domains and insights into the known conformational changes of the NRPS adenylation domains that foster the progression of the structural cycle. The manuscript is largely well-written and will provide an important advance for the field of NRPS structural biology.

There are places, however, that fail to fully explain the techniques used and that fail to distinguish between experimental observations and models. The manuscript needs to better elaborate the structures and more completely describe the cryoDRGN analysis. A clearer description of the literature on the NRPS proteins will also provide the necessary context for the presented work.

1. An important observation of this structure is that the PchE adopts a head-to-tail dimeric structure, where the dimeric interface is formed by the central Cyclization-Adenylation domain core. While the H-shaped cryoEM reconstructions are clear, the authors proceed directly into the analysis of this dimeric structure without properly putting this in context. The size exclusion column (line 88 and ED Figure 2) is presented without mw standards. I recommend a new section to begin the Results portion of the manuscript that is titled "PchE forms a dimeric quaternary structure". The experimental results here establish that the protein exists as a dimer in the size exclusion experiment. The cryoEM results in a subsequent section present the nature of this dimeric protein.

2. It is then necessary to provide a complete description of the overall architecture of the three conformations that were reconstructed. Relative to the Cyclization/Adenylation core, there are multiple highly dynamic protein domains: the ArCP, the PCP, the C-terminal subdomain of the adenylation domain and the epimerization domain. Further, there are two chains of the dimer. While this section currently gives an overview of the nature of the carrier proteins relative to the Cyclization domain, it fails to describe the overall architecture and conformational changes of the other important domains/subdomains. Perhaps the authors would consider labeling the two chains of each Dimer in Figure 1 as chains A and B to establish a basis so that the discussion of different states can be referred to as "Conformation 1, Chain A" for example. Similarly, other than the more dramatic difference between the two chains of Conformation 3, it would be helpful to have some sense (an rms comparison, perhaps) of the difference observed in the two chains for each conformation.

3. Equally important, this section should clearly establish the ligands that are present in the two chains of the three conformations. It is important to state what complexes are experimentally observed to differentiate later discussions of active site contents that may result from modeling of ligands or comparisons with prior structures.

4. The lack of clear distinction of the different structures and structural states is notable in Figure 3 and the associated discussion. There is a well-established rotation of the C-terminal subdomain of the adenylation domain between the adenylylate-forming and thioester-forming conformations. In none of the PchE conformations is the PCP bound to the adenylation domain. Comparison of the observed 155° subdomain rotation in Figure 3 to known adenylation domains states should be included. Are the observed conformations new or does PchE recapitulate the known states seen in prior structures that may now also serve to facilitate transport to the adenylation and epimerization domains? Finally, do the Epimerization and small C-terminal subdomain rotate as a rigid body, making significant domain contacts, or rotate independently?

5. Please state which structures were used for the different selections of electron density maps in ED Figure 6. Were maps of similar quality for all three final structures? While the electron density looks ok, it appears that the maps are presented with a "carve" radius that fails to depict density more than a set distance from the displayed atoms. This can be deceptive and the authors should present several regions with maps that allow the reader to better assess the maps. Electron density that

supports ligand placement is also important, as well, to further expand on Point 3 above.

6. The use of cryoDRGN to identify the overall trajectories is presented without details about how the program was run. Particularly as this is a new technique, sufficient experimental details should be provided in the Methods to understand what was done. This software package is not listed in the “software and code” section of the Reporting Summary document. Additionally, the “catalytic trajectory of PchE” section should include a brief introduction to the method being used and the nature and degree of confidence of the output trajectories. In particular, it is not clear how much experimental data supports the calculations used by this method or is it a model of realistic trajectories between experimental states? Finally, is there a validation score associated with this method that supports the different states? Is there greater confidence in some compared to others?

Other minor concerns and corrections.

Line 54. PchE does not synthesize DHA alone, as it requires loading by PchD. Additionally, ED Figure 1 is overly simplified and does not accurately depict the role of PchG. A naïve reader could take the schematic as demonstrating the PchG catalyzes release of the product, rather than formation of the thiazolidine containing peptide while still bound to PchF. The legends or figure of ED Figure 1 should also explain that the Adenylation C-terminal subdomain is colored lighter blue.

Lines 65-70. The description of existing prior structures fails to provide context for the current study. A more complete discussion of structures in multiple catalytic states will provide suitable background for understanding the advances of the current structures.

Line 78. The structural basis of specificity of many NRPS adenylation domains is known. In particular, the original Stachelhaus code continues to provide a foundation for comparisons between key residues and the substrate binding. Is there something unique about Cysteine recognition from PchE? Does the conventional code predict an alternate substrate? Similarly, on Line 179 a predicted code is provided without citation nor an explanation of the diversity of Cys activating adenylation domains.

Line 81. Related to point 1 above, the authors need a transition that establishes that the current paper describes an unusual dimeric NRPS. The previous sentence states that NRPSs are monomers.

Line 86. Please include that PchD was added to load the salicylate on the ArCP domain of PchE. As written, the paper suggests that the chromatogram of ED Figure 2 should contain PchD at a 0.5:1.0 ratio with PchE. There is a very small trailing peak but it does not appear that Figure matches what is described in the paper.

Line 102. Perhaps replace “the A sub domain is inserted by the E domain” with “the E domain is inserted into the A sub domain” or “the A sub domain is interrupted by the E domain”.

Line 124. Replace “The ArCP/PCP contacts the Cy domain” with “The ArCP and the PCP contact the Cy domain on ...”

Line 148. If Asp480 is abstracting a proton, it would be acting as a base.

Figure 2. Again, please confirm that the ligands shown bound to PPant arms are present in experimental models and not merely modeled.

Line 137, as well as 158-160. Homologs EpoB and BmdB are included in the discussion without actually citing references to these papers. The Dowling and Bloudoff papers are previously cited (line 67) among “several individual excised NRPS domains”.

Lines 172 and 173. The dramatic conformational rotation of the small C-terminal subdomain is presented without discussion of precedent from prior structural studies from our group and others.

Similarly, lines 186-187 discuss the hinge without prior discussion or references.

Line 189. It is not clear what is meant by “the substrate L-cysteine adenylation function”.

Line 198. Reference for A8 and A9 motifs is necessary.

Line 203. The active sites of Epimerization domains is well known from prior structural and functional work, as well as analysis homologous condensation domains and “might” is unnecessary here.

Line 217. As noted, methods need to be provided for the cryoDRGN technique.

Line 233. The prior literature on adenylation domains identifies an “open” conformation in which the C-terminal subdomain adopts a non-catalytic conformation. This conformation is presumed to be used for substrate binding and product release. The two catalytic conformations (adenylate-forming and thioester-forming) have both been described as closed, although the literature is not always consistent. Further, while the adenylation domains have a well-established pantetheine tunnel, it is not clear what is meant by “adenylation tunnel”. Is this a new structural feature that has not been observed previously? Is it simply the cavity formed in any of prior “open” conformations used for substrate binding.

Lines 241-242. Please also clarify what is meant by “managed” in the discussion of functional interplay between the adenylation C-terminal subdomain and the epimerization domain.

Line 250. The core interaction between the adenylation and condensation (cyclization) domains is firmly established. Please cite prior studies and note whether the stable Cy/Aden core of PchE is identical, similar, or different from prior structures.

Line 251-252. It is not clear what is meant by “... a potential platform for interactions with upstream PchD and downstream PchF modules”. Is there evidence that the core Cy/Aden could form a protein-protein interface with PchF? Wouldn't it be expected that the dynamic PCP would interact with PchF N-terminal Cyclization domain?

Line 261. Again, a better description of the cryoDGRN method could clarify whether the proper word here is “detected” or “modeled”. Additionally, the conformational changes should be highlighted in the context of details of the structural cycle. In the transition from the I. Waiting to the II. Thiolation state, does the adenylation domain adopt the standard conformational change? Does the adenylation domain adopt the adenylate-forming conformation, with catalytic A10 motif lysine present in the active site, in this “waiting” state?

Andrew Gulick,
Buffalo, USA

Reviewer #3 (Remarks to the Author):

The manuscript reports three cryo-EM reconstructions of the PchE multi-domain protein from the biosynthetic pathway for pyochelin, a siderophore from the human pathogen *Pseudomonas aeruginosa*. Pyochelin is made by a nonribosomal peptide synthetase (NRPS) biosynthetic assembly line. The PchE protein has 3 enzyme domains and 2 carrier domains (ArCP-Cy-A-E-PCP) that carry out a total of 5 catalytic steps: the adenylation (A) domain (1) forms Cys-adenylate and (2) transfers Cys to the peptidyl carrier protein (PCP) domain, the cyclization domain (Cy) (3) condenses Cys with salicylate (tethered to an aryl carrier protein (ArCP) domain) and (4) cyclizes the product to a thiazoline, and an epimerase (E) domain (5) inverts the stereo center at the Cys C-alpha. The study is

interesting because it adds a new molecular system to a small database of NRPS module structures. Large conformational changes are hallmarks of the catalytic cycle and this work illustrates an aspect of PchE flexibility.

The cryoEM reconstructions were computed from images taken from a single sample type: PchE in the presence of all its substrates (ATP, Cys and pre-loaded salicylate). The authors separated 3 PchE biochemical states by analysis of auto-picked particles. This approach of purification-by-computer greatly complicated the task of sorting particles into classes, especially as PchE is a dimer of independently functioning monomers that cannot be assumed to be in the same state. More definitive results could have been obtained by sequential addition of substrates and imaging of distinct and defined biochemical states. Moreover, it is not clear from the manuscript exactly which of the experimentally observed biochemical states correspond to which of the proposed PchE steps in the model of the catalytic cycle (Fig. 5). For example, the PCP domain was not visualized at either the A domain or the E domain (Fig. 1), so some of the key states shown in Fig. 5 are speculative.

Others have reported NRPS module structures in multiple states representing large conformational changes. The key papers are cited in the introduction, but this paper does not connect the PchE structures to these systems, which also display large motions of the A sub-domain.

The paper is not clearly written. While a few English problems contribute to the lack of clarity, by far the larger problem is the incomplete description that makes it very difficult to connect that experimentally determined structures to the model. In Fig. 2, the cage and solid renderings appear to be of the protein surface, not the EM map - this should be stated in the caption.

The cryo-EM structures appear to be of good quality. The atomic model is a good fit to the map of highest resolution (2.97 Å). However, some details are not presented clearly. For example, in Fig. S6 with detailed densities, it is not stated which map is displayed, nor what is the contour level. From my inspection of the model-to-map fit, I know that a contour level where side chains are clearly visible would have no density for the aryl group on the ArCP. Thus I suspect that the images in this figure have different density contour levels.

PchE is a dimer by the EM analysis, but the apparent MW in solution is not stated, despite the lovely gel filtration profile in Fig. S2c. Dimer formation is unusual for an NRPS module, so it should probably be labeled simply as a rare occurrence and not a major finding.

Point-by-point response to the reviewers

*Reviewer Comments: Black, Helvetica, 10

*Author Responses: Blue, Helvetica, 10

Reviewer #1 (Remarks to the Author):

In “Catalytic trajectory of a dimeric nonribosomal peptide synthetase subunit with an inserted epimerase domain”, Wang and coworkers collect and analyze cryo-EM data on PchE, an NRPS subunit with domains ArCP-Cy-A-E-PCP. The EM, gel filtration and native gel show PchE to be dimeric. Three reconstructions, at resolutions of 3.0 Å, 3.8 Å and 3.5 Å are described, and each appears to be in a biologically relevant state. The authors perform mutagenesis on Cy and E domains to evaluate contributions of individual residues. Then, the authors perform cryoDRGN analyses to find additional conformations in the dataset, as well as generating 120 maps for domain docking. The structures are interesting, mainly for the overall architecture, and because this is the first module to be determined which contains an E domain and the second with similar dimeric architecture.

We are grateful for the reviewer’s thorough and critical comments on our work. These comments are extremely professional and very helpful to the quality promotion of this manuscript. We have carefully addressed these comments point by point, added new structural comparison figures, and rewritten/reorganized the manuscript thoroughly for more clarity at our best. New biochemical turnover experiments have been performed for better supporting our conclusion, as the reviewer suggested. The manuscript has also been sent for language polishing by Springer Nature Company. Please see Supplementary_Information_Springer_nature.

The EM experiments appear well performed, with some additions / edits required:

-The cryoEM maps for the ligands (acyl-pPants, adenylates) are not shown anywhere and absolutely must be. It is quite odd that the authors seemingly observe the two substrates at the Cy domain pre-catalysis, so the evidence establishing proper modelling of substrates must be compelling.

A: Thanks. The cryo-EM maps for the ligands have been shown in Supplementary Fig. 6. The two substrates at the Cy domain shown in Fig. 2b (Condensation) were not experimentally observed. They were modelled and bordered with dashed lines (Fig. 2 and annotated in its legend), in order to distinguish them from the experimentally observed ligands (bordered with solid lines in Fig. 2).

Supplementary Fig. 6 Representative cryo-EM maps of the PchE ligands. The cryo-EM maps of the ligands observed in this study, shown as meshes, are displayed at different contour levels and generated in ChimeraX. The colors and labels of distinct conformations and chains are corresponding to Fig. 1. **a** Map of the Ser46 salicyl-pPant arm of the ArCP domain in conformation 1_chain A, contoured at 0.007 (3.0σ). This ligand was also observed similarly in conformation 3_chain B (not shown). **b** Map of the Ser1385 Hydroxyphenylthiazoliny-pPant arm of the PCP domain in conformation 3_chain A, contoured at 0.006 (2.8σ). **c** Map of the Ser46 pPant arm of the ArCP domain in conformation 2_chain A, contoured at 0.010 (4.7σ). **d** Map of the Ser1385 pPant arm of the PCP domain in conformation 2_chain B, contoured at 0.006 (2.8σ). **e** Maps were contoured at 0.012 (Left, 5.2σ ; middle, 5.5σ ; right, 5.7σ). Left, map of the Cysteiny-AMP and Mg^{2+} within A domain in conformation 1_chain A (similar in chain B and conformation 3_chain A, not shown); middle, map of the Cysteine, AMP and Mg^{2+} within A domain in conformation 3_chain B; right, map of the AMP within A domain in conformation 2_chain A (similar in chain B, not shown).

-There are no details in the methods section about the cryoDRGN analyses. It is not described how training was performed, 8 dimensions were used, nor how the points were selected to make Video 3

A: We have now added the full description for the cryoDRGN analyses in the Methods section (Line 775).

Line 775: **“Analysis of PchE dynamics by deep neural networks**

The dynamics of PchE were analyzed by deep neural networks using cryoDRGN Version 0.3.0 and following the software protocols⁵⁹.

PchE cryo-EM dataset high-resolution training: An 8-dimensional latent variable model was trained for 50 epochs using a total of 393,498 particles with a full-resolution image size of 240 x 240 (1.1 Å per pixel). The image poses and CTF parameters were extracted from the 3.67-Å consensus map, which was reconstructed in RELION (Supplementary Fig. 3). The encoder and decoder architectures were 1024 x 3 (nodes per layer x layer).

Generation of six representative cryo-EM maps: After training, we used *k*-means clustering to partition the latent space into *k* regions, and *k*=400 was used with the latent encodings for our PchE cryo-EM dataset. The latent encoding closest in Euclidean distance to the *k*-means cluster center was defined as the ‘on-data’ cluster center, and volumes (cryo-EM maps) were generated at the ‘on-data’ cluster centers using the decoder network. Six structurally distinct representative structures were manually selected for visualization in Supplementary Fig. 22.

PchE molecular trajectory generation: For the graph traversal of the PchE in Supplementary Movie 2, the cryoDRGN software creates a nearest-neighbors graph from the latent encodings of the PchE particle images, in which a neighbor was defined if the Euclidean distance was below a threshold computed from the statistics of all pairwise distances. For the PchE cryo-EM dataset, we selected a value of 2.5 for the average number of neighbors across all nodes. We then used the latent encodings of the six representative maps shown in Supplementary Fig. 22 as the anchor points. Dijkstra’s algorithm was used to find the shortest path along the graph connecting these six anchor points, and a total of 120 cryo-EM maps were generated along the latent space data manifold at all the visited nodes. Subsequently, the series of structures were visualized, and the movie depicting the PchE dynamics was prepared using ChimeraX.”

59. Zhong, E. D., Bepler, T., Berger, B. & Davis, J. H. CryoDRGN: reconstruction of heterogeneous cryo-EM structures using neural networks. *Nat. Methods* **18**, 176–185 (2021).

-It is not appropriate to claim “near-continuous structural dynamics of the full catalytic cycle, from thiolation to epimerization.” Video 3 clearly shows most of the transition being a disappearance of a domain in one confirmation followed by its reappearance elsewhere.

A: We have revised the sentence (Line 34) and throughout the text: “Moreover, using deep neural network calculations, we visualized the direct structural dynamics of the full catalytic cycle from thiolation to epimerization.”

For the biochemical assay, the authors do not show any of the data, just bar graphs of the integrated measurements. Mass spectra, EICs and UV traces much be shown to be able to evaluate data. (I believe Nature Publishing group now requires this, as well.) Also, in this data, are the authors summing signal only cyclized hydroxyphenylthiazoline, or also the uncyclized hydroxyphenylcysteine? That would be important to know for proper interpretation of Cy domain mutations, in particular.

Furthermore, the assay design does not seem optimal. The assay performed appears to include no enzymatic nor chemical release step. It seems to rely on spontaneous hydrolysis of hydroxyphenylthiazoline from the PCP domain which is not very rapid and thus is likely the rate-limiting step, at least for the wildtype PchE. This very likely makes the data (which has not been shown) much weaker than what could be obtained in a true multiple turnover assay, and makes the mutational effects more indirectly related to the observed decreases in products.

A: Many thanks! For the true multiple-turnover assay, we use SrfD, an external thioesterase enzyme involved in surfactin biosynthesis by *Bacillus subtilis*⁴⁰, to release the intermediate from PchE. The data for the biochemical assay has been added as Supplementary Figs. 14-17. The associated text has been revised.

Line 211: “The significance of these active site residues for the Cy domain of PchE was also confirmed by site-directed mutagenesis (Supplementary Figs. 14-17). We used SrfD, an external thioesterase enzyme

involved in surfactin biosynthesis by *Bacillus subtilis*⁴⁰, to release linear and heterocyclic products. Mutations of N368, Q482, S472, Y114 and T474 to alanine reduced the heterocyclic product of PchE to approximately 28.0%, 9.8%, 50.3%, 4.6%, and 15.3%, respectively, whereas the linear intermediate production of Q482A and T474A tripled. The mutation of D480A almost abolished the peptide formation activity and was particularly detrimental to the cyclization reaction, as indicated by the substantially large ratio of the linear to heterocyclic product (Supplementary Fig. 14a). Thus, consistent with previous mutagenesis analyses^{36,37}, D480 may act.....”

36. Bloudoff, K., Fage, C. D., Marahiel, M. A. & Schmeing, T. M. Structural and mutational analysis of the nonribosomal peptide synthetase heterocyclization domain provides insight into catalysis. *Proc. Natl. Acad. Sci. U. S. A.* **114**, 95–100 (2017).

37. Dowling, D. P. *et al.* Structural elements of an NRPS cyclization domain and its intermodule docking domain. *Proc. Natl. Acad. Sci. U. S. A.* **113**, 12432–12437 (2016).

40. Steller, S. *et al.* Initiation of surfactin biosynthesis and the role of the SrfD-thioesterase protein. *Biochemistry* **43**, 11331–11343 (2004).

Supplementary Fig. 14 Activity assay of structure-guided mutations in the Cy domain. **a** The plot shows relative activities of the heterocyclic (black) and linear (purple, the intermediate) products of the key residue mutants in the Cy domain. The product ratios of linear to heterocyclic are labeled in the right table. Error bars indicate standard deviations (\pm SD) from three biologically independent experiments ($n=3$). Source data are provided as a Source Data file. **b** Representative mass spectra of the heterocyclic (left, black peak) and linear product (right, purple peak) in positive ion mode. The heterocyclic product has calculated and experimentally determined m/z $[M+H]^+$ values of 224.0376 and 224.0378, and the linear product has calculated and experimentally determined m/z $[M+H]^+$ values of 242.0482 and 242.0483.

Supplementary Fig. 15 Representative HPCL chromatogram profiles of all the PchE mutants. a Representative HPLC traces at $\lambda=254$ nm showing the products of the *in vitro* reactions catalyzed by PchD with PchE or PchE mutants. SrfD, an external thioesterase enzyme involved in surfactin biosynthesis by *Bacillus subtilis*¹², was included in the reaction mixture to release the products. **b** The plot shows relative activities of F372A in the PchE Cy domain. Error bars indicate standard deviations (\pm SD) from three biologically independent experiments ($n=3$). Source data are provided as a Source Data file.

12. Steller, S. *et al.* Initiation of surfactin biosynthesis and the role of the SrfD-thioesterase protein. *Biochemistry* **43**, 11331–11343 (2004).

Supplementary Fig. 16 Representative extracted ion chromatogram profiles of the key mutants in the Cy domain. Representative EIC traces in positive ion mode showing the heterocyclic (black) and linear (purple) products of the *in vitro* reactions catalyzed by PchD with PchE or PchE mutants, which the products were released by the thioesterase SrfD. The selective m/z $[M+H]^+$ values used to detect the products are labeled over the traces, and the heterocyclic and linear products have the calculated m/z $[M+H]^+$ values of 224.03 and 242.04, respectively.

Supplementary Fig. 17 Representative mass spectrum profiles. **a** Representative mass spectrum of the heterocyclic product (HPT-COOH) in positive ion mode. HPT-COOH has calculated and experimentally determined m/z $[M+H]^+$ values of 224.0376 and 224.0378. **b** Representative tandem mass spectrum (MS/MS) fragmentation of HPT-COOH, with the putative fragmentation pathway (**c**).

Also importantly, the manuscript requires substantial rewriting before it is acceptable for publication anywhere. First, the authors must read Katsuyama et al. *Angew Chem Int Ed Engl.* 2021, which was published online in March 2021, and which includes crystallography and cryoEM of FmoA3, and adjust their manuscript accordingly. FmoA3 is an NRPS subunit with domains Cy-A-PCP which also forms a head-to-tail dimer. The authors must include with a structural comparison between PchE and FmoA3. Then, the manuscript must be rewritten in a more scholarly fashion. There are almost no references called in the results or discussion, and essentially no structural comparisons with known structures. This will both lead

non-field expert readers to assume many well-established observations are novel to this study and prevent a field expert from understanding these data in the context of the existing body of published work. For example, the features discussed in the text about the Cy and A appear to be already well known (including their binding to CP domains), though it is difficult to tell because there is almost direct comparison to published structures. Conversely, from the E domain might be a novel type of E domain, unlike the NRPS E domain that has been solved previously, but it is difficult to tell from the text and the authors only mention that it is noncanonical. (The methods state an E domain homolog was used for modelling, so perhaps it is not novel.) In any case, the text descriptions of the domains of which tens of homologues are known (A, Cy domain) and of which this is perhaps the first (E domain) are so similar in nature that the reader would not know what is new and what is not.

It is not particularly easy to properly place a novel structural study in the context of a large body of existing literature, but including one sentence in the introduction with 11 references to say that other structures exist and then largely ignoring published work for the rest of the manuscript is not sufficient.

A: Sorry for missing the reference (Katsuyama et al. *Angew Chem Int Ed Engl.* 2021), we have now added the structural comparison between PchE and FmoA in Supplementary Fig. 10, and adjust the manuscript accordingly. Then, we have now compared the individual PchE domains (Cy, ArCP:Cy:PCP, A, and E) with the known structures, and revised the manuscript with the papers of published work cited throughout the Results section.

Please see our point-to-point answers on the “specific examples of the issues” below, for Supplementary Figs. 10-12, 18-21.

Specific examples of the issues raised above and other issues to address:

Line 21-23: “A typical synthetic elongation accomplished in a module involves thiolation, peptidyl carrier protein (PCP) donation, condensation and post-condensation steps”:

This is not a good description of a typical elongation cycle, because:

-No mention of adenylation

-Despite the PCP domain being also called the thiolation domain, “thiolation” does not describe well the transfer of the aminoacyl moiety onto the thioester of the ppant of the PCP domain

-“PCP donation” seems to imply the PCP domain is the moiety being donated

-donation by the PCP domain is the same as condensation. The upstream PCP donates its attached acyl group in the condensation reaction

-Many (most?) elongation cycles do not have post-condensation steps.

A: We have revised the sentence as follows (Line 23): “A typical synthetic elongation accomplished by a module involves substrate adenylation, intermediate shuttling, and condensation steps.....”

Line 31-32: “The multilevel structural transitions guided by a large-scale rotation of the adenylation domain, together with the inserted epimerase”:

The authors show that it is a large-scale rotation of the Asub domain, not the main A domain, but this sentence makes it seem like the whole A domain (or main portion of the A domain). This is important because the Asub rotation has been seen many times, originally by Gulick and coworkers, but a rotation of the main A domain (relative to the Cy domain) would be novel and unexpected. That the Asub rotates is made clear in the main body, and is easy to fix in the Abstract by just changing to “large-scale rotation of the adenylation C-terminal subdomain”.

A: Thanks! It's definitely a more precise description. We have revised the sentence in the Abstract section and throughout the manuscript. Line 31: “The multilevel structural transitions guided by a large-scale rotation of the adenylation C-terminal subdomain.....”

Line 44: Inappropriate sentence structure: “select, incorporate, and modify amino acids into the growing peptide”

A: We have revised the sentence as follows (Line 43): “The effectiveness of the catalysis relies on the intramodule and intermodule organization of the assembly lines of NRPSs. These enzymes select specific amino acids, incorporate them into the growing peptide, and modify the peptide backbone in a ribosome-independent manner, and altogether the family of NRPSs are able to use more than 500 precursors.”

Line 45: “with more than 500 precursors.” Add a modifying phrase or it implies NRPSs all use more than 500 substrates.

A: We have revised the sentence as follows (Line 44): “These enzymes select specific amino acids, incorporate them into the growing peptide, and modify the peptide backbone in a ribosome-independent manner, and altogether the family of NRPSs are able to use more than 500 precursors.”

ED Fig 1

-As the A_{sub} is drawn each time in light blue in this figure, it should be also drawn that way in the second module of Vlm1.

A: It has been revised in Supplementary Fig. 1b.

Lines 79-82, NRPS dimers. The authors mention VibF, but not SfmC as biochemically identified as dimers. The recent FmoA3 structure, solved by both EM and crystallography is a Cy-A-T domain head-to-tail dimer, and StsA crystals also showed a dimeric arrangement. Therefore, this statement is incorrect: “all available crystal structures support the widely accepted idea is that NRPSs are monomeric”. Moreover, lines 79-82 implies that the field was unaware that some NRPSs are dimers and that this study will reveal that NRPSs are dimers. The overwhelming majority of NRPSs are monomers, shown not only by crystal structures but also by solution techniques such as gel filtration experiments.

A: We have revised these sentences as follows (Line 90): “Moreover, despite the structurally solved dimeric FmoA3 (by both EM and crystallography)²⁰ and StsA (in crystals)²⁶; the biochemically identified dimeric VibF (by ultracentrifugation)²⁷ and SfmC (by BN-PAGE analysis)²⁸, the NRPS dimer formation has remained a minority nature in the field. Understanding the organization of E domain-embedded NRPS dimerization is hampered by the lack of such a dimeric architecture.”

20. Katsuyama, Y. *et al.* Structural and functional analyses of the tridomain-nonribosomal peptide synthetase FmoA3 for 4-methylxazoline ring formation. *Angew. Chem. Int. Ed.* **60**, 14554–14562 (2021).

26. Alonzo, D. A., Chiche-Lapierre, C., Tarry, M. J., Wang, J. & Schmeing, T. M. Structural basis of keto acid utilization in nonribosomal depsipeptide synthesis. *Nat. Chem. Biol.* **16**, 493–496 (2020).

27. Hillson, N. J. & Walsh, C. T. Dimeric structure of the six-domain VibF subunit of vibriobactin synthetase: mutant domain activity regain and ultracentrifugation studies. *Biochemistry* **42**, 766–775 (2003).

28. Koketsu, K., Watanabe, K., Suda, H., Oguri, H. & Oikawa, H. Reconstruction of the saframycin core scaffold defines dual Pictet-Spengler mechanisms. *Nat. Chem. Biol.* **6**, 408–410 (2010).

ED Fig 6. The authors are showing maps “carved” close to their model. This is fine, but can be used to artificially make the maps look better than they use a low carving distance. The carving distance should not be less than 2.5-3 Å, and must be stated in the figure legend.

A: We have revised this figure and showed several regions without carving as Supplementary Fig. 7.

Supplementary Fig. 7 Representative cryo-EM maps of the PchE domains. The displayed map including the ArCP, Cy, A_{core}, A_{sub}, and E domains is from the conformation 1_chain A structure. The map of the PCP domain is from conformation 3_chain A structure. The maps are shown as meshes, contoured at 0.012 (5.2 σ) for the Cy and A_{core} domains, and 0.010 (4.3 σ) for the ArCP, A_{sub}, and E domains, and 0.010 (4.4 σ) for the PCP domain. The figure was generated in ChimeraX. **a** The cryo-EM maps for each of the PchE domains, fitted with the cartoon represented atomic models (shown as lines). **b** Close-up views of the regions of each domain, fitted with the full-atom represented atomic models (shown as sticks).

Line 102: Edit the phrase: “the A_{sub} domain is inserted by the E domain”

A: We have revised the sentence as follows (Line 124): “.....the A_{sub} domain is interrupted by the E domain.....”.

Lines 106 – 108. The terminology being used is unclear.

In the condensation reaction, ArCP donates its acyl moiety to Cys-ppant-PCP. Therefore, the ArCP donating state and the condensing state should be the same, at least with respect to the ArCP. It would be much clearer to say in state 1 ArCP is bound to Cy, in state 2, ArCP and PCP are bound to Cy, and in state 3, one protomer has ArCP bound to Cy and the other has PCP bound to Cy.

Likewise, in Fig 2b, why it is called “Condensation/Cyclization” state? What evidence exists that both CPs are required to be bound for cyclization? On the contrary, the existence of NRPSs with Cy domains dedicated for either condensation or cyclization suggests that the donor CP is not required for cyclization.

A: We agree. We have revised this paragraph for labeling the specific conformations and chains, followed by stating the domain positions. The terminology of Fig. 2b was also revised.

Line 127: “All PchEs in the three conformations are dimeric which each of them is composed of two chains. The stable Cy- A_{core} didomain forms the main body of the chain, which is nearly identical to the FmoA3²⁰ Cy- A_{core} domain arrangement. The Cy- A_{core} didomain is similar to the common C- A_{core} interactions of the previously characterized NRPSs^{17,18,21,23} with slightly varied rotational ranges of the positions of the C domain (Supplementary Fig. 8). Relative to Cy- A_{core} , there are multiple highly dynamic domains, namely, the ArCP, PCP, and A_{sub} -E- A_{sub} domains, and these domains allow the identification of different catalytic states. In conformation 1 (Fig. 1a), chains A and B are identical with an r.m.s.d. of 0.037 Å. The ArCP domains of both chains bind at the donor sites of the Cy domains, with the salicyl-pPant ligands observed, representing the substrate donation states. The A_{core} with interrupted A_{sub} domains (complexed with Cys-AMP and Mg^{2+} ligands) are in the thioester-forming conformation but without the PCP domain captured. The E domain in this conformation may represent the epimerization conformation because the low-resolution map of the PCP domain was only observed binding to the E domain in this position (described later in Fig. 5). In conformation 2 (Fig. 1b), both ArCP and PCP domains in chain A bind to the Cy domain, representing the condensation state; the pPant arm of the ArCP domain was observed without tethered salicylate, and the pPant of PCP is merely modeled in Fig. 2b. In chain B, only the PCP domain contacts the acceptor site of the Cy domain, and the pPant arm of the PCP domain was obtained. Both chains have their A_{sub} domain rotated relative to the A_{core} (complexed with AMP), resulting in the adenylate-forming conformations. The superposition between two chains gives an r.m.s.d. of 0.851 Å (without ArCP). In conformation 3 (Fig. 1c), the PCP domain of chain A binds at the acceptor site of the Cy domain, with the hydroxyphenylthiazoliny-pPant ligand observed, representing the post-condensation state. The A domain (complexed with Cys-AMP and Mg^{2+} ligands) adopts the adenylate-forming conformation. Chain B duplicates the catalytic state as both chains of conformation 1. In addition to the dynamic domains, the Cy- A_{core} domains between two chains of conformation 3 have an r.m.s.d. of 0.566 Å. In summary, because each protomer of the three conformations is observed in the different catalytic states, dimeric PchE may still be monofunctional³⁵, with two chains forming separate catalytic units.”

17. Tanovic, A., Samel, S. A., Essen, L.-O. & Marahiel, M. A. Crystal structure of the termination module of a nonribosomal peptide synthetase. *Science* **321**, 659–663 (2008).

18. Drake, E. J. *et al.* Structures of two distinct conformations of holo-non-ribosomal peptide synthetases. *Nature* **529**, 235–238 (2016).

20. Katsuyama, Y. *et al.* Structural and functional analyses of the tridomain-nonribosomal peptide synthetase FmoA3 for 4-methylxazoline ring formation. *Angew. Chem. Int. Ed.* **60**, 14554–14562 (2021).

21. Kreidler, D. F., Gemmel, E. M., Schaffer, J. E., Wenczewicz, T. A. & Gulick, A. M. The structural basis of N-acyl- α -amino- β -lactone formation catalyzed by a nonribosomal peptide synthetase. *Nat. Commun.* **10**, 3432 (2019).

23. Reimer, J. M. *et al.* Structures of a dimodular nonribosomal peptide synthetase reveal conformational flexibility. *Science* **366**, eaaw4388 (2019).

Line 110: “two chains forming separate catalytic chambers” – This is borrowing terminology from FAS/PKS field, which would be fine, but those systems have largely enclosed catalytic chambers, which does not appear to be the case here.

A: We have revised the sentence as follows (Line 153): “.....two chains forming separate catalytic units”.

Line 111-121: This description of head-to-tail dimerization sounds very much like the recently-described FmoA3 structure. A full comparison, including text passage and figure is necessary.

A: We have now added a paragraph for the two structures comparison and a comparison figure as Supplementary Fig. 10.

Line 162: “A comparison of the overall architectures between PchE and the very recently reported FmoA3²⁰ revealed similar head-to-tail dimeric arrangements (Supplementary Fig. 10a). The dimer formations are entirely contributed by the stable Cy- A_{core} didomain interaction, whereas the dynamic domains of A_{sub} and PCP for FmoA3 and the ArCP, PCP, and A_{sub} -E- A_{sub} for PchE make no contact with the dimer interface. However, the relative angle between the two protomers is different. The interface helices (S473-N484) of FmoA3 are roughly parallel (170.3°), and this finding is distinct from that obtained for corresponding helices

(P534-G546) of PchE, which adopt a “V”-shaped arrangement with an interhelix angle of 141.8° (Supplementary Fig. 10b). Subsequently, rather than the coplanar dimer of FmoA3, one Cy- A_{core} protomer of PchE bends in the opposite direction relative to the other one, resulting in an overall tilted conformation (Supplementary Fig. 10c). Consequently, the number of interacting residues and the buried surface between the two protomers of PchE is more than twice that of FmoA (Supplementary Fig. 10d). Together, PchE forms a more complicated and compact dimeric structure than that of FmoA.”

20. Katsuyama, Y. *et al.* Structural and functional analyses of the tridomain-nonribosomal peptide synthetase FmoA3 for 4-methylxazoline ring formation. *Angew. Chem. Int. Ed.* **60**, 14554–14562 (2021).

Supplementary Fig. 10 Structural comparisons of the dimeric organization between PchE and FmoA3³. **a** Comparison of the two overall architectures and linear organizations. **b** Comparison of the two inter-helix arrangements. Close-up views of interface helices (with angles labeled) are bordered with dashed lines. **c** Comparison of the two $(\text{Cy}-A_{\text{core}})_{\text{chain A}}-(A_{\text{core}}-\text{Cy})_{\text{chain B}}$ quadrangular stable cores, shown in front, side, and top views (top panel to bottom, respectively). Left, chains A of PchE and FmoA3 are superposed and shown as lines, with chains B shown as cylinders for comparison purpose. The tilt angle between chain B of PchE and FmoA3 is labeled (bottom). Right, schematics showing the inter-chain arrangements of PchE and FmoA3. **d** Comparison of the two dimeric interfaces. The interacting residues are colored and shown as sticks. Bottom, Schematics showing contacts between chains with only chains B shown as surface representation.

Lines 123-139: There are decades of structural work on C-superfamily domains as well as a handful of structures that observe CP:C or C:CP binding. It is unscholarly not to reference them here. For example, without a reference and acknowledging that this was expected, a statement like “The PchE Cy domain is

a “V”-shaped pseudodimer that has two lobes, with each adopting the chloramphenicol acetyltransferase fold” implies this as a present discovery.

A: We have now revised this paragraph, compared with known structures, cited the references, and added the Cy comparison figure as Supplementary Fig. 11, and the CP:CP:CP binding comparison figure as Supplementary Fig. 12.

Line 179: “Similar to all the C and Cy domain structures^{17,18,20,23,36–38}, the PchE Cy domain is a “V”-shaped pseudodimer with two lobes, each of which adopt the chloramphenicol acetyltransferase fold (Fig. 2). The superposition of PchE Cy with known Cy domains^{20,36,37} shows a subtle motion of the overall structure, whereas the comparison of PchE Cy with C domains^{17,18,23,38} displays different rotational offsets and varied ranges of “openness”³⁸ between the N- and C-lobes (Supplementary Fig. 11a, b). The ArCP and the PCP contact the Cy domain on the two opposite sites, mainly through $\alpha 2$ and $\alpha 3$ of both carrier proteins. The overall PchE ArCP:CP:PCP binding mode is similar to that of the homologs^{18,23,39} with the orientation of CPs rotated slightly (Supplementary Fig. 12a-c) but is distinct from the SrfA-C¹⁷ and FmoA3²⁰ at the acceptor CP sites in which the pPant-attached serine residues of PCPs mutated to alanine (Supplementary Fig. 12c).”

17. Tanovic, A., Samel, S. A., Essen, L.-O. & Marahiel, M. A. Crystal structure of the termination module of a nonribosomal peptide synthetase. *Science* **321**, 659–663 (2008).

18. Drake, E. J. *et al.* Structures of two distinct conformations of holo-non-ribosomal peptide synthetases. *Nature* **529**, 235–238 (2016).

20. Katsuyama, Y. *et al.* Structural and functional analyses of the tridomain-nonribosomal peptide synthetase FmoA3 for 4-methyloxazoline ring formation. *Angew. Chem. Int. Ed.* **60**, 14554–14562 (2021).

23. Reimer, J. M. *et al.* Structures of a dimodular nonribosomal peptide synthetase reveal conformational flexibility. *Science* **366**, eaaw4388 (2019).

36. Bloudoff, K., Fage, C. D., Marahiel, M. A. & Schmeing, T. M. Structural and mutational analysis of the nonribosomal peptide synthetase heterocyclization domain provides insight into catalysis. *Proc. Natl. Acad. Sci. U. S. A.* **114**, 95–100 (2017).

37. Dowling, D. P. *et al.* Structural elements of an NRPS cyclization domain and its intermodule docking domain. *Proc. Natl. Acad. Sci. U. S. A.* **113**, 12432–12437 (2016).

38. Bloudoff, K., Rodionov, D. & Schmeing, T. M. Crystal structures of the first condensation domain of CDA synthetase suggest conformational changes during the synthetic cycle of nonribosomal peptide synthetases. *J. Mol. Biol.* **425**, 3137–3150 (2013).

39. Zhang, J. *et al.* Structural basis of nonribosomal peptide macrocyclization in fungi. *Nat. Chem. Biol.* **12**, 1001–1003 (2016).

Supplementary Fig. 11 Structural comparisons of Cy and C domain conformations. Top, superposition of PchE Cy domain with other NRPSs Cy domains (**a**) and C domains (**b**) on the C-lobes. Bottom, individual superposition of PchE Cy domain with BmdB⁸ (PDB 5T3E), EpoB⁹ (PDB 5T81), and FmoA3³ (PDB 6LTA) Cy domains (**a**); and with CDA¹⁰ (PDB 4JN3), SrfAC⁷ (PDB 2VSQ), EntF⁴ (PDB 5T3D), LgrA⁵ (PDB 6MFW) C domains (**b**). PchE Cy domain is colored purple, other Cy and C domains are shown in gray (transparent), with regions variously colored and bordered for highlighting the structural differences (loop linked β -sheet and α -helix for the Cy domains, and the entire N-lobes for the C domains). The relative rotations between the N-lobe of C domains with PchE Cy domain are highlighted, showing the inter-lobe conformational variability.

Supplementary Fig. 12 Structural comparisons of CP: Cy/C binding. All structures are superposed onto the PchE Cy domain for CP domains binding comparison at the donor site (a), acceptor site (c) and both sites (b). PchE ArCP: Cy: PCP domains are colored red, purple and yellow, respectively. Other NRPSs CPs are variously colored (transparent) for highlighting the CPs binding differences between PchE and TqaA¹¹ (a, PDB 5EJD); LgrA⁵ (b, PDB 6MFZ); and AB3403⁴ (c, PDB 4ZXH), FmoA3³ (c, PDB 6LTA), and SrfA-C⁷ (c, PDB 2VSQ). The $\alpha 1$, 2, and 4 helices of CP domains are labeled and the pPant-attached Ser residues (or Ser-Ala) are shown as balls.

Lines 134-153: Likewise, please cite references 23 and 24 in this sentence: “The catalytic residue D480 aligns well with D1226 and D449 in the BmdB (r.m.s.d. 1.80 Å) and EpoB (r.m.s.d. 2.27 Å) Cy homologs, respectively”? A comparison between the mutagenesis analyses here and in those papers, or at the very least a statement about the mechanism of the Cy domain based on the conclusions of those papers is necessary to understand the mutational analysis here.

A: We have now cited the references and included the brief statement about the mechanism of Cy based on the conclusions of the two papers. Please also see our answer on the major point of “For the biochemical assay, the authors do not show any of the data....”.

Line 200: “The catalytic residue D480 aligns well with D1226 and D449 in BmdB³⁶ (r.m.s.d. 1.837 Å) and EpoB³⁷ (r.m.s.d. 2.127 Å) Cy homologs, respectively.”

Line 208: “A typical Cy domain catalyzes two separate reactions: condensation and a two-step cyclodehydration. Based on mutational analyses of BmdB and EpoB, an N residue was proposed to facilitate the positioning role in the condensation reaction, and D and T residues were identified as crucial for the heterocyclization reaction^{36,37}. The significance of these active site residues for the Cy domain of PchE was also confirmed by site-directed mutagenesis.....”

36. Bloudoff, K., Fage, C. D., Marahiel, M. A. & Schmeing, T. M. Structural and mutational analysis of the nonribosomal peptide synthetase heterocyclization domain provides insight into catalysis. *Proc. Natl. Acad. Sci. U. S. A.* **114**, 95–100 (2017).

37. Dowling, D. P. *et al.* Structural elements of an NRPS cyclization domain and its intermodule docking domain. *Proc. Natl. Acad. Sci. U. S. A.* **113**, 12432–12437 (2016).

Lines 157-162: By “loop lid”, do the authors mean the element which has been called “the latch” by many papers? If so, the statement “Consequently, these open conformation-adopting two lids relieve a strong steric clash that would otherwise exist upon ArCP and PCP joining with Cy and subsequent pPant substrate entry” requires further explanation. There are many C domains that have the latch closed and still show a conformation that have an open tunnel that would accommodate pPant binding. Why would latch opening be required here? In general, the authors seem to only make structural comparisons of their Cy domain to the two excised Cy domain structures (and not the recent Cy-A-PCP structure), but ignore the large body of structural information from the related C domain.

A: Yes, one of them is the latch and we agree this statement is inappropriate. We have now deleted this statement and revised this sentence as follows (Line 232): “.....the opposite loop (the latch⁴¹) moved ~11.5 Å on the side of the ArCP-binding region compared with EpoB³⁷ (Fig. 2d, left and right, respectively). Although there is no evidence showing that the lid loop (near PCP) and the latch open and close³⁸, the translations between the two loops with the Cy homologs reveal their conformational dynamics.”

37. Dowling, D. P. *et al.* Structural elements of an NRPS cyclization domain and its intermodule docking domain. *Proc. Natl. Acad. Sci. U. S. A.* **113**, 12432–12437 (2016).

38. Bloudoff, K., Rodionov, D. & Schmeing, T. M. Crystal structures of the first condensation domain of CDA synthetase suggest conformational changes during the synthetic cycle of nonribosomal peptide synthetases. *J. Mol. Biol.* **425**, 3137–3150 (2013).

41. Keating, T. A., Marshall, C. G., Walsh, C. T. & Keating, A. E. The structure of VibH represents nonribosomal peptide synthetase condensation, cyclization and epimerization domains. *Nat. Struct. Biol.* **9**, 522–526 (2002).

Lines 163-171: Was the blockage of the donor side of the tunnel discussed in BmdB or EpoB papers? This is presented as novel and not referenced. Furthermore, a ~20% reduction in activity is not compelling for an important gatekeeping residue. Why isn't this just as easily explained by the mutation causing partial loss of interaction between Cy and ArCP domains?

A: Yes, it was discussed in BmdB paper. In the BmdB Cy structure, the corresponding F residue completely blocks the tunnel and was proposed that it must move to allow alanyl-PCP to bind and participate in the condensation reaction³⁶. The novelty presented here is that we observed the two distinct conformations of the catalytic states-related residue F372 which caused the open or closed tunnel.

We agree that the word of “gatekeeping” is overstated, and have now revised this sentence as follows (Line 239): “In the BmdB Cy structure, the residue F1118 completely blocked the tunnel. It was previously proposed that F1118 must move to allow alanyl-PCP to bind and subsequently participate in the condensation reaction³⁶. Similarly, the “Y” tunnel of PchE Cy is truncated at the side of ArCP by the residue F372. In contrast, this residue is rotated ~106° away.....”

Line 245: “The mutagenesis of F372A reduced the activity of PchE to 31.6%. Together, the results indicate that the residue F372 may play a role in changing the catalytic state in the Cy domain of PchE by switching the closed tunnel to open for donor ArCP-attached substrate interactions and the subsequent condensation reaction.”

36. Bloudoff, K., Fage, C. D., Marahiel, M. A. & Schmeing, T. M. Structural and mutational analysis of the nonribosomal peptide synthetase heterocyclization domain provides insight into catalysis. *Proc. Natl. Acad. Sci. U. S. A.* **114**, 95–100 (2017).

Lines 172-185: There are ~60 structures of A domains available, including with aminoacyl adenylates captured by crystallography. Is the brief comparison with PheA the most relevant? Is this the first Cys-specific A domain to be determined?

A: We briefly compared the A domain of PchE with PheA mainly because it is the first structurally solved A domain of NRPS. We have now compared five A domains with PchE in Supplementary Fig. 18, and yes, to our knowledge, this is the first determined Cys-specific A domain.

Line 249: “Similar to the known adenylation domain structures^{17–19,42,43} (Supplementary Fig. 18a), the A domain comprises two subdomains.....”

Line 260: “.....The active site of PchE is closely similar to that of the AB3403¹⁸ A domain, which contains two pairs of hydrophobic interaction residues, namely, I846:W847 (V768:W769 for AB3403) and F741:L743 (F669:I671 for AB3403), whereas the latter pair does not exist in the LgrA¹⁹, DhbE⁴², PheA⁴³, and SrfA-C¹⁷ A domains (Supplementary Fig. 18b). Compared with the homologs, the two pairs of hydrophobic interactions narrow the pocket to create a specific cysteine sidechain-binding surface.”

Supplementary Fig. 18 Structural comparison of A domains. **a** Superposition of PchE A domain with other NRPSs A domains on the A_{core} (ribbons, transparent), with the A_{sub} domain shown as lines. **b** The

active sites of PchE, AB3403⁴ (PDB 4ZXH), LgrA¹³ (PDB 5ES7), DhbE¹⁴ (PDB 1MDB), PheA¹⁵ (PDB 1AMU), and SrfA-C⁷ (PDB 2VSQ) A domains, shown as sticks, with the ligands colored green.

17. Tanovic, A., Samel, S. A., Essen, L.-O. & Marahiel, M. A. Crystal structure of the termination module of a nonribosomal peptide synthetase. *Science* **321**, 659–663 (2008).

18. Drake, E. J. *et al.* Structures of two distinct conformations of holo-non-ribosomal peptide synthetases. *Nature* **529**, 235–238 (2016).

19. Reimer, J. M., Aloise, M. N., Harrison, P. M. & Schmeing, T. M. Synthetic cycle of the initiation module of a formylating nonribosomal peptide synthetase. *Nature* **529**, 239–242 (2016).

42. May, J. J., Kessler, N., Marahiel, M. A. & Stubbs, M. T. Crystal structure of DhbE, an archetype for aryl acid activating domains of modular nonribosomal peptide synthetases. *Proc. Natl. Acad. Sci. U. S. A.* **99**, 12120–12125 (2002).

43. Conti, E., Stachelhaus, T., Marahiel, M. A. & Brick, P. Structural basis for the activation of phenylalanine in the non-ribosomal biosynthesis of gramicidin S. *Embo J.* **16**, 4174–4183 (1997).

Lines 178-182: “The predicted specificity-conferring code for L-cysteine recognition is the DLFNLSLIWK 10 AA signature sequence.... The inaccuracy of prediction by the structure-projection method probably results...” The 10 AA signature sequence is meant to allow prediction of substrate specificity, and it does. Submitting the PchE sequence into “PKS/NRPS Analysis Web-site” or “NRPS Predictor2” for example accurately predicts cysteine as the substrate. The code is not necessarily meant to list the residues that contact the substrate amino acid. For example, A domains specific for glycine will have a 10 AA signature sequence as well, but 10 residues won’t contact the substrate glycine.

A: Yes, we apologize and deleted the wrong vocabulary of “inaccuracy”, and revised this description as follows (Line 256): “The structure-projection method successfully predicted the specificity-conferring code⁴⁴ for L-cysteine recognition and yielded the DLFNLSLIWK 10 AA signature sequence⁴⁵.”

44. Stachelhaus, T., Mootz, H. D. & Marahiel, M. A. The specificity-conferring code of adenylation domains in nonribosomal peptide synthetases. *Chem. Biol.* **6**, 493–505 (1999).

45. Röttig, M. *et al.* NRSPredictor2—a web server for predicting NRPS adenylation domain specificity. *Nucleic Acids Res.* **39**, W362–W367 (2011).

Lines 186-195: It is impossible for the reader to understand whether this movement is the same of different to previously-observed movements. And because there is no reference to the statement “It is known that the A_{sub} domain rotates relative to the A_{core} domain”, the authors don’t direct the readers to the papers they would need to read / structures they would need to download to figure that out. Also in Video 2, it’s somewhat confusing that the open confirmation for substrate binding (called the “open” confirmation here and in the literature) is labelled as “closed” for PCP blocking.

A: We have now revised this paragraph with references cited, and added the comparison figure of A_{sub} movements as Supplementary Fig. 19. Video 2 has been deleted due to firmly established movements of this kind.

Line 266: “The domain alteration strategy of the adenylation domain has been firmly established⁴⁶: the A_{sub} domain rotates relative to A_{core} to adopt distinct catalytic states for adenylation or thiolation reactions. Similar to the known states of homologs^{18,19,47,48} (Supplementary Fig. 19), in PchE, a large-scale ~155° A_{sub} rotation is mediated by the hinge residue⁴⁹ D944 and coupled to catalytic state switching (Fig. 3a).”

18. Drake, E. J. *et al.* Structures of two distinct conformations of holo-non-ribosomal peptide synthetases. *Nature* **529**, 235–238 (2016).

19. Reimer, J. M., Aloise, M. N., Harrison, P. M. & Schmeing, T. M. Synthetic cycle of the initiation module of a formylating nonribosomal peptide synthetase. *Nature* **529**, 239–242 (2016).

46. Gulick, A. M. Conformational dynamics in the acyl-CoA synthetases, adenylation domains of non-ribosomal peptide synthetases, and firefly luciferase. *ACS Chem. Biol.* **4**, 811–827 (2009).

47. Gulick, A. M., Starai, V. J., Horswill, A. R., Homick, K. M. & Escalante-Semerena, J. C. The 1.75 Å crystal structure of acetyl-CoA synthetase bound to adenosine-5'-propylphosphate and coenzyme A. *Biochemistry* **42**, 2866–2873 (2003).

48. Gulick, A. M., Lu, X. & Dunaway-Mariano, D. Crystal structure of 4-chlorobenzoate:CoA ligase/synthetase in the unliganded and aryl substrate-bound states. *Biochemistry* **43**, 8670–8679 (2004).

49. Wu, R., Reger, A. S., Lu, X., Gulick, A. M. & Dunaway-Mariano, D. The mechanism of domain alternation in the acyl-adenylate forming ligase superfamily member 4-chlorobenzoate: coenzyme A ligase. *Biochemistry* **48**, 4115–4125 (2009).

Supplementary Fig. 19 Structural comparisons of the small C-terminal A_{sub} domain rotation. All structures are superposed onto the PchE A_{core} domain for A_{sub} domains rotation comparison. The A_{core} domains are colored blue with structures of comparison shown as transparent, the A_{sub} domains are highlighted in distinct colors, and the embedded E domain of PchE are colored green (transparent). **a** Superposition of the thioester-forming conformations of PchE A domain structure with LgrA¹³ (PDB 5ES8), EntF⁴ (PDB 5T3D), and Acs¹⁶ (Acetyl-CoA synthetase, PDB 1PG4) structures. **b** Superposition of the adenylate-forming conformations of PchE A domain structure with LgrA¹³ (PDB 5ES5), AB3403⁴ (PDB 4ZXH), and CBL¹⁷ (4-Chlorobenzoate:CoA ligase, PDB 1T5H) structures. **c** Individual superposition of different adenylating enzyme structures reveals the similar C-terminal subdomain movements. The rotation angles were reported by DynDom software¹⁸, and the direction of the arrowed lines indicate the catalytic states transitions from adenylate-forming to thioester-forming.

18. Hayward, S. & Berendsen, H. J. C. Systematic analysis of domain motions in proteins from conformational change: new results on citrate synthase and T4 lysozyme. *Proteins* **30**, 144–154 (1998).

Lines 196-200: How does the E domain insertion compare structurally to Ref 22? Are similar non-covalent contacts made between E and A_{sub} as are made between MT and A_{sub}? Is the insertion point precisely the same? (“Between A8 and A9 motifs” does not precisely define the insertion point.)

A: We have now revised this paragraph and added the comparison figure of the interrupted A_{sub} domain as Supplementary Fig. 20.

Line 283: “.....Rather than an independent domain generally following the carrier protein, the embedded E domain is inserted into the A_{sub} domain between the A8 and A9 motifs⁵¹ (Fig. 4a, b), and this domain followed the residue P990 and located just before the residue D1291. Similar to the insertion point and noncovalent interactions of the MT domain-embedded TioS⁵³ structure, the PchE E domain significantly contacts the small A_{sub} domain with a buried surface of 865 Å² (Supplementary Fig. 20b-f). Due to this

architecture, the large-scale A_{sub} domain rotation results in the dramatic backward swinging of the E domain (Fig. 3a), and the E and A_{sub} domains rotate as a rigid body (Supplementary Fig. 20a).”

51. Marahiel, M. A., Stachelhaus, T. & Mootz, H. D. Modular peptide synthetases involved in nonribosomal peptide synthesis. *Chem. Rev.* **97**, 2651–2674 (1997).

53. Mori, S. *et al.* Structural basis for backbone N-methylation by an interrupted adenylation domain. *Nat. Chem. Biol.* **14**, 428–430 (2018).

Supplementary Fig. 20 Structural comparisons of the tailoring domain-embedded didomains of PchE and TioS¹⁹. **a** Superposition of the $A_{\text{sub}}\text{-E-}A_{\text{sub}}$ didomains of the two conformations of PchE. The small difference between them indicates that they rotate as a rigid body. **b** The interface between A_{sub} and E domains of PchE, with the contacting residues shown as balls. **c** Cartoon representation of $A_{\text{sub}}\text{-M-}A_{\text{sub}}$ didomain of TioS¹⁹ (PDB 5WMM) structure, with the interface residues (**d**) shown as balls. **e** Superposition of the tailoring domain-embedded structures on the interrupted A_{sub} domains, with the close-up views of the insertion sites (**f**).

Lines 197 and 206: The E domain is referred to as a “stuffed E domain”. This follows one group’s terminology from work with MT domains, but it is not used by any other group in the field and is jargony. Please use a more formal term than “stuffed”, perhaps that the E domain is “embedded” or the A_{sub} is “interrupted” by the E domain.

A: Thank you for the suggestion! We have now used the term of “embedded” in the manuscript.

Line 284: “.....the embedded E domain is inserted into the A_{sub} domain between the A8 and A9 motifs⁵¹.....”

Lines 201-214: Canonical E domains are C domain superfamily proteins. This E domain appears to be of different evolutionary origin. The authors simply mention that it is “a noncanonical epimerase”. What is the E domain evolutionarily related to? Is this a novel domain structure?

A: It is more structurally related to the methyltransferase domain. We have added the comparison figure of PchE epimerase domain with MT domains as Supplementary Fig. 21.

Line 291: “Rather than the canonical form⁵⁴, which belongs to the C domain superfamily proteins, the PchE E domain is more structurally similar to the methyltransferase (MT) domain^{53,55,56} (Supplementary Fig. 21).”

Supplementary Fig. 21 Structural comparisons of the PchE noncanonical E domain, the C superfamily canonical E domain, and the methyltransferase (MT) domains. Superposition of PchE E domain (green) with MT domains on the SAM-binding regions (dashed line bordered) of GPPCMeT²⁰ (geranyl diphosphate C-methyltransferase, PDB 3VC2), LovB_CMeT²¹ (lovastatin nonaketide synthase C-methyltransferase, PDB 7CPX), and TioS_M4¹⁹ (PDB 5WMM) domains. The substrate entrances are indicated by arrows. The canonical epimerase of GrsA²² (PDB 5ISX) structure is dashed line bordered and colored pink, showing the structural difference between the C domain superfamily epimerase and noncanonical epimerase of PchE.

53. Mori, S. *et al.* Structural basis for backbone N-methylation by an interrupted adenylation domain. *Nat. Chem. Biol.* **14**, 428–430 (2018).

54. Chen, W.-H., Li, K., Guntaka, N. S. & Bruner, S. D. Interdomain and intermodule organization in epimerization domain containing nonribosomal peptide synthetases. *ACS Chem. Biol.* **11**, 2293–2303 (2016).

55. Köksal, M., Chou, W. K. W., Cane, D. E. & Christianson, D. W. Structure of geranyl diphosphate C-methyltransferase from *Streptomyces coelicolor* and implications for the mechanism of isoprenoid modification. *Biochemistry* **51**, 3003–3010 (2012).

56. Wang, J. *et al.* Structural basis for the biosynthesis of lovastatin. *Nat. Commun.* **12**, 867 (2021).

Line 211: “E1234 could donate a proton”: A mention of the typical protonation state and pKa of glutamate residues, and how that might be altered here is warranted.

A: E1234 could also function as a general base. Line 297-304 have been revised: “The mechanism of the catalysis of the epimerization reaction by canonical epimerization domains has been well investigated and proposed^{54,57,58}. Briefly, a pair of histidine and glutamate residues function as the general acid-base catalyst.

In PchE, a general acid residue needed for the mechanism has not been pinpointed due to the lack of an embedded epimerase structure. In the search for acidic residues, we found E1234 located at the end of the tunnel. Nonetheless, the H1204 and E1234 residues likely act as general acid and base and catalyze the formation of β -thiazoline, which mimics the mechanism observed with the epimerization domain from the initiation module of *B. brevis* gramicidin S synthetase⁵⁴ and tyrocidine synthetase A⁵⁸.”

54. Chen, W.-H., Li, K., Guntaka, N. S. & Bruner, S. D. Interdomain and intermodule organization in epimerization domain containing nonribosomal peptide synthetases. *ACS Chem. Biol.* **11**, 2293–2303 (2016).

57. Stachelhaus, T. & Walsh, C. T. Mutational analysis of the epimerization domain in the initiation module PheATE of gramicidin S synthetase. *Biochemistry* **39**, 5775–5787 (2000).

58. Samel, S. A., Czodrowski, P. & Essen, L.-O. Structure of the epimerization domain of tyrocidine synthetase A. *Acta Cryst. D* **70**, 1442–1452 (2014).

Line 230: “near-continuous conformations”, Video 3 and Line 33-35 “Moreover, using deep neural network calculations, we visualized the near-continuous structural dynamics of the full catalytic cycle, from thiolation to epimerization.”

The cryoDRGN work is nice, but the Video 3 clearly shows most of the transition being a disappearance of a domain at one position followed by its reappearance elsewhere. This should not be described as near-continuous.

A: We have revised the sentence and throughout the text (Line 34): “Moreover, using deep neural network calculations, we visualized the direct structural dynamics of the full catalytic cycle from thiolation to epimerization.”

Lines 247-254. Must be modified in the re-write after authors read Katsuyama et al. *Angew Chem Int Ed Engl.* 2021, which was published online in March 2021.

A: Thanks. We have now revised this paragraph as follows (Line 347): “Among the NRPS family of megaenzymes, homodimerization is a minor and unique phenomenon, and architecture information is indispensable for the engineering of chimeric NRPS modules. Compared with the very recently reported dimeric FmoA3²⁰ structure, PchE exhibits a similar antiparallel head-to-tail architecture. However, the curved PchE dimer results in a significantly more extensive interface between the two Cy-A_{core} protomers connected with additional moving domains (ArCP, PCP, and A_{sub}-E-A_{sub}). The Cy-A_{core} dimer is rigid and stable in all the conformations observed in this study, and the observation of two carrier protein condensations within one monomer indicates that they are most likely monofunctional, with each monomer catalyzing reactions independently. The overall structural arrangements of FmoA3 and PchE complement each other and provide a better understanding of NRPS homodimerization.”

20. Katsuyama, Y. et al. Structural and functional analyses of the tridomain-nonribosomal peptide synthetase FmoA3 for 4-methylxazoline ring formation. *Angew. Chem. Int. Ed.* **60**, 14554–14562 (2021).

Figure 2, 3, ED Fig 6.

-Maps absolutely have to be shown for all the ligands. Only models and distances (to the tenth on an Å) are shown, and the reviewers and readers must be able to judge the quality of the data that leads to this model. State 2 is modelled as pre-condensation and state 3 is modelled as post-cyclization without showing the evidence.

A: Please see the previous answers (“The cryoEM maps for the ligands (acyl-pPants, adenylates) are not shown anywhere and absolutely must be” of major point 1, “ED Fig 6. The authors are showing maps “carved” close to their model” of specific examples of the issues).

We have now added the figure for the cryo-EM maps of the experimentally observed ligands as Supplementary Fig. 6, and revised the ED Fig 6. to show several regions of PchE domains without carving as Supplementary Fig. 7.

Figure 5 and elsewhere in the text: Cryo-EM maps are Coulomb potential maps, not density maps.

A: As suggested, we have now deleted the “density” term, and used the cryo-EM maps throughout the manuscript.

ED Fig 10 and Video 3: The point cloud representation of the cryoDRGN processing saturates, so the ability to see how many particles are where is lost. This representation is far more useful if you can see localised accumulations of species within the cloud.

A: We have now revised the ED Fig. 10 to show the localized accumulations of six representative cryoDRGN reconstructions in the separate subplots (Supplementary Fig. 22c). We apologize that we were unable to show every species of the trajectory using the software and therefore keep the Video 3 as unchanged.

Supplementary Fig. 22 Structural heterogeneity in the PchE dataset analyzed by using cryoDRGN software (related to Fig. 5 and Supplementary Movie 2). **a** UMAP visualization of the latent space representation of particle images of PchE after training an 8-D latent variable model with cryoDRGN. **b** PCA projection of the 8-D latent encodings from cryoDRGN with 6 representative sample points. **c** Subplots of the localized accumulations of six representative cryoDRGN reconstructions. **d** Cryo-EM maps generated at points shown in (b). These maps are docked with atomic models of each domain in Fig. 5.

References: There are several formatting issues (e.g. capitalization, journal names) in the references.

A: We have now corrected the formatting issues in the References section.

Reviewer #2 (Remarks to the Author):

Nonribosomal peptide synthetase (NRPS) proteins are modular assembly line proteins that produce important peptide natural products. The NRPS enzymes use a multidomain architecture and a fascinating biosynthetic strategy in which the nascent peptide is tethered to an integrated carrier protein domain and transported to catalytic domains for synthesis. Recent structure advances have provided clearer views of multiple states of the NRPS modules in different stages of the active cycle. Additional structures of multi-domain NRPS proteins, as well as insights into the molecular trajectories between individual states, are important to understand completely the nature of the NRPS structural and catalytic cycle.

Here, Wang and colleagues describe the cryo-Electron Microscopy structures of PchE, an NRPS protein from the *Pseudomonas pyochelin* biosynthesis pathway. This protein contains several interesting features. It is a five domain protein with a ArCP-Cyclization-Adenylation-Epimerization-PCP architecture. The Adenylation domain contains two subdomains (N- and C-terminal) and the epimerization domains is inserted into the highly dynamic C-terminal subdomain. The observation of multiple states of the PchE protein provides a better understanding of the interactions of the different domains and insights into the known conformational changes of the NRPS adenylation domains that foster the progression of the structural cycle. The manuscript is largely well-written and will provide an important advance for the field of NRPS structural biology.

Dear Professor Andrew Gulick,

We are grateful for your highly positive comments on our work. Your research articles and review papers on the NRPS proteins are insightful and quite exceptional.

These comments are very helpful to the quality promotion of this manuscript. The points have been carefully addressed point by point, the new figures have been added and revised, and the manuscript has been reorganized and corrected to a clearer understandable version for readers, thanks to your professional suggestions.

There are places, however, that fail to fully explain the techniques used and that fail to distinguish between experimental observations and models. The manuscript needs to better elaborate the structures and more completely describe the cryoDRGN analysis. A clearer description of the literature on the NRPS proteins will also provide the necessary context for the presented work.

1. An important observation of this structure is that the PchE adopts a head-to-tail dimeric structure, where the dimeric interface is formed by the central Cyclization-Adenylation domain core. While the H-shaped cryoEM reconstructions are clear, the authors proceed directly into the analysis of this dimeric structure without properly putting this in context. The size exclusion column (line 88 and ED Figure 2) is presented without mw standards. I recommend a new section to begin the Results portion of the manuscript that is titled "PchE forms a dimeric quaternary structure". The experimental results here establish that the protein exists as a dimer in the size exclusion experiment. The cryoEM results in a subsequent section present the nature of this dimeric protein.

A: Thanks, it is definitely a better manuscript organization in this way. We have now included the section titled "PchE forms a homo-oligomeric quaternary structure" in the beginning of the Results section.

Line 97: **"PchE forms a homo-oligomeric quaternary structure"**

PchE and PchF were previously successfully purified with biochemical activities⁹ reconstituted by the Christopher T. Walsh lab, and the predicted (theoretical) monomer molecular weights of the two subunits were approximately 156 kDa (PchE) and 197 kDa (PchE) (Supplementary Fig. 2a). However, the size exclusion chromatography profile clearly shows that the apparent molecular weight of PchE is larger than that of PchF, and the native-PAGE analysis also supports this observation (Supplementary Fig. 2b, c). Based on these experiments, we speculated that PchE exhibits a homo-oligomeric organization. To gain further insight into this enzyme, we determined the structure of PchE by cryo-EM analysis."

Supplementary Fig. 2 Purification and oligomeric state analysis of PchE in solution. **a** SDS-PAGE analysis of purified PchE, PchD and PchF. The predicted (theoretical) molecular weights are labeled respectively. **b** Native-PAGE analysis of PchE, PchF, and the mixture of PchE and PchF. A 4-20% gradient was used. Note that the migration of PchE in the gel is slower than that of PchF. **c** Size exclusion chromatography (SEC, Superose 6 Increase) profile of PchE and PchF. Note that PchE elutes earlier than PchF. The apparent molecular weights (WMs) of the two proteins are approximately 484 kDa (PchE) and 322 kDa (PchF). The obtained overestimated apparent WMs of PchE and PchF may be due to their elongated and dynamic conformations, because the shape and Stokes radius of the samples can vary significantly from the largely commercial globular standard proteins, which the extended shape of proteins can easily result in an anomalously earlier elution from the size exclusion chromatographic column^{1,2}. Despite the overestimated apparent MWs of the two proteins, the hypothesis of PchE being homo-oligomeric is reasonable, based on the MWs comparison between Supplementary Fig. 2a and b-c. **d** SDS-PAGE analysis of purified PchE with a C-terminus Strep-tag II and PchD with an N-terminus His-tag. **e** Native-PAGE analysis of PchE. A 4-20% gradient was used. **f** Size exclusion chromatography profile of PchE used for cryo-EM. One representative result from at least three independent experiments is shown (a, b, d, and e). Source data are provided as a Source Data file.

1. Erickson, H. P. Size and shape of protein molecules at the nanometer level determined by sedimentation, gel filtration, and electron microscopy. *Biol. Proced. Online* **11**, 32–51 (2009).
 2. Sorensen, B. R. & Shea, M. A. Calcium binding decreases the stokes radius of calmodulin and mutants R74A, R90A, and R90G. *Biophys. J.* **71**, 3407–3420 (1996).

2. It is then necessary to provide a complete description of the overall architecture of the three conformations that were reconstructed. Relative to the Cyclization/Adenylation core, there are multiple highly dynamic protein domains: the ArCP, the PCP, the C-terminal subdomain of the adenylation domain

and the epimerization domain. Further, there are two chains of the dimer. While this section currently gives an overview of the nature of the carrier proteins relative to the Cyclization domain, it fails to describe the overall architecture and conformational changes of the other important domains/subdomains. Perhaps the authors would consider labeling the two chains of each Dimer in Figure 1 as chains A and B to establish a basis so that the discussion of different states can be referred to as “Conformation 1, Chain A” for example. Similarly, other than the more dramatic difference between the two chains of Conformation 3, it would be helpful to have some sense (an rms comparison, perhaps) of the difference observed in the two chains for each conformation.

A: Thanks, we have labeled the two chains of each dimer conformation in Fig. 1 and revised this paragraph to provide a complete description of the overall architecture of the three conformations.

Line 127: “All PchEs in the three conformations are dimeric which each of them is composed of two chains. The stable Cy- A_{core} didomain forms the main body of the chain, which is nearly identical to the FmoA3²⁰ Cy- A_{core} domain arrangement. The Cy- A_{core} didomain is similar to the common C- A_{core} interactions of the previously characterized NRPSs^{17,18,21,23} with slightly varied rotational ranges of the positions of the C domain (Supplementary Fig. 8). Relative to Cy- A_{core} , there are multiple highly dynamic domains, namely, the ArCP, PCP, and A_{sub} -E- A_{sub} domains, and these domains allow the identification of different catalytic states. In conformation 1 (Fig. 1a), chains A and B are identical with an r.m.s.d. of 0.037 Å. The ArCP domains of both chains bind at the donor sites of the Cy domains, with the salicyl-pPant ligands observed, representing the substrate donation states. The A_{core} with interrupted A_{sub} domains (complexed with Cys-AMP and Mg^{2+} ligands) are in the thioester-forming conformation but without the PCP domain captured. The E domain in this conformation may represent the epimerization conformation because the low-resolution map of the PCP domain was only observed binding to the E domain in this position (described later in Fig. 5). In conformation 2 (Fig. 1b), both ArCP and PCP domains in chain A bind to the Cy domain, representing the condensation state; the pPant arm of the ArCP domain was observed without tethered salicylate, and the pPant of PCP is merely modeled in Fig. 2b. In chain B, only the PCP domain contacts the acceptor site of the Cy domain, and the pPant arm of the PCP domain was obtained. Both chains have their A_{sub} domain rotated relative to the A_{core} (complexed with AMP), resulting in the adenylate-forming conformations. The superposition between two chains gives an r.m.s.d. of 0.851 Å (without ArCP). In conformation 3 (Fig. 1c), the PCP domain of chain A binds at the acceptor site of the Cy domain, with the hydroxyphenylthiazolanyl-pPant ligand observed, representing the post-condensation state. The A domain (complexed with Cys-AMP and Mg^{2+} ligands) adopts the adenylate-forming conformation. Chain B duplicates the catalytic state as both chains of conformation 1. In addition to the dynamic domains, the Cy- A_{core} domains between two chains of conformation 3 have an r.m.s.d. of 0.566 Å. In summary, because each protomer of the three conformations is observed in the different catalytic states, dimeric PchE may still be monofunctional³⁵, with two chains forming separate catalytic units.”

17. Tanovic, A., Samel, S. A., Essen, L.-O. & Marahiel, M. A. Crystal structure of the termination module of a nonribosomal peptide synthetase. *Science* **321**, 659–663 (2008).

18. Drake, E. J. *et al.* Structures of two distinct conformations of holo-non-ribosomal peptide synthetases. *Nature* **529**, 235–238 (2016).

20. Katsuyama, Y. *et al.* Structural and functional analyses of the tridomain-nonribosomal peptide synthetase FmoA3 for 4-methyloxazoline ring formation. *Angew. Chem. Int. Ed.* **60**, 14554–14562 (2021).

21. Kreidler, D. F., Gemmell, E. M., Schaffer, J. E., Wenczewicz, T. A. & Gulick, A. M. The structural basis of N-acyl- α -amino- β -lactone formation catalyzed by a nonribosomal peptide synthetase. *Nat. Commun.* **10**, 3432 (2019).

23. Reimer, J. M. *et al.* Structures of a dimodular nonribosomal peptide synthetase reveal conformational flexibility. *Science* **366**, eaaw4388 (2019).

35. Tang, G.-L., Cheng, Y.-Q. & Shen, B. Chain initiation in the leinamycin-producing hybrid nonribosomal peptide/polyketide synthetase from *Streptomyces atroolivaceus* S-140 discrete, monofunctional adenylation enzyme and peptidyl carrier protein that directly load d-alanine. *J. Biol. Chem.* **282**, 20273–20282 (2007).

3. Equally important, this section should clearly establish the ligands that are present in the two chains of the three conformations. It is important to state what complexes are experimentally observed to differentiate later discussions of active site contents that may result from modeling of ligands or comparisons with prior structures.

A: Thanks, we have also included the ligands description in the same paragraph for the overall architecture of the three conformations (Line 127). Please see the previous answer (major point 2) and Supplementary Fig. 6.

Supplementary Fig. 6 Representative cryo-EM maps of the PchE ligands. The cryo-EM maps of the ligands observed in this study, shown as meshes, are displayed at different contour levels and generated in ChimeraX. The colors and labels of distinct conformations and chains are corresponding to Fig. 1. **a** Map of the Ser46 salicyl-pPant arm of the ArCP domain in conformation 1_chain A, contoured at 0.007 (3.0σ). This ligand was also observed similarly in conformation 3_chain B (not shown). **b** Map of the Ser1385 Hydroxyphenylthiazoliny-pPant arm of the PCP domain in conformation 3_chain A, contoured at 0.006 (2.8σ). **c** Map of the Ser46 pPant arm of the ArCP domain in conformation 2_chain A, contoured at 0.010 (4.7σ). **d** Map of the Ser1385 pPant arm of the PCP domain in conformation 2_chain B, contoured at 0.006 (2.8σ). **e** Maps were contoured at 0.012 (Left, 5.2σ ; middle, 5.5σ ; right, 5.7σ). Left, map of the Cysteiny-AMP and Mg^{2+} within A domain in conformation 1_chain A (similar in chain B and conformation 3_chain A, not shown); middle, map of the Cysteine, AMP and Mg^{2+} within A domain in conformation 3_chain B; right, map of the AMP within A domain in conformation 2_chain A (similar in chain B, not shown).

4. The lack of clear distinction of the different structures and structural states is notable in Figure 3 and the associated discussion. There is a well-established rotation of the C-terminal subdomain of the adenylation

domain between the adenylate-forming and thioester-forming conformations. In none of the PchE conformations is the PCP bound to the adenylation domain. Comparison of the observed 155° subdomain rotation in Figure 3 to known adenylation domains states should be included. Are the observed conformations new or does PchE recapitulate the known states seen in prior structures that may now also serve to facilitate transport to the adenylation and epimerization domains? Finally, do the Epimerization and small C-terminal subdomain rotate as a rigid body, making significant domain contacts, or rotate independently?

A: We have now revised the Fig. 3 and the associated text to clearly label the adenylate-forming and thioester-forming conformations which PchE adopt, and added the comparison figure of A_{sub} domain rotation as Supplementary Fig. 19, please also see the answer on the “Lines 172 and 173. The dramatic conformational rotation of the small C-terminal subdomain is presented.....” of minor concerns; and yes, the observed A domain conformations of PchE adopt the two known states. The E domain significantly contacts the small A_{sub} domain with the buried surface of 865 Å², and they rotate as a rigid body (Supplementary Fig. 20).

Line 266: “The domain alteration strategy of the adenylation domain has been firmly established⁴⁶: the A_{sub} domain rotates relative to A_{core} to adopt distinct catalytic states for adenylation or thiolation reactions. Similar to the known states of homologs^{18,19,47,48} (Supplementary Fig. 19), in PchE, a large-scale ~155° A_{sub} rotation is mediated by the hinge residue⁴⁹ D944 and coupled to catalytic state switching (Fig. 3a). Chain A of conformation 3 adopts the adenylate-forming conformation, as featured by the universally conserved residue⁵⁰ K1325 in the A10 motif⁵¹, which forms a hydrogen bond with the 5' bridging oxygen of Cys-AMP (Fig. 3d), similar to the results found for the A domain of the DhbE⁴² structure. In the thioester-forming conformation of conformation 1_chain B, the aromatic residue F741 in the A4 motif⁵¹ was observed in two side chain conformations with a rotation angle of ~68° (Fig. 3e), similar to the H207 movement of 4CBL⁵² crystal structures. When the side chain of F741 points toward the active site, the pantetheine tunnel is closed, whereas the tunnel is restored to the open state when this aromatic residue rotates out of the active site, making the tunnel ready for the subsequent PCP-pPant entering. The observation of two F741 conformations in one chain indicates that these two states coexist in the map of the thioester-forming conformation, and these states were not completely separated in our cryo-EM dataset.”

18. Drake, E. J. *et al.* Structures of two distinct conformations of holo-non-ribosomal peptide synthetases. *Nature* **529**, 235–238 (2016).

19. Reimer, J. M., Aloise, M. N., Harrison, P. M. & Schmeing, T. M. Synthetic cycle of the initiation module of a formylating nonribosomal peptide synthetase. *Nature* **529**, 239–242 (2016).

42. May, J. J., Kessler, N., Marahiel, M. A. & Stubbs, M. T. Crystal structure of DhbE, an archetype for aryl acid activating domains of modular nonribosomal peptide synthetases. *Proc. Natl. Acad. Sci. U. S. A.* **99**, 12120–12125 (2002).

46. Gulick, A. M. Conformational dynamics in the acyl-CoA synthetases, adenylation domains of non-ribosomal peptide synthetases, and firefly luciferase. *ACS Chem. Biol.* **4**, 811–827 (2009).

47. Gulick, A. M., Starai, V. J., Horwill, A. R., Homick, K. M. & Escalante-Semerena, J. C. The 1.75 Å crystal structure of acetyl-CoA synthetase bound to adenosine-5'-propylphosphate and coenzyme A. *Biochemistry* **42**, 2866–2873 (2003).

48. Gulick, A. M., Lu, X. & Dunaway-Mariano, D. Crystal structure of 4-chlorobenzoate:CoA ligase/synthetase in the unliganded and aryl substrate-bound states. *Biochemistry* **43**, 8670–8679 (2004).

49. Wu, R., Reger, A. S., Lu, X., Gulick, A. M. & Dunaway-Mariano, D. The mechanism of domain alternation in the acyl-adenylate forming ligase superfamily member 4-chlorobenzoate: coenzyme A ligase. *Biochemistry* **48**, 4115–4125 (2009).

50. Branchini, B. R., Murtiashaw, M. H., Magyar, R. A. & Anderson, S. M. The role of lysine 529, a conserved residue of the acyl-adenylate-forming enzyme superfamily, in firefly luciferase. *Biochemistry* **39**, 5433–5440 (2000).

51. Marahiel, M. A., Stachelhaus, T. & Mootz, H. D. Modular peptide synthetases involved in nonribosomal peptide synthesis. *Chem. Rev.* **97**, 2651–2674 (1997).

52. Reger, A. S., Wu, R., Dunaway-Mariano, D. & Gulick, A. M. Structural characterization of a 140° domain movement in the two-step reaction catalyzed by 4-chlorobenzoate:CoA ligase. *Biochemistry* **47**, 8016–8025 (2008).

Fig. 3 Conformational changes coupled with A_{Sub} domain rotation. **a** Mediation of conformational changes by the A_{Sub} rotation relative to the A_{Core} domain. Structures of the A (blue) and E domains (green) in conformation 1_chain B and conformation 3_chain A are superposed and shown as ribbon representations to reveal the large-scale rotation of the $A_{\text{Sub}}-E_{\text{Sub}}$ didomains. **b** Close-up view of cysteinyl-AMP with Mg^{2+} binding to the substrate pocket of the A domain (mesh) in conformation 1_chain B with the open tunnel and a detailed view of the binding site residues (**c**, sticks) in conformation 3_chain A. **d** In the adenylate-forming conformation, the K1324 in the A10 motif forms the hydrogen bond with the oxygen of Cys-AMP; the distance is labeled. **e** Alternative side-chain conformation of F741 in the thioester-forming conformation (conformation 1_chain B). The $\sim 68^\circ$ switchable F741 in the A4 motif causes the pantetheine tunnel to transition from closed (F741 colored green) to open (F741 colored blue). The tunnel is shown as a cyan surface, and the direction of pPant entering is indicated. **f** Alternative side-chain conformations of residues R942, R1328, and R1329 (near cysteinyl-AMP) are shown with distances.

Supplementary Fig. 20 Structural comparison of the tailoring domain-embedded didomains of PchE and TioS¹⁹. **a** Superposition of the A_{sub}-E-A_{sub} didomains of the two conformations of PchE. The small difference between them indicates that they rotate as a rigid body. **b** The interface between A_{sub} and E domains of PchE, with the contacting residues shown as balls. **c** Cartoon representation of A_{sub}-M-A_{sub} didomain of TioS¹⁹ (PDB 5WMM) structure, with the interface residues (**d**) shown as balls. **e** Superposition of the tailoring domain-embedded structures on the interrupted A_{sub} domains, with the close-up views of the insertion sites (**f**).

19. Mori, S. *et al.* Structural basis for backbone N-methylation by an interrupted adenylation domain. *Nat. Chem. Biol.* **14**, 428–430 (2018).

5. Please state which structures were used for the different selections of electron density maps in ED Figure 6. Were maps of similar quality for all three final structures? While the electron density looks ok, it appears that the maps are presented with a “carve” radius that fails to depict density more than a set distance from the displayed atoms. This can be deceptive and the authors should present several regions with maps that allow the reader to better assess the maps. Electron density that supports ligand placement is also important, as well, to further expand on Point 3 above.

A: We have now revised the ED Figure 6 as Supplementary Fig. 7 to show several regions without carving, and stated which structures were used for displaying. The maps for all three final structures were not of similar quality. The integrity and resolution of the maps of conformation 1 and 3 are better than that of conformation 2. For cryo-EM maps of ligands, please see the answer of major point 3 and Supplementary Fig. 6.

Supplementary Fig. 7 Representative cryo-EM maps of the PchE domains. The displayed map including the ArCP, Cy, A_{core}, A_{sub}, and E domains is from the conformation 1_chain A structure. The map of the PCP domain is from conformation 3_chain A structure. The maps are shown as meshes, contoured at 0.012 (5.2 σ) for the Cy and A_{core} domains, and 0.010 (4.3 σ) for the ArCP, A_{sub}, and E domains, and 0.010 (4.4 σ) for the PCP domain. The figure was generated in ChimeraX. **a** The cryo-EM maps for each of the PchE domains, fitted with the cartoon represented atomic models (shown as lines). **b** Close-up views of the regions of each domain, fitted with the full-atom represented atomic models (shown as sticks).

6. The use of cryoDRGN to identify the overall trajectories is presented without details about how the program was run. Particularly as this is a new technique, sufficient experimental details should be provided in the Methods to understand what was done. This software package is not listed in the “software and code” section of the Reporting Summary document. Additionally, the “catalytic trajectory of PchE” section should include a brief introduction to the method being used and the nature and degree of confidence of the output trajectories. In particular, it is not clear how much experimental data supports the calculations used by this method or is it a model of realistic trajectories between experimental states? Finally, is there a validation score associated with this method that supports the different states? Is there greater confidence in some compared to others?

A: We have now added the full description for the cryoDRGN analyses in the Methods section with the software package listed in the Reporting Summary document, and we have included a brief introduction of this method as following (Line 308): “.....By creating neural networks based on cryo-EM particle images, cryoDRGN is able to encode particle images in a latent variable that describes these images using several

dimensions, learns the underlying structures including any motions, and decodes them into three-dimensional volumes (structures), including any structural heterogeneity in the cryo-EM dataset. Because cryo-EM images themselves do not contain temporal and kinetic information, cryoDRGN does not inform the kinetic order of conformational changes. Nevertheless, this method provides greater insights than just the single static structure.....”

For being proper and clearer: the cryo-EM maps reconstructed by cryoDRGN were realistic, but the trajectory, the possible trajectory was proposed and modeled. The experimental data supporting the calculation was based on the cryo-EM particle images (393,498) of the PchE consensus map, and it was a model between different experimental states. However, the cryo-EM data do not contain the temporal and kinetic information in itself, hence, the cryoDRGN software itself does not judge any trajectories. It does not give a validation score between different molecular trajectories. We propose the trajectory of PchE based on the previously well-established biochemical data and NRPSs structural knowledge. Therefore, we agree that the “modeled” is the proper word, not the “detected” (“Line 261. Again, a better description ofwhether the proper word here is “detected” or “modeled”.” of the Other minor concerns and corrections).

Methods section

Line 775: “Analysis of PchE dynamics by deep neural networks

The dynamics of PchE were analyzed by deep neural networks using cryoDRGN Version 0.3.0 and following the software protocols⁵⁹.

PchE cryo-EM dataset high-resolution training: An 8-dimensional latent variable model was trained for 50 epochs using a total of 393,498 particles with a full-resolution image size of 240 x 240 (1.1 Å per pixel). The image poses and CTF parameters were extracted from the 3.67-Å consensus map, which was reconstructed in RELION (Supplementary Fig. 3). The encoder and decoder architectures were 1024 x 3 (nodes per layer x layer).

Generation of six representative cryo-EM maps: After training, we used *k*-means clustering to partition the latent space into *k* regions, and *k*=400 was used with the latent encodings for our PchE cryo-EM dataset. The latent encoding closest in Euclidean distance to the *k*-means cluster center was defined as the ‘on-data’ cluster center, and volumes (cryo-EM maps) were generated at the ‘on-data’ cluster centers using the decoder network. Six structurally distinct representative structures were manually selected for visualization in Supplementary Fig. 22.

PchE molecular trajectory generation: For the graph traversal of the PchE in Supplementary Movie 2, the cryoDRGN software creates a nearest-neighbors graph from the latent encodings of the PchE particle images, in which a neighbor was defined if the Euclidean distance was below a threshold computed from the statistics of all pairwise distances. For the PchE cryo-EM dataset, we selected a value of 2.5 for the average number of neighbors across all nodes. We then used the latent encodings of the six representative maps shown in Supplementary Fig. 22 as the anchor points. Dijkstra’s algorithm was used to find the shortest path along the graph connecting these six anchor points, and a total of 120 cryo-EM maps were generated along the latent space data manifold at all the visited nodes. Subsequently, the series of structures were visualized, and the movie depicting the PchE dynamics was prepared using ChimeraX.”

59. Zhong, E. D., Bepler, T., Berger, B. & Davis, J. H. CryoDRGN: reconstruction of heterogeneous cryo-EM structures using neural networks. *Nat. Methods* **18**, 176–185 (2021).

Other minor concerns and corrections.

Line 54. PchE does not synthesize DHA alone, as it requires loading by PchD. Additionally, ED Figure 1 is overly simplified and does not accurately depict the role of PchG. A naïve reader could take the schematic as demonstrating the PchG catalyzes release of the product, rather than formation of the thiazolidine containing peptide while still bound to PchF. The legends or figure of ED Figure 1 should also explain that the Adenylation C-terminal subdomain is colored lighter blue.

A: Thanks. The sentence has been revised (Line 55): “After the loading of salicylate by PchD, PchE synthesizes dihydroaeruginosic acid.”, and we have also revised the ED Figure 1c and its legends as Supplementary Fig. 1c.

Supplementary Fig. 1c The biosynthetic pathway by which PchDEFG generates pyochelin. The catalytic active domains in each reaction step are highlighted, with non-active domains shown as transparent. The reaction is initiated by PchD catalytically activating salicylate, which is subsequently transferred to the ArCP domain of PchE and condensed with an A domain-activated L-cysteine by the Cy domain. The epimerization catalyzed by the E domain is performed at this stage. The upstream intermediate is transferred to the PCP domain of PchF and condensed with the second activated L-cysteine. Mature pyochelin is released by the TE domain of PchF, after the second thiazoline is reduced to thiazolidine by reductase (PchG). Note that the A_{sub} domain is colored lighter blue.

Lines 65-70. The description of existing prior structures fails to provide context for the current study. A more complete discussion of structures in multiple catalytic states will provide suitable background for understanding the advances of the current structures.

A: We have revised the paragraph for the more complete description of the multiple catalytic states. Line 67: “The current structural knowledge of full NRPS modules was derived from the termination module structures of SrfA-C¹⁷, EntF¹⁸, and AB3403¹⁸, the initial module structures of LgrA¹⁹ that adopt multiple catalytic states, the tri-domain module structure of FmoA3²⁰, the penta-domain module structure of ObiF²¹, the cross-module structure of DhbF²², and the LgrA di-module²³ structures. Depending on the dynamic A_{sub} and PCP domain positions, the NRPS synthetic cycle has been observed in different catalytic states: adenylation, thiolation, and condensation (or tailoring). In the adenylation state, the A_{sub} domain of AB3403¹⁸ adopts the closed state for substrate activation, and this finding has also been observed with LgrA¹⁹ and ObiF²¹. In the thiolation state of EntF¹⁸, DhbF²², and LgrA¹⁹, the A_{sub} domain rotates dramatically relative to A_{core} to adopt the thiolation-forming conformation, and the PCP domain contacts the A domain for loading the activated substrate. Subsequently, in the condensation or tailoring state, the PCP domain with its pAnt-tethered substrate binds at the C domain (or formylation domain of LgrA¹⁹) for peptide bond formation or formylation, whereas the A_{sub} domain is restored to the adenylation-state position for the next round of substrate activation. Together, these crystal structures have provided excellent insights into how these enzymes are assembled, catalyze reactions, and are structurally rearranged during catalytic state transitions²⁴.”

17. Tanovic, A., Samel, S. A., Essen, L.-O. & Marahiel, M. A. Crystal structure of the termination module of a nonribosomal peptide synthetase. *Science* **321**, 659–663 (2008).

18. Drake, E. J. *et al.* Structures of two distinct conformations of holo-non-ribosomal peptide synthetases. *Nature* **529**, 235–238 (2016).

19. Reimer, J. M., Aloise, M. N., Harrison, P. M. & Schmeing, T. M. Synthetic cycle of the initiation module of a formylating nonribosomal peptide synthetase. *Nature* **529**, 239–242 (2016).

20. Katsuyama, Y. *et al.* Structural and functional analyses of the tridomain-nonribosomal peptide synthetase FmoA3 for 4-methylxazoline ring formation. *Angew. Chem. Int. Ed.* **60**, 14554–14562 (2021).

21. Kreitler, D. F., Gemmill, E. M., Schaffer, J. E., Wenczewicz, T. A. & Gulick, A. M. The structural basis of N-acyl- α -amino- β -lactone formation catalyzed by a nonribosomal peptide synthetase. *Nat. Commun.* **10**, 3432 (2019).
22. Tarry, M. J., Haque, A. S., Bui, K. H. & Schmeing, T. M. X-ray crystallography and electron microscopy of cross- and multi-module nonribosomal peptide synthetase proteins reveal a flexible architecture. *Structure* **25**, 783-793.e4 (2017).
23. Reimer, J. M. *et al.* Structures of a dimodular nonribosomal peptide synthetase reveal conformational flexibility. *Science* **366**, eaaw4388 (2019).
24. Reimer, J. M., Haque, A. S., Tarry, M. J. & Schmeing, T. M. Piecing together nonribosomal peptide synthesis. *Curr. Opin. Struct. Biol.* **49**, 104–113 (2018).

Line 78. The structural basis of specificity of many NRPS adenylation domains is known. In particular, the original Stachelhaus code continues to provide a foundation for comparisons between key residues and the substrate binding. Is there something unique about Cysteine recognition from PchE? Does the conventional code predict an alternate substrate? Similarly, on Line 179 a predicted code is provided without citation nor an explanation of the diversity of Cys activating adenylation domains.

A: Yes, we agree. We have deleted the sentence and revised the Line 179 sentence with citation as follows (Line 256): “The structure-projection method successfully predicted the specificity-conferring code⁴⁴ for L-cysteine recognition and yielded the DLFNLSLIWK 10 AA signature sequence⁴⁵.”

44. Stachelhaus, T., Mootz, H. D. & Marahiel, M. A. The specificity-conferring code of adenylation domains in nonribosomal peptide synthetases. *Chem. Biol.* **6**, 493–505 (1999).

45. Röttig, M. *et al.* NRSPredictor2—a web server for predicting NRPS adenylation domain specificity. *Nucleic Acids Res.* **39**, W362–W367 (2011).

Line 81. Related to point 1 above, the authors need a transition that establishes that the current paper describes an unusual dimeric NRPS. The previous sentence states that NRPSs are monomers.

A: We have now included the transition section titled “PchE forms a homo-oligomeric quaternary structure” in the beginning of the Results section. Please see the answer of major point 1.

Line 86. Please include that PchD was added to load the salicylate on the ArCP domain of PchE. As written, the paper suggests that the chromatogram of ED Figure 2 should contain PchD at a 0.5:1.0 ratio with PchE. There is a very small trailing peak but it does not appear that Figure matches what is described in the paper.

A: We have now added the description of PchD and revised the size exclusion chromatography figure as Supplementary Fig. 2f, with the peak of PchD labeled. Please also see the answer of major point 1 for Supplementary Fig. 2f.

Line 107: “We purified full-length His-tagged PchD and Strep II-tagged PchE separately (Supplementary Fig. 2d, e). PchD is responsible for loading salicylate onto the ArCP domain of PchE. We then mixed PchD and PchE at a stoichiometric ratio of 0.5:1 in the presence of all the substrates and cofactors. PchE was then further purified using a final size-exclusion column (Supplementary Fig. 2f).”

Line 102. Perhaps replace “the A_{sub} domain is inserted by the E domain” with “the E domain is inserted into the A_{sub} domain” or “the A_{sub} domain is interrupted by the E domain”.

A: Line 124 revised as: “.....the A_{sub} domain is interrupted by the E domain.....”

Line 124. Replace “The ArCP/PCP contacts the Cy domain” with “The ArCP and the PCP contact the Cy domain on ...”

A: Line 184 revised as: “The ArCP and the PCP contact the Cy domain on.....”

Line 148. If Asp480 is abstracting a proton, it would be acting as a base.

A: Line 220 revised as: “D480 may act as a base and abstract a proton from the thiol sidechain of L-cysteinyll.....”

Figure 2. Again, please confirm that the ligands shown bound to PPant arms are present in experimental models and not merely modeled.

A: Thanks. The cryo-EM maps of the ligands have been shown in Supplementary Fig. 6. The two pPant-tethered substrates at the Cy domain pre-catalysis were not experimentally observed. They were merely modelled and bordered with dashed lines (Fig. 2 and annotated in its legend), in order to distinguish them from the experimentally observed ligands (bordered with solid lines in Fig. 2). Please also see the answer of major point 3 for Supplementary Fig. 6.

Line 137, as well as 158-160. Homologs EpoB and BmdB are included in the discussion without actually citing references to these papers. The Dowling and Bloudoff papers are previously cited (line 67) among “several individual excised NRPS domains”.

A: We have now cited the papers in the text.

Line 200: “The catalytic residue D480 aligns well with D1226 and D449 in BmdB³⁶ (r.m.s.d. 1.837 Å) and EpoB³⁷ (r.m.s.d. 2.127 Å) Cy homologs, respectively.”

Line 230: “By overlaying Cy with the two known Cy structures, we observed that the lid loop shifted ~6.5 Å (compared with BmdB³⁶) or ~17.8 Å (compared with EpoB³⁷) on the side of the PCP-binding region; the opposite loop (the latch⁴¹) moved ~11.5 Å on the side of the ArCP-binding region compared with EpoB³⁷ (Fig. 2d, left and right, respectively).”

36. Bloudoff, K., Fage, C. D., Marahiel, M. A. & Schmeing, T. M. Structural and mutational analysis of the nonribosomal peptide synthetase heterocyclization domain provides insight into catalysis. *Proc. Natl. Acad. Sci. U. S. A.* **114**, 95–100 (2017).

37. Dowling, D. P. *et al.* Structural elements of an NRPS cyclization domain and its intermodule docking domain. *Proc. Natl. Acad. Sci. U. S. A.* **113**, 12432–12437 (2016).

41. Keating, T. A., Marshall, C. G., Walsh, C. T. & Keating, A. E. The structure of VibH represents nonribosomal peptide synthetase condensation, cyclization and epimerization domains. *Nat. Struct. Biol.* **9**, 522–526 (2002).

Lines 172 and 173. The dramatic conformational rotation of the small C-terminal subdomain is presented without discussion of precedent from prior structural studies from our group and others. Similarly, lines 186-187 discuss the hinge without prior discussion or references.

A: We have now added the comparison figure of the small C-terminal A_{sub} domain rotation as Supplementary Fig. 19 and cited the papers in the text. Please also see the answer of major point 4.

Line 266: “The domain alteration strategy of the adenylation domain has been firmly established⁴⁶: the A_{sub} domain rotates relative to A_{core} to adopt distinct catalytic states for adenylation or thiolation reactions. Similar to the known states of homologs^{18,19,47,48} (Supplementary Fig. 19), in PchE, a large-scale ~155° A_{sub} rotation is mediated by the hinge residue⁴⁹ D944 and coupled to catalytic state switching (Fig. 3a).....”

18. Drake, E. J. *et al.* Structures of two distinct conformations of holo-non-ribosomal peptide synthetases. *Nature* **529**, 235–238 (2016).

19. Reimer, J. M., Aloise, M. N., Harrison, P. M. & Schmeing, T. M. Synthetic cycle of the initiation module of a formylating nonribosomal peptide synthetase. *Nature* **529**, 239–242 (2016).

46. Gulick, A. M. Conformational dynamics in the acyl-CoA synthetases, adenylation domains of non-ribosomal peptide synthetases, and firefly luciferase. *ACS Chem. Biol.* **4**, 811–827 (2009).

47. Gulick, A. M., Starai, V. J., Horswill, A. R., Homick, K. M. & Escalante-Semerena, J. C. The 1.75 Å crystal structure of acetyl-CoA synthetase bound to adenosine-5'-propylphosphate and coenzyme A. *Biochemistry* **42**, 2866–2873 (2003).

48. Gulick, A. M., Lu, X. & Dunaway-Mariano, D. Crystal structure of 4-chlorobenzoate:CoA ligase/synthetase in the unliganded and aryl substrate-bound states. *Biochemistry* **43**, 8670–8679 (2004).

49. Wu, R., Reger, A. S., Lu, X., Gulick, A. M. & Dunaway-Mariano, D. The mechanism of domain alternation in the acyl-adenylate forming ligase superfamily member 4-chlorobenzoate: coenzyme A ligase. *Biochemistry* **48**, 4115–4125 (2009).

Supplementary Fig. 19 Structural comparisons of the small C-terminal A_{sub} domain rotation. All structures are superposed onto the PchE A_{core} domain for A_{sub} domains rotation comparison. The A_{core} domains are colored blue with structures of comparison shown as transparent, the A_{sub} domains are highlighted in distinct colors, and the embedded E domain of PchE are colored green (transparent). **a** Superposition of the thioester-forming conformations of PchE A domain structure with LgrA¹³ (PDB 5ES8), EntF⁴ (PDB 5T3D), and Acs¹⁶ (Acetyl-CoA synthetase, PDB 1PG4) structures. **b** Superposition of the adenylate-forming conformations of PchE A domain structure with LgrA¹³ (PDB 5ES5), AB3403⁴ (PDB 4ZXH), and CBL¹⁷ (4-Chlorobenzoate:CoA ligase, PDB 1T5H) structures. **c** Individual superposition of different adenylating enzyme structures reveals the similar C-terminal subdomain movements. The rotation angles were reported by DynDom software¹⁸, and the direction of the arrowed lines indicate the catalytic states transitions from adenylate-forming to thioester-forming.

18. Hayward, S. & Berendsen, H. J. C. Systematic analysis of domain motions in proteins from conformational change: new results on citrate synthase and T4 lysozyme. *Proteins* **30**, 144–154 (1998).

Line 189. It is not clear what is meant by “the substrate L-cysteine adenylation function”.

A: Sorry for the inappropriate terminology. We have now deleted this sentence and revised this paragraph for description of the adenylate-forming and thioester-forming conformations of PchE (Line 266). Please also see the answer of major point 4.

Line 198. Reference for A8 and A9 motifs is necessary.

A: We have now cited the reference for the A8 and A9 motifs in Line 284: “.....the embedded E domain is inserted into the A_{sub} domain between the A8 and A9 motifs⁵¹.....”.

51. Marahiel, M. A., Stachelhaus, T. & Mootz, H. D. Modular peptide synthetases involved in nonribosomal peptide synthesis. *Chem. Rev.* **97**, 2651–2674 (1997).

Line 203. The active sites of Epimerization domains is well known from prior structural and functional work, as well as analysis homologous condensation domains and “might” is unnecessary here.

A: The word of “might” has been deleted from the sentence.

Line 217. As noted, methods need to be provided for the cryoDRGN technique.

A: We have now added the full description for the cryoDRGN analyses in the Methods section (Line775). Please also see the answer of major point 6.

Line 233. The prior literature on adenylation domains identifies an “open” conformation in which the C-terminal subdomain adopts a non-catalytic conformation. This conformation is presumed to be used for substrate binding and product release. The two catalytic conformations (adenylate-forming and thioester-forming) have both been described as closed, although the literature is not always consistent. Further, while the adenylation domains have a well-established pantetheine tunnel, it is not clear what is meant by “adenylation tunnel”. Is this a new structural feature that has not been observed previously? Is it simply the cavity formed in any of prior “open” conformations used for substrate binding.

A: We apologize that we have made a mistake, the same as in major point 4. We have revised this paragraph and corrected it throughout the manuscript. Please also see the answers of major point 4 and last minor point.

Lines 241-242. Please also clarify what is meant by “managed” in the discussion of functional interplay between the adenylation C-terminal subdomain and the epimerization domain.

A: We have revised this sentence as Line 341: “Due to the significant surface contact between A_{sub} and the E domain, the embedded epimerase simultaneously rotates as a rigid body.....”

Line 250. The core interaction between the adenylation and condensation (cyclization) domains is firmly established. Please cite prior studies and note whether the stable Cy/Aden core of PchE is identical, similar, or different from prior structures.

A: We have put this sentence ahead, in the paragraph of PchE overall architecture description (Line 127), and added the Supplementary Fig. 8 for the comparison of the Cy-A_{core} of PchE with prior C-A didomain interactions.

Line 127: “All PchEs in the three conformations are dimeric which each of them is composed of two chains. The stable Cy-A_{core} didomain forms the main body of the chain, which is nearly identical to the FmoA3²⁰ Cy-A_{core} domain arrangement. The Cy-A_{core} didomain is similar to the common C-A_{core} interactions of the previously characterized NRPSs^{17,18,21,23} with slightly varied rotational ranges of the positions of the C domain (Supplementary Fig. 8).”

17. Tanovic, A., Samel, S. A., Essen, L.-O. & Marahiel, M. A. Crystal structure of the termination module of a nonribosomal peptide synthetase. *Science* **321**, 659–663 (2008).

18. Drake, E. J. *et al.* Structures of two distinct conformations of holo-non-ribosomal peptide synthetases. *Nature* **529**, 235–238 (2016).

20. Katsuyama, Y. *et al.* Structural and functional analyses of the tridomain-nonribosomal peptide synthetase FmoA3 for 4-methylxazoline ring formation. *Angew. Chem. Int. Ed.* **60**, 14554–14562 (2021).

21. Kreitler, D. F., Gemmill, E. M., Schaffer, J. E., Wencewicz, T. A. & Gulick, A. M. The structural basis of N-acyl- α -amino- β -lactone formation catalyzed by a nonribosomal peptide synthetase. *Nat. Commun.* **10**, 3432 (2019).

23. Reimer, J. M. *et al.* Structures of a dimodular nonribosomal peptide synthetase reveal conformational flexibility. *Science* **366**, eaaw4388 (2019).

Supplementary Fig. 8 Comparisons of the Cy:A_{core} (C:A_{core}) conformations and domain positions. Superposition of the PchE structure (conformation 3_chain A) with FmoA3³ (PDB 6LTA), EntF⁴ (PDB 5T3D), LgrA_M2⁵ (PDB 6MFZ), ObiF1⁶ (PDB 6N8E), SrfA-C⁷ (PDB 2VSQ), and AB3403⁴ (PDB 4ZXH) on the A_{core} domain. The linear organization for each comparison is shown above, with all the domains labeled. The relative stable Cy:A_{core} (C:A_{core}) domains are displayed as ribbons, and the other dynamic domains are shown as lines. For each superposition, the PchE structure is colored solid with the comparison structure shown as transparent. The double-headed arrows indicate the relative rotation between the position of PchE Cy domain and C domain of five NRPS modules.

Line 251-252. It is not clear what is meant by "... a potential platform for interactions with upstream PchD and downstream PchF modules". Is there evidence that the core Cy/Aden could form a protein-protein interface with PchF? Wouldn't it be expected that the dynamic PCP would interact with PchF N-terminal Cyclization domain?

A: Yes, it is expected that the dynamic PCP would interact with PchF N-terminal Cyclization domain. We have now deleted this sentence, due to the lack of further evidence.

Line 261. Again, a better description of the cryoDGRN method could clarify whether the proper word here is "detected" or "modeled". Additionally, the conformational changes should be highlighted in the context of details of the structural cycle. In the transition from the I. Waiting to the II. Thiolation state, does the adenylation domain adopt the standard conformational change? Does the adenylation domain adopt the adenylate-forming conformation, with catalytic A10 motif lysine present in the active site, in this "waiting" state?

A: Yes, we thought that the "modeled" is the proper word here. We have added the full description of the cryoDRGN in the Methods section (Line 775, see also the answer of major point 6) and revised this paragraph for conformational changes (Line 326).

Line 326: "Based on biochemical knowledge and structural information of both static and dynamic conformations, we proposed the trajectory of the PchE catalytic cycle (Fig. 5, Supplementary Movie 2). The cycle initiates from state i. At this state, ArCP contacts the Cy domain at the donor site, representing the substrate donation conformation, the A domain adopts the thioester-forming conformation, and PCP is resting on the C-terminus site (i). PCP then moves and binds to the A domain, which adopts the thiolation state for loading the L-cysteine onto the pAnt arm of the PCP domain (ii). The A_{sub} domain then rotates dramatically together with the embedded tailoring E domain to adopt the adenylate-forming conformation with catalytic A10 motif⁵¹ lysine present in the active site for activation of the next L-cysteine molecule (iii), which is the same conformation as that found in the iv and v states. For the subsequent condensation reaction to occur, the PCP domain moves and binds at the acceptor docking site of Cy. The Cy domain catalyzes condensation and heterocyclization (iv). Subsequently, ArCP departs from the donor docking site on the Cy domain (v), quite likely interacts with upstream PchD to load a new molecule of salicylate, returns to Cy in conformation vi, restores the substrate donation conformation, and prepares to donate for the next round of condensation. At state vi, the A_{sub} domain rotates back to the position of the thioester-forming conformation. Due to the significant surface contact between A_{sub} and the E domain, the embedded epimerase simultaneously rotates as a rigid body; PCP shifts to the docking site of the E domain (vi), altering the stereochemistry of the thiazoline moiety. After delivery of the intermediate to the downstream PchF, the PCP domain returns to the C-terminus (i) and is then ready to begin the next round of the catalytic cycle."

Line 362: "Using this neural network calculation, we further modeled the thiolation and epimerization conformations and ultimately visualized the full catalytic cycle (Fig. 5)."

Andrew Gulick,
Buffalo, USA

Reviewer #3 (Remarks to the Author):

The manuscript reports three cryo-EM reconstructions of the PchE multi-domain protein from the biosynthetic pathway for pyochelin, a siderophore from the human pathogen *Pseudomonas aeruginosa*. Pyochelin is made by a nonribosomal peptide synthetase (NRPS) biosynthetic assembly line. The PchE protein has 3 enzyme domains and 2 carrier domains (ArCP-Cy-A-E-PCP) that carry out a total of 5 catalytic steps: the adenylation (A) domain (1) forms Cys-adenylate and (2) transfers Cys to the peptidyl carrier protein (PCP) domain, the cyclization domain (Cy) (3) condenses Cys with salicylate (tethered to an aryl carrier protein (ArCP) domain) and (4) cyclizes the product to a thiazoline, and an epimerase (E) domain (5) inverts the stereo center at the Cys C-alpha. The study is interesting because it adds a new molecular system to a small database of NRPS module structures. Large conformational changes are hallmarks of the catalytic cycle and this work illustrates an aspect of PchE flexibility.

We are grateful for the reviewer's positive and professional comments on our work, which are very helpful to the quality promotion of this manuscript. The points have been carefully addressed point by point, the figures have been revised as suggested, the new figures have been added for better supporting our conclusions, the manuscript has been revised and reorganized thoroughly for more clarity, and the manuscript has also been sent for language polishing by Springer Nature Company. Please see Supplementary_Information_Springer_nature.

The cryoEM reconstructions were computed from images taken from a single sample type: PchE in the presence of all its substrates (ATP, Cys and pre-loaded salicylate). The authors separated 3 PchE biochemical states by analysis of auto-picked particles. This approach of purification-by-computer greatly complicated the task of sorting particles into classes, especially as PchE is a dimer of independently functioning monomers that cannot be assumed to be in the same state. More definitive results could have been obtained by sequential addition of substrates and imaging of distinct and defined biochemical states.

A: Thank you for your suggestions and we absolutely agree that the results could have been more definitive by sequentially adding the substrates and subsequently imaging the corresponding structures. However, during the preliminary experimental test, we were unable to obtain the good quality cryo-EM particles without adding all the substrates, the PchE particles were heterogeneous and tended to aggregate. This problem hindered us from solving the structure. In the end, we only determined the PchE structures by adding all the substrates at the same time.

Moreover, it is not clear from the manuscript exactly which of the experimentally observed biochemical states correspond to which of the proposed PchE steps in the model of the catalytic cycle (Fig. 5). For example, the PCP domain was not visualized at either the A domain or the E domain (Fig. 1), so some of the key states shown in Fig. 5 are speculative.

A: Sorry for the unclarity. The experimentally observed structures of PchE are the high resolution three cryo-EM maps (Fig. 1 a-c, left) reconstructed by RELION software and corresponding atomic models (Fig. 1 a-c, middle) which correspond to the PchE biochemical states. The catalytic cycle in Fig. 5 is derived from in silico docking each PchE domain of the atomic models into the medium or low resolution cryo-EM maps reconstructed by cryoDRGN software which was used to detect remnant possible protein conformations in our cryo-EM dataset. So yes, the catalytic cycle in Fig. 5 is speculative and modeled, but based on the well-established biochemical knowledge and the structural information of PchE and other known NRPSs catalytic states transitions.

For more clarity, we have now added the complete description of the three PchE structures to clearly state the overall architecture of the three conformations and clearly establish what are the experimentally observed ligands, and revised the Fig. 5-related text to clearly state that it was modeled.

Please see details of the text on our answers of the related question of “The paper is not clearly written. While a few English problems.....” below.

Others have reported NRPS module structures in multiple states representing large conformational changes. The key papers are cited in the introduction, but this paper does not connect the PchE structures to these systems, which also display large motions of the A sub-domain.

A: For better connecting the PchE structures to these systems, we have now compared the A_{sub} domain rotation of PchE with several known structures which adopt similar large conformational movements (Line 266) as Supplementary Fig. 19, and compared the architecture and domain position of PchE with several previously reported NRPS modules (Line 127) as Supplementary Fig. 8. The papers were cited throughout the manuscript and figure captions. Then, we have revised the introduction paragraph for a more complete description of the multiple states of prior reported NRPS modules, to provide suitable background (Line 67) for readers to better understand the following PchE structures solved in this study.

Line 266: “The domain alteration strategy of the adenylation domain has been firmly established⁴⁶: the A_{sub} domain rotates relative to A_{core} to adopt distinct catalytic states for adenylation or thiolation reactions. Similar to the known states of homologs^{18,19,47,48} (Supplementary Fig. 19), in PchE, a large-scale $\sim 155^\circ$ A_{sub} rotation is mediated by the hinge residue⁴⁹ D944 and coupled to catalytic state switching (Fig. 3a).....”

Line 127: “All PchEs in the three conformations are dimeric which each of them is composed of two chains. The stable $Cy-A_{\text{core}}$ didomain forms the main body of the chain, which is nearly identical to the FmoA3²⁰ $Cy-A_{\text{core}}$ domain arrangement. The $Cy-A_{\text{core}}$ didomain is similar to the common $C-A_{\text{core}}$ interactions of the previously characterized NRPSs^{17,18,21,23} with slightly varied rotational ranges of the positions of the C domain (Supplementary Fig. 8).....”

Line 67: “The current structural knowledge of full NRPS modules was derived from the termination module structures of SrfA-C¹⁷, EntF¹⁸, and AB3403¹⁸, the initial module structures of LgrA¹⁹ that adopt multiple catalytic states, the tri-domain module structure of FmoA3²⁰, the penta-domain module structure of ObiF²¹, the cross-module structure of DhbfF²², and the LgrA di-module²³ structures. Depending on the dynamic A_{sub} and PCP domain positions, the NRPS synthetic cycle has been observed in different catalytic states: adenylation, thiolation, and condensation (or tailoring). In the adenylation state, the A_{sub} domain of AB3403¹⁸ adopts the closed state for substrate activation, and this finding has also been observed with LgrA¹⁹ and ObiF²¹. In the thiolation state of EntF¹⁸, DhbfF²², and LgrA¹⁹, the A_{sub} domain rotates dramatically relative to A_{core} to adopt the thiolation-forming conformation, and the PCP domain contacts the A domain for loading the activated substrate. Subsequently, in the condensation or tailoring state, the PCP domain with its pPant-tethered substrate binds at the C domain (or formylation domain of LgrA¹⁹) for peptide bond formation or formylation, whereas the A_{sub} domain is restored to the adenylation-state position for the next round of substrate activation. Together, these crystal structures have provided excellent insights into how these enzymes are assembled, catalyze reactions, and are structurally rearranged during catalytic state transitions²⁴.”

Supplementary Fig. 19 Structural comparisons of the small C-terminal A_{sub} domain rotation. All structures are superposed onto the PchE A_{core} domain for A_{sub} domains rotation comparison. The A_{core} domains are colored blue with structures of comparison shown as transparent, the A_{sub} domains are highlighted in distinct colors, and the embedded E domain of PchE are colored green (transparent). **a** Superposition of the thioester-forming conformations of PchE A domain structure with LgrA¹³ (PDB 5ES8), EntF⁴ (PDB 5T3D), and Acs¹⁶ (Acetyl-CoA synthetase, PDB 1PG4) structures. **b** Superposition of the adenylate-forming conformations of PchE A domain structure with LgrA¹³ (PDB 5ES5), AB3403⁴ (PDB 4ZXH), and CBL¹⁷ (4-Chlorobenzoate:CoA ligase/synthetase in the unliganded and aryl substrate-bound states, PDB 1T5H) structures. **c** Individual superposition of different adenylating enzyme structures reveals the similar C-terminal subdomain movements. The rotation angles were reported by DynDom software¹⁸, and the direction of the arrowed lines indicate the catalytic states transitions from adenylate-forming to thioester-forming.

- Drake, E. J. *et al.* Structures of two distinct conformations of holo-non-ribosomal peptide synthetases. *Nature* **529**, 235–238 (2016).
- Reimer, J. M., Aloise, M. N., Harrison, P. M. & Schmeing, T. M. Synthetic cycle of the initiation module of a formylating nonribosomal peptide synthetase. *Nature* **529**, 239–242 (2016).
- Gulick, A. M., Starai, V. J., Horswill, A. R., Homick, K. M. & Escalante-Semerena, J. C. The 1.75 Å crystal structure of acetyl-CoA synthetase bound to adenosine-5'-propylphosphate and coenzyme A. *Biochemistry* **42**, 2866–2873 (2003).
- Gulick, A. M., Lu, X. & Dunaway-Mariano, D. Crystal structure of 4-chlorobenzoate:CoA ligase/synthetase in the unliganded and aryl substrate-bound states. *Biochemistry* **43**, 8670–8679 (2004).
- Hayward, S. & Berendsen, H. J. C. Systematic analysis of domain motions in proteins from conformational change: new results on citrate synthase and T4 lysozyme. *Proteins* **30**, 144–154 (1998).

Supplementary Fig. 8 Comparisons of the Cy:A_{core} (C:A_{core}) conformations and domain positions. Superposition of the PchE structure (conformation 3_chain A) with FmoA3³ (PDB 6LTA), EntF⁴ (PDB 5T3D), LgrA_M2⁵ (PDB 6MFZ), ObiF1⁶ (PDB 6N8E), SrIA-C⁷ (PDB 2V5SQ), and AB3403⁴ (PDB 4ZXH) on the A_{core} domain. The linear organization for each comparison is shown above, with all the domains labeled. The relative stable Cy:A_{core} (C:A_{core}) domains are displayed as ribbons, and the other dynamic domains are shown as lines. For each superposition, the PchE structure is colored solid with the comparison structure shown as transparent. The double-headed arrows indicate the relative rotation between the position of PchE Cy domain and C domain of five NRPS modules.

3. Katsuyama, Y. *et al.* Structural and functional analyses of the tridomain-nonribosomal peptide synthetase FmoA3 for 4-methylloxazoline ring formation. *Angew. Chem. Int. Ed.* **60**, 14554–14562 (2021).
4. Drake, E. J. *et al.* Structures of two distinct conformations of holo-non-ribosomal peptide synthetases. *Nature* **529**, 235–238 (2016).
5. Reimer, J. M. *et al.* Structures of a dimodular nonribosomal peptide synthetase reveal conformational flexibility. *Science* **366**, eaaw4388 (2019).
6. Kreitler, D. F., Gemmell, E. M., Schaffer, J. E., Wenczewicz, T. A. & Gulick, A. M. The structural basis of N-acyl- α -amino- β -lactone formation catalyzed by a nonribosomal peptide synthetase. *Nat. Commun.* **10**, 3432 (2019).
7. Tanovic, A., Samel, S. A., Essen, L.-O. & Marahiel, M. A. Crystal structure of the termination module of a nonribosomal peptide synthetase. *Science* **321**, 659–663 (2008).

The paper is not clearly written. While a few English problems contribute to the lack of clarity, by far the larger problem is the incomplete description that makes it very difficult to connect that experimentally determined structures to the model. In Fig. 2, the cage and solid renderings appear to be of the protein surface, not the EM map - this should be stated in the caption.

A: Thanks. We have now revised the English problems of the manuscript at our best, and stated that it is the protein surface representation (shown in mesh) of the PchE atomic model in the caption of Fig. 2.

Line 388: “.....The protein surfaces of the atomic models are shown as mesh in the cage and solid renderings (middle).”

For the point of what was experimentally determined and what was modeled, we have revised the manuscript from two aspects:

1. We have now added the complete description of the three PchE structures to state the overall architecture of the three conformations and clearly establish what are the experimentally observed ligands (Line 127).
2. We have revised the catalytic cycle (Fig. 5) related text to clearly state that it was modeled (Line 362), and added the description of the cryoDRGN software in the Methods section (Line 775).

Line 127: “All PchEs in the three conformations are dimeric which each of them is composed of two chains. The stable Cy- A_{core} didomain forms the main body of the chain, which is nearly identical to the FmoA3²⁰ Cy- A_{core} domain arrangement. The Cy- A_{core} didomain is similar to the common C- A_{core} interactions of the previously characterized NRPSs^{17,18,21,23} with slightly varied rotational ranges of the positions of the C domain (Supplementary Fig. 8). Relative to Cy- A_{core} , there are multiple highly dynamic domains, namely, the ArCP, PCP, and A_{sub} -E- A_{sub} domains, and these domains allow the identification of different catalytic states. In conformation 1 (Fig. 1a), chains A and B are identical with an r.m.s.d. of 0.037 Å. The ArCP domains of both chains bind at the donor sites of the Cy domains, with the salicyl-pPant ligands observed, representing the substrate donation states. The A_{core} with interrupted A_{sub} domains (complexed with Cys-AMP and Mg^{2+} ligands) are in the thioester-forming conformation but without the PCP domain captured. The E domain in this conformation may represent the epimerization conformation because the low-resolution map of the PCP domain was only observed binding to the E domain in this position (described later in Fig. 5). In conformation 2 (Fig. 1b), both ArCP and PCP domains in chain A bind to the Cy domain, representing the condensation state; the pPant arm of the ArCP domain was observed without tethered salicylate, and the pPant of PCP is merely modeled in Fig. 2b. In chain B, only the PCP domain contacts the acceptor site of the Cy domain, and the pPant arm of the PCP domain was obtained. Both chains have their A_{sub} domain rotated relative to the A_{core} (complexed with AMP), resulting in the adenylate-forming conformations. The superposition between two chains gives an r.m.s.d. of 0.851 Å (without ArCP). In conformation 3 (Fig. 1c), the PCP domain of chain A binds at the acceptor site of the Cy domain, with the hydroxyphenylthiazoliny-pPant ligand observed, representing the post-condensation state. The A domain (complexed with Cys-AMP and Mg^{2+} ligands) adopts the adenylate-forming conformation. Chain B duplicates the catalytic state as both chains of conformation 1. In addition to the dynamic domains, the Cy- A_{core} domains between two chains of conformation 3 have an r.m.s.d. of 0.566 Å. In summary, because each protomer of the three conformations is observed in the different catalytic states, dimeric PchE may still be monofunctional³⁵, with two chains forming separate catalytic units.”

Line 362: “Using this neural network calculation, we further modeled the thiolation and epimerization conformations and ultimately visualized the full catalytic cycle (Fig. 5).”

Line 775: **“Analysis of PchE dynamics by deep neural networks**

The dynamics of PchE were analyzed by deep neural networks using cryoDRGN Version 0.3.0 and following the software protocols⁵⁹.

PchE cryo-EM dataset high-resolution training: An 8-dimensional latent variable model was trained for 50 epochs using a total of 393,498 particles with a full-resolution image size of 240 x 240 (1.1 Å per pixel). The image poses and CTF parameters were extracted from the 3.67-Å consensus map, which was reconstructed in RELION (Supplementary Fig. 3). The encoder and decoder architectures were 1024 x 3 (nodes per layer x layer).

Generation of six representative cryo-EM maps: After training, we used *k*-means clustering to partition the latent space into *k* regions, and *k*=400 was used with the latent encodings for our PchE cryo-EM dataset. The latent encoding closest in Euclidean distance to the *k*-means cluster center was defined as the ‘on-data’ cluster center, and volumes (cryo-EM maps) were generated at the ‘on-data’ cluster centers using the decoder network. Six structurally distinct representative structures were manually selected for visualization in Supplementary Fig. 22.

PchE molecular trajectory generation: For the graph traversal of the PchE in Supplementary Movie 2, the cryoDRGN software creates a nearest-neighbors graph from the latent encodings of the PchE particle images, in which a neighbor was defined if the Euclidean distance was below a threshold computed from the statistics of all pairwise distances. For the PchE cryo-EM dataset, we selected a value of 2.5 for the average number of neighbors across all nodes. We then used the latent encodings of the six representative maps shown in Supplementary Fig. 22 as the anchor points. Dijkstra’s algorithm was used to find the shortest path along the graph connecting these six anchor points, and a total of 120 cryo-EM maps were generated along the latent space data manifold at all the visited nodes. Subsequently, the series of structures were visualized, and the movie depicting the PchE dynamics was prepared using ChimeraX.”

59. Zhong, E. D., Bepler, T., Berger, B. & Davis, J. H. CryoDRGN: reconstruction of heterogeneous cryo-EM structures using neural networks. *Nat. Methods* **18**, 176–185 (2021).

The cryo-EM structures appear to be of good quality. The atomic model is a good fit to the map of highest resolution (2.97 Å). However, some details are not presented clearly. For example, in Fig. S6 with detailed densities, it is not stated which map is displayed, nor what is the contour level. From my inspection of the model-to-map fit, I know that a contour level where side chains are clearly visible would have no density for the aryl group on the ArCP. Thus I suspect that the images in this figure have different density contour levels.

A: Thank you for your highly experienced suggestion. We have now revised the Fig. S6 as Supplementary Fig. 6 and Supplementary Fig. 7 to show the cryo-EM maps of experimentally observed ligands and each of the PchE domains, respectively. Yes, the displayed maps are in different contour levels, and we have now stated in the figure legends that which structures and what contour levels were used in these two figures.

Supplementary Fig. 6 Representative cryo-EM maps of the PchE ligands. The cryo-EM maps of the ligands observed in this study, shown as meshes, are displayed at different contour levels and generated in ChimeraX. The colors and labels of distinct conformations and chains are corresponding to Fig. 1. **a** Map of the Ser46 salicyl-pPant arm of the ArCP domain in conformation 1_chain A, contoured at 0.007 (3.0 σ). This ligand was also observed similarly in conformation 3_chain B (not shown). **b** Map of the Ser1385 Hydroxyphenylthiazoliny-pPant arm of the PCP domain in conformation 3_chain A, contoured at 0.006 (2.8 σ). **c** Map of the Ser46 pPant arm of the ArCP domain in conformation 2_chain A, contoured at 0.010 (4.7 σ). **d** Map of the Ser1385 pPant arm of the PCP domain in conformation 2_chain B, contoured at 0.006 (2.8 σ). **e** Maps were contoured at 0.012 (Left, 5.2 σ ; middle, 5.5 σ ; right, 5.7 σ). Left, map of the Cysteiny-AMP and Mg²⁺ within A domain in conformation 1_chain A (similar in chain B and conformation 3_chain A, not shown); middle, map of the Cysteine, AMP and Mg²⁺ within A domain in conformation 3_chain B; right, map of the AMP within A domain in conformation 2_chain A (similar in chain B, not shown).

Supplementary Fig. 7 Representative cryo-EM maps of the PchE domains. The displayed map including the ArCP, Cy, A_{core}, A_{sub}, and E domains is from the conformation 1_chain A structure. The map of the PCP domain is from conformation 3_chain A structure. The maps are shown as meshes, contoured at 0.012 (5.2 σ) for the Cy and A_{core} domains, and 0.010 (4.3 σ) for the ArCP, A_{sub}, and E domains, and 0.010 (4.4 σ) for the PCP domain. The figure was generated in ChimeraX. **a** The cryo-EM maps for each of the PchE domains, fitted with the cartoon represented atomic models (shown as lines). **b** Close-up views of the regions of each domain, fitted with the full-atom represented atomic models (shown as sticks).

PchE is a dimer by the EM analysis, but the apparent MW in solution is not stated, despite the lovely gel filtration profile in Fig. S2c. Dimer formation is unusual for an NRPS module, so it should probably be labeled simply as a rare occurrence and not a major finding.

A: Thanks. We have revised the Fig. S2c with the molecular weight (MW) standards labeled, and the apparent MW of PchE stated in the figure legends of Supplementary Fig. 2f, and yes, we agree that the dimer formation of NRPS is unusual and remain a minority nature in the field. We have weakened this observation and clearly stated that it is a minor phenomenon in the manuscript, and compared our structure with the very recently reported FmoA3 structure which also adopt a similar dimeric architecture in Supplementary Fig. 10, to show a direct architecture comparison of this kind of rare occurrence. Line 347: "Among the NRPS family of megaenzymes, homodimerization is a minor and unique phenomenon....."

Supplementary Fig. 2 Purification and oligomeric state analysis of PchE in solution. **a** SDS-PAGE analysis of purified PchE, PchD and PchF. The predicted (theoretical) molecular weights are labeled respectively. **b** Native-PAGE analysis of PchE, PchF, and the mixture of PchE and PchF. A 4-20% gradient was used. Note that the migration of PchE in the gel is slower than that of PchF. **c** Size exclusion chromatography (SEC, Superose 6 Increase) profile of PchE and PchF. Note that PchE elutes earlier than PchF. The apparent molecular weights (WMs) of the two proteins are approximately 484 kDa (PchE) and 322 kDa (PchF). The obtained overestimated apparent WMs of PchE and PchF may be due to their elongated and dynamic conformations, because the shape and Stokes radius of the samples can vary significantly from the largely commercial globular standard proteins, which the extended shape of proteins can easily result in an anomalously earlier elution from the size exclusion chromatographic column^{1,2}. Despite the overestimated apparent MWs of the two proteins, the hypothesis of PchE being homo-oligomeric is reasonable, based on the MWs comparison between Supplementary Fig. 2a and b-c. **d** SDS-PAGE analysis of purified PchE with a C-terminus Strep-tag II and PchD with an N-terminus His-tag. **e** Native-PAGE analysis of PchE. A 4-20% gradient was used. **f** Size exclusion chromatography profile of PchE used for cryo-EM. One representative result from at least three independent experiments is shown (a, b, d, and e). Source data are provided as a Source Data file.

1. Erickson, H. P. Size and shape of protein molecules at the nanometer level determined by sedimentation, gel filtration, and electron microscopy. *Biol. Proced. Online* **11**, 32–51 (2009).
2. Sorensen, B. R. & Shea, M. A. Calcium binding decreases the stokes radius of calmodulin and mutants R74A, R90A, and R90G. *Biophys. J.* **71**, 3407–3420 (1996).

Supplementary Fig. 10 Structural comparison of the dimeric organization between PchE and FmoA3³. **a** Comparison of the two overall architectures and linear organizations. **b** Comparison of the two inter-helix arrangements. Close-up views of interface helices (with angles labeled) are bordered with dashed lines. **c** Comparison of the two (Cy-A_{core})_{chain A}-(A_{core}-Cy)_{chain B} quadrangular stable cores, shown in front, side, and top views (top panel to bottom, respectively). Left, chains A of PchE and FmoA3 are superposed and shown as lines, with chains B shown as cylinders for comparison purpose. The tilt angle between chain B of PchE and FmoA3 is labeled (bottom). Right, schematics showing the inter-chain arrangements of PchE and FmoA3. **d** Comparison of the two dimeric interfaces. The interacting residues are colored and shown as sticks. Bottom, Schematics showing contacts between chains with only chains B shown as surface representation.

3. Katsuyama, Y. *et al.* Structural and functional analyses of the tridomain-nonribosomal peptide synthetase FmoA3 for 4-methyloxazoline ring formation. *Angew. Chem. Int. Ed.* **60**, 14554–14562 (2021).

REVIEWER COMMENTS

Reviewer #1 (Remarks to the Author):

The revised manuscript "Catalytic trajectory of a dimeric nonribosomal peptide synthetase subunit with an inserted epimerase domain" is much improved.

Ligands: The maps for the ppant intermediates (supplemental fig 6) are not particularly definitive for their conformations. Also, as previously reviewed for the other map figure in my original review: The authors are showing maps "carved" close to their model. This is fine, but can be used to artificially make the maps look better than they use a low carving distance. The carving distance should not be less than 2.5-3 Å, and must be stated in the figure legend.

Supplemental figure 14: Please modify "Linear to heterocyclic product Ratio (%)" – this is an odd way of stating the ratio of linear to heterocyclic product. For example the D480A has a 1.9% : 1% ratio of linear cyclic (as % of wt cyclic). How is that a ratio of 463.4?

Reviewer #2 (Remarks to the Author):

I thank the authors for their careful response to the initial round of reviews. It was comforting to see that all three reviewers were largely consistent in our concerns and I do feel that the authors have done a nice job of addressing them. The revisions result in an improved manuscript that properly places these new structures in the context of the state of the field.

Additionally, the clearer description of the cryoDRGN approach to explore the potential movement of the domains provides a potential approach for the full structural cycle.

The addition of the electron density is also critical for the reader to assess the models. The density of Supplementary Figure 7 appears adequate, particularly for placing models of individual domains for which high resolution structures exist. However, the density shown for the ligands in SI Figure 6 is much more tenuous; nonetheless, I believe that the ligands can still be included in the models and reported as such. The placement of the adenylate ligand is (as shown in SI Figure 18) similar to known structures so it is likely positioned correctly. Further, I don't believe that the precise position of the ligands is relevant for the most important conclusions of this paper that instead focus on the overall domain architecture and conformational dynamics of the potential different states.

Most important, the inclusion of the density in the current version of the paper accurately depicts the quality of the density and allows the reader to judge for himself or herself.

There is something unconventional about the density in Supplemental Figure 6. The grid lines are thick in places however in others adopt a more faint line. I cannot tell if this is because a (transparent?) ligand bond is sitting on the density. If so, this could be interpreted as showing that the Mg ions in Panel E (left and middle) are not embedded within the density but rather are sitting on top of it. Additionally, until I zoomed in and saw very faint grid lines in panel D, I thought that the entire pantetheine was sitting atop the density. If one looks at the phosphate of panel A, where the grid lines are all thick, and the phosphate of panel C, where the gridlines are thinner, it appears that only the phosphate of Panel A is truly embedded within the density. Please review and confirm that the density depiction is appropriate.

Andrew Gulick

Reviewer #3 (Remarks to the Author):

The authors have clarified the confusing points in the original submission and have added considerable detail to explain which of the presented results are from direct experiment and which are modeled. The paper adds to the growing, but still small, database of NRPS module structures and now includes sufficient connection to prior results from other investigators.

Point-by-point response to the reviewers (the second round)

*Reviewer Comments: Black, Helvetica, 10

*Author Responses: Blue, Helvetica, 10

Reviewer #1 (Remarks to the Author):

The revised manuscript "Catalytic trajectory of a dimeric nonribosomal peptide synthetase subunit with an inserted epimerase domain" is much improved.

Again, we would like to express our gratitude to the Reviewer #1 for your professionalism. Your critical comments have truly improved this manuscript, particularly in a way of citing and connecting our work with priorly published results from other research groups, and also helped us to further understand the NRPSs megaenzymes and writing the manuscript more scholarly. We can tell that you must be a highly experienced scientist who have made great contributions in this field, thank you very much indeed.

Ligands: The maps for the pPant intermediates (supplemental fig 6) are not particularly definitive for their conformations. Also, as previously reviewed for the other map figure in my original review: The authors are showing maps "carved" close to their model. This is fine, but can be used to artificially make the maps look better than they use a low carving distance. The carving distance should not be less than 2.5-3 Å, and must be stated in the figure legend.

A: We agree that the maps for pPant intermediates are a bit tenuous. The resolutions of the ligands solved by cryo-EM in this study are not absolutely definitive, compared with prior high-resolution structures solved by X-ray crystallography. Nevertheless, the placements of the ligands are similar to the known structures so they are likely positioned correctly. We believe that the ligands can still be included in the models and reported as such, as the Reviewer #2 suggested. We have now stated in the figure legend of Supplementary Fig. 6 that the carving distance is 3 Å and the maps for the pPant intermediates are not particularly definitive. The maps of Supplementary Fig. 7 were not carved.

Supplementary Fig. 6 Representative cryo-EM maps of the PchE ligands. The cryo-EM maps of the ligands observed in this study, shown as meshes and surfaces in two different rotational views, are displayed at different contour levels and generated in ChimeraX. The carving distance is 3 Å. Note that the maps for the pPant intermediates are not particularly definitive for their absolute conformations.

Supplemental figure 14: Please modify "Linear to heterocyclic product Ratio (%)" – this is an odd way of stating the ratio of linear to heterocyclic product. For example the D480A has a 1.9% : 0.4% ratio of linear to cyclic (as % of wt cyclic). How is that a ratio of 463.4?

A: Sorry for the unclarity. This ratio is not the same as the left histogram which the peak area of each mutant was divided by that of WT cyclic (%). We have revised this title as "The average peak area of linear to heterocyclic product ratio (as -fold per mutant)". For example, the D480A itself has a 1.9%:0.4% ratio of linear:cyclic, the division results in a 4.634-fold (the exact numeric values of average peak area amounts are provided in the Source Data file).

Supplementary Fig. 14 Activity assay of structure-guided mutations in the Cy domain. **a** The plot shows relative activities of the heterocyclic (black) and linear (purple, the intermediate) products of the key residue mutants in the Cy domain (as % of the average peak area of WT heterocyclic product). The product ratios of linear to heterocyclic are labeled in the right table (as -fold of the average peak area per mutant).

Reviewer #2 (Remarks to the Author):

I thank the authors for their careful response to the initial round of reviews. It was comforting to see that all three reviewers were largely consistent in our concerns and I do feel that the authors have done a nice job of addressing them. The revisions result in an improved manuscript that properly places these new structures in the context of the state of the field.

Dear Professor Andrew Gulick,

Again, we would like to express our respects to you for your professionalism and long-term dedicated contributions in the NRPSs field. Your comments and concerns during the entire review processes have been a treasure to us, and we are truly happy for your highly positive evaluation on our work and the responses to the initial round of reviews.

We can't wait to see more of your research works in the future, thank you!

Additionally, the clearer description of the cryoDRGN approach to explore the potential movement of the domains provides a potential approach for the full structural cycle.

A: We couldn't agree more. We believe that this new kind of computational approach will help the researchers to better capture the potential motions of the NRPSs biosynthetic machines, and provide a more complete understanding of the NRPSs molecular trajectory in the future.

The addition of the electron density is also critical for the reader to assess the models. The density of

Supplementary Figure 7 appears adequate, particularly for placing models of individual domains for which high resolution structures exist. However, the density shown for the ligands in SI Figure 6 is much more tenuous; nonetheless, I believe that the ligands can still be included in the models and reported as such. The placement of the adenylate ligand is (as shown in SI Figure 18) similar to known structures so it is likely positioned correctly. Further, I don't believe that the precise position of the ligands is relevant for the most important conclusions of this paper that instead focus on the overall domain architecture and conformational dynamics of the potential different states.

A: It's really comforting and relieved!

Most important, the inclusion of the density in the current version of the paper accurately depicts the quality of the density and allows the reader to judge for himself or herself.

There is something unconventional about the density in Supplemental Figure 6. The grid lines are thick in places however in others adopt a more faint line. I cannot tell if this is because a (transparent?) ligand bond is sitting on the density. If so, this could be interpreted as showing that the Mg ions in Panel E (left and middle) are not embedded within the density but rather are sitting on top of it. Additionally, until I zoomed in and saw very faint grid lines in panel D, I thought that the entire pantetheine was sitting atop the density. If one looks at the phosphate of panel A, where the grid lines are all thick, and the phosphate of panel C, where the gridlines are thinner, it appears that only the phosphate of Panel A is truly embedded within the density. Please review and confirm that the density depiction is appropriate.

A: Sorry for the unclarity, we believe the perception of the ligands being sitting atop the density is mainly due to the views of the specific orientation of the maps prepared in Chimera-X. We have reviewed and confirmed that the ligands are embedded within the density, and revised Supplementary Fig. 6 to show the different rotational viewpoints of the maps.

Supplementary Fig. 6 Representative cryo-EM maps of the PchE ligands. The cryo-EM maps of the ligands observed in this study, shown as meshes and surfaces in two different rotational views, are displayed at different contour levels and generated in ChimeraX.

Reviewer #3 (Remarks to the Author):

The authors have clarified the confusing points in the original submission and have added considerable detail to explain which of the presented results are from direct experiment and which are modeled. The

paper adds to the growing, but still small, database of NRPS module structures and now includes sufficient connection to prior results from other investigators.

Again, we would like to show our appreciation to the Reviewer #3 for evaluating our work. In particular, your comments about “this paper does not connect the PchE structures to prior results from other investigators” for our manuscript have hit the nail on the head, and truly helped us to better organize our work, thank you very much!